# The Relative Instability of Model Comparison with Cross-validation

**Alexandre Bayle** [1]  **Lucas Janson** [1]  **Lester Mackey** [2]

## Abstract

Cross-validation (CV) is known to provide asymptotically exact tests and confidence intervals for model improvement but only when the model comparison is *relatively stable*. Surprisingly, we prove that even simple, individually stable models can generate relatively unstable comparisons, calling into question the validity of CV inference. Specifically, we show that the Lasso and its close cousin, soft-thresholding, generate relatively unstable comparisons and invalid CV inferences, even in the most favorable of learning settings and even when both models are individually stable. These findings highlight the importance of verifying relative stability before deploying CV for model comparison.

## 1. Introduction

In machine learning, statistics, and the natural sciences, cross-validation (CV) (Stone, 1974; Geisser, 1975) is routinely used to compare the performance of learning algorithms (see, e.g., Yates et al., 2023; Bradshaw et al., 2023). A common practice is to pair a CV performance estimate with a hypothesis test or confidence interval that accounts for uncertainty. Yet the validity of such uncertainty quantification has been poorly understood until recently, and it is now understood to be closely related to notions of algorithmic stability (Austern & Zhou, 2020; Bayle et al., 2020). A wide variety of learning algorithms are known to be stable (Devroye & Wagner, 1979a; Bousquet & Elisseeff, 2002; Elisseeff et al., 2005; Hardt et al., 2016; Celisse & Guedj, 2016; Arsov et al., 2019) and hence also enjoy asymptotically exact CV inference when assessed individually. However, as we will show, stable algorithms do not always give rise to stable comparisons, calling into question the validity of CV tests and intervals for some model

[1]Department of Statistics, Harvard University, Cambridge, MA, USA [2]Microsoft Research New England, Cambridge, MA, USA. Correspondence to: Alexandre Bayle <alexandre_bayle@alumni.harvard.edu>.

*Proceedings of the 43$^{rd}$ International Conference on Machine Learning*, Seoul, South Korea. PMLR 306, 2026. Copyright 2026 by the author(s).

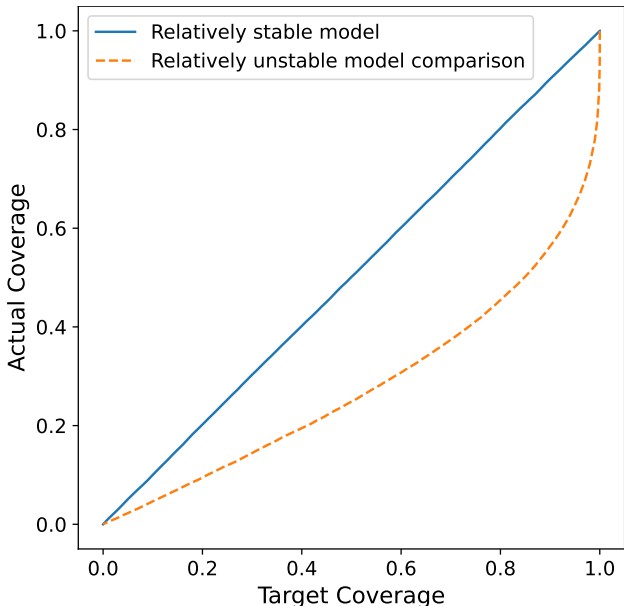

*Figure 1.* The cross-validation central limit theorem (Bayle et al., 2020) yields accurate coverage for the relatively stable Lasso algorithm but severely undercovers for the relatively unstable comparison of two Lasso fits. See Appendix B for full experiment details.

comparisons. As a concrete illustration, Figure 1 displays CV-based confidence intervals which accurately cover the performance of a single stable algorithm but sorely undercover the comparison between two stable instances of that same model with slightly different tuning parameters.

**Our contributions** We expose an important failure mode of cross-validation inference that is tightly linked to the *relative* stability of model comparisons, a notion formally introduced in Section 2. In Section 3 we show that, even in a simple learning setting, the popular Lasso (Tibshirani, 1996) and soft-thresholding (ST) (Donoho & Johnstone, 1994) algorithms fail to generate relatively stable comparisons (Theorem 3.1), despite being individually relatively stable (Theorem 3.2). As we explain in Section 4, this lack of relative stability can render standard CV-based hypothesis tests and confidence intervals invalid, yielding the severe undercoverage that we observe in Figure 1 and in our extended experiments in Section 5.

**Related work** The importance of the stability of an algorithm with respect to its generalization error (Bousquet & Elisseeff, 2002) has prompted numerous studies of the stability of popular classes of algorithms (Bousquet & Elisseeff, 2002; Elisseeff et al., 2005; Hardt et al., 2016; Celisse & Guedj, 2016; Arsov et al., 2019). Across the years, different notions of stability have been introduced (Devroye & Wagner, 1979a;b; Kearns & Ron, 1999; Kutin & Niyogi, 2002; Kale et al., 2011; Kumar et al., 2013), and, building upon the domain of algorithmic stability, multiple studies (Kale et al., 2011; Kumar et al., 2013; Celisse & Guedj, 2016; Austern & Zhou, 2020; Bayle et al., 2020) have established interesting relationships between the theoretical properties of cross-validation and the stability properties of the algorithms involved. Austern & Zhou (2020) and Bayle et al. (2020) derive central limit theorems and consistent variance estimators for the CV estimator under sufficient conditions on algorithmic stability. The former paper does so with the *mean-square stability* (Kale et al., 2011) and the latter with *loss stability* (Kumar et al., 2013), both of which are known to decay to zero for a variety of algorithms. However, to our knowledge, no prior works have explicitly studied the sufficient conditions for asymptotic normality in the case when the asymptotic variance in these central limit theorems goes to zero, as would be expected in the common scenario of comparing the performance of two algorithms that converge to the same prediction rule (e.g., if they are both consistent for the optimal prediction rule). This is the focus of this paper, leading to novel negative results about stability and validity of CV confidence intervals even in very regular settings.

Luo & Barber (2024) study the role of algorithmic stability in model comparison, but while our work focuses on drawing inferences about the test error $R_n$, defined in (6), theirs focuses on the expected test error $\mathbb{E}[R_n]$ and shows that inference concerning $\mathbb{E}[R_n]$ is often difficult even when inference concerning $R_n$ is easy. We note that some recent works (Lei, 2020; Li, 2023; Bates et al., 2024) have studied various other aspects of asymptotic distributional properties of CV, but none present negative results comparable to ours. Again focusing on expected test error $\mathbb{E}[R_n]$, Bates et al. (2024, Sec. 5.2.1) consider the case of low-dimensional linear regression with standard Gaussian features and noise and observe that large sample sizes may be needed for valid CV coverage of single algorithm performance using asymptotically exact intervals. Meanwhile, we demonstrate that, for model comparisons, valid test error coverage can fail to materialize for any sample size, even when it does for single algorithm evaluation.

**Notation** For each $n \in \mathbb{N}$, we define the set $[n] \triangleq \{1, \ldots, n\}$. We denote by $\lambda_{\min}(\mathbf{A})$ the minimum eigenvalue of a matrix $\mathbf{A}$. For deterministic sequences $(f_n)_n$ and $(g_n)_n$, $f_n = o(g_n)$ if $\frac{f_n}{g_n} \to 0$ as $n \to \infty$, and $f_n = O(g_n)$ if $\frac{f_n}{g_n}$ is asymptotically bounded. Following canonical notations, we write $f_n = \omega(g_n)$ to mean that $g_n = o(f_n)$ as $n \to \infty$, we write $f_n = \Omega(g_n)$ to mean that $g_n = O(f_n)$ as $n \to \infty$, and we write $f_n = \Theta(g_n)$ to mean that $f_n = O(g_n)$ and $f_n = \Omega(g_n)$ as $n \to \infty$. Finally, we write $f_n \sim g_n$ to mean that $\frac{f_n}{g_n} \to 1$ as $n \to \infty$.

## 2. Preliminaries

Before presenting our results, we establish some necessary definitions, largely following the notation and nomenclature of Bayle et al. (2020). We will consider a sequence of random data points $(Z_i = (X_i, Y_i))_{i \geq 0}$ taking values in a set $\mathcal{Z} = \mathcal{X} \times \mathcal{Y}$ and a scalar performance measure $h_n(Z_0, \mathbf{Z})$ where $\mathbf{Z}$ is a training set of size $n$. A typical choice for $h_n$ in the regression setting is squared error loss,

$$h_n(Z_0, \mathbf{Z}) = (Y_0 - \hat{f}(X_0; \mathbf{Z}))^2,$$

applied to the predicted response value of a test point $Z_0 = (X_0, Y_0)$, obtained from an algorithm fitting a prediction rule $\hat{f}(\cdot; \mathbf{Z})$ to training data $\mathbf{Z}$. When comparing the performance of two algorithms, we will choose $h_n$ to be the difference between the losses of two prediction rules. When we switch between the settings of single algorithm assessment and comparison of algorithms, we will make the distinction clear by adding a superscript to $h_n$: $h_n^{\mathrm{sing}}$ and $h_n^{\mathrm{diff}}$, respectively. In addition, our asymptotic statements should all be interpreted as taking $n \to \infty$.

At the center of our study is a notion of relative algorithmic stability based on the loss stability of Kumar et al. (2013).

**Definition 2.1** (Relative loss stability). *For $n > 0$, let $Z_0$ and $Z_1', Z_1, \ldots, Z_n$ be i.i.d. data points with $\mathbf{Z} = (Z_1, \ldots, Z_n)$ and $\mathbf{Z}' = (Z_1', Z_2, \ldots, Z_n)$. For any function $h_n : \mathcal{Z} \times \mathcal{Z}^n \to \mathbb{R}$ that is invariant to the order of the $n$ elements of its second argument, the* loss stability *(Kumar et al., 2013) is defined as*

$$\gamma(h_n) \triangleq \mathbb{E}[(h_n(Z_0, \mathbf{Z}) - \mathbb{E}[h_n(Z_0, \mathbf{Z}) \mid \mathbf{Z}] \\ - (h_n(Z_0, \mathbf{Z}') - \mathbb{E}[h_n(Z_0, \mathbf{Z}') \mid \mathbf{Z}']))^2].$$

*We also define $\sigma^2(h_n) \triangleq \mathrm{Var}(\mathbb{E}[h_n(Z_0, \mathbf{Z}) \mid Z_0])$. Finally, we define the* relative loss stability *as*

$$r(h_n) \triangleq \frac{n \cdot \gamma(h_n)}{\sigma^2(h_n)}. \tag{1}$$

We introduced these quantities for a function $h_n$, but we will generically refer to the loss stability and the relative loss stability of an algorithm or a comparison of algorithms when $h_n$ is clear from context. Note that we include the factor of $n$ in the numerator of (1) because it sets the baseline rate of $r(h_n)$ to constant order: we will say that an algorithm

or a comparison of algorithms satisfies the *relative loss stability condition* if $r(h_n) = o(1)$, which is equivalent to a key sufficient condition for the central limit theorem and consistent variance estimation for CV proved in Bayle et al. (2020); see Section 4 for more on the connection between relative stability and CV.

We will illustrate the importance of relative stability for CV by studying the stability of two widely-used regression algorithms in a simple learning setting. Throughout, we will consider i.i.d. data points $Z_i = (X_i, Y_i)$ from the (intercept-free) linear model

$$Y_i = X_i^\top \beta^\star + \varepsilon_i, \qquad (2)$$
$$X_i \sim \mathcal{N}(0, \mathbf{I}), \quad \varepsilon_i \sim \mathcal{N}(0, \tau^2), \quad \varepsilon_i \perp\!\!\!\perp X_i,$$

parametrized by the unknown vector $\beta^\star \in \mathbb{R}^p$ with $p \leq n$ and the noise level $\tau > 0$. Here, $\mathbf{Y} = (Y_1, \ldots, Y_n) \in \mathbb{R}^n$ is the vector of response variables or targets, $\mathbf{X} = (X_1, \ldots, X_n)^\top \in \mathbb{R}^{n \times p}$ is the matrix of regressors or features, and $\varepsilon = (\varepsilon_1, \ldots, \varepsilon_n) \in \mathbb{R}^n$ is the noise vector. Note that $\mathbf{X}^\top \mathbf{X}$ is almost surely invertible since $p \leq n$ (Eaton, 2007, Prop. 8.2).

When assessing a single linear prediction rule, we will use the squared error loss

$$h_n^{\mathrm{sing}}(Z_0, \mathbf{Z}) \triangleq (Y_0 - X_0^\top \hat{\beta})^2,$$

where the estimated parameter vector $\hat{\beta}$ is learned from the training set $\mathbf{Z} = (Z_1, \ldots, Z_n)$. When comparing two prediction rules, we will consider the loss difference

$$h_n^{\mathrm{diff}}(Z_0, \mathbf{Z}) \triangleq (Y_0 - X_0^\top \hat{\beta}^{(1)})^2 - (Y_0 - X_0^\top \hat{\beta}^{(2)})^2 \quad (3)$$

for $\hat{\beta}^{(1)}$ and $\hat{\beta}^{(2)}$ both learned on the training set $\mathbf{Z}$.

A classical way to estimate $\beta^\star$ is the ordinary least squares (OLS) estimator defined as

$$\hat{\beta}_{\mathrm{OLS}} \triangleq (\mathbf{X}^\top \mathbf{X})^{-1} \mathbf{X}^\top \mathbf{Y}.$$

To simplify notation, we leave the dependence of $\hat{\beta}_{\mathrm{OLS}}$ on the sample size $n$ implicit. When the parameter vector $\beta^\star$ is expected to exhibit some level of sparsity, that is to have some number of zero entries, a popular alternative estimator is the Lasso (Tibshirani, 1996),

$$\hat{\beta}_{\lambda_n}^{\mathrm{LASSO}} \in \operatorname{argmin}_{\beta \in \mathbb{R}^p} \frac{1}{2n} \|\mathbf{Y} - \mathbf{X}\beta\|_2^2 + \frac{\lambda_n}{n} \|\beta\|_1, \quad (4)$$

which uses a penalty parameter $\lambda_n$ to determine the level of sparsity in the learned parameter vector. As a convenient stepping stone for our Lasso analysis, we also study its close cousin, soft-thresholding.

**Definition 2.2** (Soft-thresholding (ST)). *We define the* ST$(\lambda_n)$ *estimator* $\hat{\beta}_{\lambda_n}$ *elementwise as*

$$\hat{\beta}_{\lambda_n, i} \triangleq \operatorname{sign}(\hat{\beta}_{\mathrm{OLS}, i})(|\hat{\beta}_{\mathrm{OLS}, i}| - \tfrac{\lambda_n}{n})_+, \ \ i = 1, \ldots, p. \ (5)$$

Our first result shows that the ST and Lasso estimators are close whenever $\mathbf{X}^\top \mathbf{X}/n$ is close to the identity. In the sequel, we will use this proximity to deduce Lasso stability and instability results from ST stability and instability.

**Lemma 2.3** (Lasso-ST proximity). *For any* $\lambda_n > 0$, $\mathbf{Y} \in \mathbb{R}^n$, *and* $\mathbf{X} \in \mathbb{R}^{n \times p}$, *the* ST$(\lambda_n)$ *estimator* $\hat{\beta}_{\lambda_n}$ (5) *and* Lasso$(\lambda_n)$ *estimator* $\hat{\beta}_{\lambda_n}^{\mathrm{LASSO}}$ (4) *satisfy*

$$\|\hat{\beta}_{\lambda_n} - \hat{\beta}_{\lambda_n}^{\mathrm{LASSO}}\|_2 \leq \frac{\|\mathbf{X}^\top \mathbf{X}/n - \mathbf{I}\|_{\mathrm{op}}}{\mu_n} \frac{\lambda_n \sqrt{p}}{n},$$

*where* $\mu_n \triangleq \lambda_{\min}(\mathbf{X}^\top \mathbf{X}/n)$. *Moreover, if* $\mathbf{X} = (X_1, \ldots, X_n)^\top$ *with* $X_i \overset{i.i.d.}{\sim} \mathcal{N}(0, \mathbf{I})$, *then* $\mathbb{E}\left[\frac{\|\mathbf{X}^\top \mathbf{X}/n - \mathbf{I}\|_{\mathrm{op}}^q}{\mu_n^q}\right] = O(\frac{1}{n^{q/2}})$ *for any* $q \in \mathbb{N}$.

The proof of Lemma 2.3 in Appendix C follows from viewing $\hat{\beta}_{\lambda_n}^{\mathrm{LASSO}}$ and $\hat{\beta}_{\lambda_n}$ as the optimizers of closely related objective functions and using the optimizer comparison lemma of Wilson et al. (2020, Lem. 1) to deduce their proximity.

## 3. Main Results

We now state the main results of this work. Our primary theoretical result is that even a simple learning algorithm (ST) in a simple, well-behaved learning setting can fail to generate relatively stable comparisons.

**Theorem 3.1** (Relative instability of ST comparisons). *Assume the linear model* (2) *with* $\|\beta^\star\|_0 < p$. *For* $\lambda_n = O(\sqrt{n})$, $\lambda_n = \omega(1)$, *and* $\delta_n = \Theta(1)$, *consider the algorithm comparison of* ST$(\lambda_n)$ *with* ST$(\lambda_n + \delta_n)$, *i.e.,* $h_n^{\mathrm{diff}}$ *is defined via* (3) *with* $\hat{\beta}^{(1)} = \hat{\beta}_{\lambda_n}$ *and* $\hat{\beta}^{(2)} = \hat{\beta}_{\lambda_n + \delta_n}$. *Then,*

$$\tfrac{n^2}{\delta_n^2} \sigma^2(h_n^{\mathrm{diff}}) \to 4\tau^2 \|\beta^\star\|_0 \quad \text{and} \quad \gamma(h_n^{\mathrm{diff}}) = \Omega(\tfrac{1}{n^2 \sqrt{n}}).$$

*Thus this ST comparison is relatively unstable with*

$$r(h_n^{\mathrm{diff}}) = \Omega(\sqrt{n}) \neq o(1).$$

The proof of Theorem 3.1 can be found in Appendix D. Notably, since this main result is a *lower* bound, the stringent assumptions like linearity, sparsity, covariate independence, and Gaussianity serve to strengthen the result, indicating that failure occurs even in this best-case scenario.

Perhaps surprisingly, under the same conditions, we have relative stability for single algorithm assessment.

**Theorem 3.2** (Relative stability of ST). *Assume the linear model* (2). *For the single algorithm assessment of* ST$(\lambda_n)$, *define the loss* $h_n^{\mathrm{sing}}(Z_0, \mathbf{Z}) = (Y_0 - X_0^\top \hat{\beta}_{\lambda_n})^2$. *If* $\lambda_n = o(n)$, *then*

$$\sigma^2(h_n^{\mathrm{sing}}) \to 2\tau^4 \quad \text{and} \quad \gamma(h_n^{\mathrm{sing}}) \sim \tfrac{C}{n^2}$$

*for a constant* $C > 0$ *defined explicitly in* (8), *so ST is relatively stable with*

$$r(h_n^{\mathrm{sing}}) \sim \tfrac{C}{2\tau^4} \cdot \tfrac{1}{n} = o(1).$$

The proof of Theorem 3.2 can be found in Appendix G. This secondary result is also significant as it shows that a CV user can be easily duped into thinking that confidence intervals that rely on stability will yield valid inference in the algorithm comparison setting simply because an algorithm is stable when considered in isolation.

We can think of Theorem 3.1 as a stylized version of a setting where one wants to compare two similar machine learning algorithms, such as when the two only differ by a tuning parameter. Note that $\lambda_n = O(\sqrt{n})$ implies $\lambda_n = o(n)$, and that $\Theta(1)$ is also $o(n)$ as it is asymptotically lower- and upper-bounded by a constant, which means that $\lambda_n + \delta_n = o(n)$ under the conditions of Theorem 3.1 and thus both ST($\lambda_n$) and ST($\lambda_n + \delta_n$) individually satisfy the relative loss stability condition thanks to Theorem 3.2. So, taken together, Theorems 3.1 and 3.2 show that even if two learning algorithms are individually well-behaved, their comparison may not be, even when the data comes from a very regular distribution.

We now discuss the penalty parameter regimes appearing in Theorems 3.1 and 3.2. In our Section 5 simulations with features and targets sampled in the same conditions as the theorems, we observe that the values selected for $\lambda_n$ via CV are concentrated around a constant times $\sqrt{n}$. It therefore makes sense to compare two versions of ST with penalization of order $\sqrt{n}$ in Theorem 3.1, and we do so by setting the base level of penalization to $\lambda_n$ of order $\sqrt{n}$ and parameterizing the difference in penalization of the ST algorithms by $\delta_n$ of order 1. Note that both $\lambda_n$ and $\delta_n$ are assumed deterministic in the theorems, but we will also present simulations with stochastic $\lambda_n$ selected via inner CV in Section 5. Under some regularity conditions on the features, Knight & Fu (2000, Thm. 1) proved that choosing $\lambda_n = o(n)$ ensures weak consistency of the Lasso estimator for $\beta^\star$, i.e., it converges in probability to $\beta^\star$, and it is therefore natural that the regimes we study are always within this weak consistency regime. As for the $\sqrt{n}$ order of the penalization specific to our primary result, it has been shown to be a regime of interest for variable selection consistency (Wainwright, 2009; 2019).

The powerful Lasso-ST proximity bound of Lemma 2.3 allows us to translate Theorems 3.1 and 3.2 into identical results for the popular Lasso algorithm, showing that our conclusions about ST are by no means specific to that method.

**Theorem 3.3** (Relative instability of Lasso comparisons). *Assume the linear model* (2) *with* $\|\beta^\star\|_0 < p$. *For* $\lambda_n = O(\sqrt{n})$, $\lambda_n = \omega(1)$, *and* $\delta_n = \Theta(1)$, *consider the algorithm comparison of* Lasso($\lambda_n$) *with* Lasso($\lambda_n + \delta_n$), *i.e.,* $\tilde{h}_n^{\mathrm{diff}}$ *is defined via* (3) *with* $\hat{\beta}^{(1)} = \hat{\beta}_{\lambda_n}^{\mathrm{LASSO}}$ *and* $\hat{\beta}^{(2)} =$

$\hat{\beta}_{\lambda_n + \delta_n}^{\mathrm{LASSO}}$. *Then*

$$\sigma^2(\tilde{h}_n^{\mathrm{diff}}) = O(\tfrac{1}{n^2}) \quad \text{and} \quad \gamma(\tilde{h}_n^{\mathrm{diff}}) = \Omega(\tfrac{1}{n^2 \sqrt{n}}).$$

*Thus this Lasso comparison is relatively unstable with*

$$r(\tilde{h}_n^{\mathrm{diff}}) = \Omega(\sqrt{n}) \neq o(1).$$

Our proof in Appendix K combines the ST instability bounds of Theorem 3.1 with the powerful Lasso-ST proximity bound of Lemma 2.3.

As with ST, Lasso comparison instability occurs even though the Lasso algorithm itself is relatively stable.

**Theorem 3.4** (Relative stability of the Lasso). *Assume the linear model* (2). *For the single algorithm assessment of* Lasso($\lambda_n$), *define the loss* $\tilde{h}_n^{\mathrm{sing}}(Z_0, \mathbf{Z}) = (Y_0 - X_0^\top \hat{\beta}_{\lambda_n}^{\mathrm{LASSO}})^2$. *If* $\lambda_n = o(n)$, *then*

$$\sigma^2(\tilde{h}_n^{\mathrm{sing}}) = \Omega(1) \quad \text{and} \quad \gamma(\tilde{h}_n^{\mathrm{sing}}) = o(\tfrac{1}{n}),$$

*so the Lasso algorithm is relatively stable with*

$$r(\tilde{h}_n^{\mathrm{sing}}) = o(1).$$

Our Lasso stability proof in Appendix L may be of independent interest as the Lasso is known to be unstable under the more stringent notion of *uniform stability* (Xu et al., 2012).

## 4. Importance of Relative Stability for Cross-validation

Let us now consider the implications of our results for CV. When discussing CV, we will use $n$ to refer to the size of each CV training set (rather than the size of the full data sample as in Bayle et al. (2020)). We also let $k$ denote the number of CV folds, note that $k$ can depend on $n$ (for example, leave-one-out CV corresponds to $k = n + 1$), and assume that $k - 1$ evenly divides $n$. In this notation, the full data sample size equals $\frac{nk}{k-1}$.

Consider $k$ vectors of integers, $\{B'_j\}_{j=1}^k$, each of length $\frac{n}{k-1}$, with elements that partition the indices $[\frac{nk}{k-1}]$. For each $B'_j$, define $B_j$ as a vector of the $n$ indices in $[\frac{nk}{k-1}]$ that are not in $B'_j$, so that we can consider each $(B_j, B'_j)$ as a *train-validation split*. For $B$ a vector of indices in $[\frac{nk}{k-1}]$, we denote by $Z_B$ the subvector of $(Z_1, \ldots, Z_{\frac{nk}{k-1}})$ corresponding to the entries of $B$. Then for a scalar performance measure $h_n$, we define the *k-fold cross-validation error*

$$\hat{R}_n \triangleq \tfrac{k-1}{nk} \sum_{j=1}^k \sum_{i \in B'_j} h_n(Z_i, Z_{B_j})$$

and our inferential target, the *k-fold test error*

$$R_n \triangleq \tfrac{k-1}{nk} \sum_{j=1}^k \sum_{i \in B'_j} \mathbb{E}[h_n(Z_i, Z_{B_j}) \mid Z_{B_j}], \quad (6)$$

where the conditioning in $\mathbb{E}[h_n(Z_i, Z_{B_j}) \mid Z_{B_j}]$ is on the data points from the $j$-th validation set $Z_{B_j}$ and thus the expectation is taken over only the test point $Z_i$, as the function $h_n$ is treated as non-random.

In our notation, Bayle et al. (2020) used the stability condition $\gamma(h_n) = o(\frac{\sigma^2(h_n)}{n})$, equivalent to $r(h_n) = o(1)$, to establish the relative error consistency (i.e., $\frac{\hat{\sigma}_n(h_n)}{\sigma(h_n)} \to 1$) of the within-fold variance estimator $\hat{\sigma}_n^2(h_n)$ introduced in Austern & Zhou (2020, Prop. 1) and to prove the central limit theorem (CLT)

$$\frac{\sqrt{\frac{nk}{k-1}}}{\sigma(h_n)}(\hat{R}_n - R_n) \xrightarrow{d} \mathcal{N}(0, 1). \tag{7}$$

Together, these results enable the construction of asymptotically exact confidence intervals for $R_n$.

When assessing a single algorithm, unless we are in a fully noiseless setting, we might expect $\sigma^2(h_n^{\mathrm{sing}})$ to be of constant order in general. In this case, relative loss stability is implied by absolute loss stability of the form $\gamma(h_n^{\mathrm{sing}}) = o(1/n)$. For instance, we state here in Lemma 4.1 and prove in Appendix H that in the linear model with noise, $\sigma^2(h_n^{\mathrm{sing}})$ converges to a positive constant for any linear predictor satisfying a certain consistency condition.

**Lemma 4.1** (Convergence of $\sigma^2(h_n^{\mathrm{sing}})$ and $\sigma^2(h_n^{\mathrm{diff}})$). *Consider a generalization of the linear model* (2) *with* $(X_i)_{i \geq 0}$ *drawn i.i.d. from a distribution with mean 0 and identity covariance. If a linear predictor $\hat{\beta}_n$ satisfies $\mathbb{E}[\hat{\beta}_n] \to \beta^\star$ and $\mathbb{E}[\hat{\beta}_n \hat{\beta}_n^\top] \to \beta^\star \beta^{\star\top}$, then $\sigma^2(h_n^{\mathrm{sing}}) \to 2\tau^4$, where $\tau^2$ is the noise variance. For two linear predictors, if we have $\mathbb{E}[\hat{\beta}_n^{(1)} - \hat{\beta}_n^{(2)}] \to 0$ and $\mathbb{E}[\hat{\beta}_n^{(1)} \hat{\beta}_n^{(1)\top} - \hat{\beta}_n^{(2)} \hat{\beta}_n^{(2)\top}] \to 0$, then $\sigma^2(h_n^{\mathrm{diff}}) \to 0$.*

However, Lemma 4.1 also establishes that when comparing two consistent algorithms, $\sigma^2(h_n^{\mathrm{diff}})$ often goes to 0 as the performance of the learned algorithms become increasingly similar. In this setting, absolute stability alone is insufficient. In fact, in Theorem 3.1 it turns out that $\gamma(h_n^{\mathrm{diff}}) = O(1/n^2)$ (see Appendix F), so the ST comparison is loss stable in the absolute sense. However, the *relative* loss stability condition does not hold because it properly accounts for the fact that $\sigma^2(h_n^{\mathrm{diff}})$ also goes to zero at a $1/n^2$ rate. When relative loss stability is violated, the CLT and consistency guarantees of Bayle et al. (2020) need no longer hold, and confidence intervals based on (7) can lead to asymptotically invalid inferences. Unfortunately, this worst case does indeed occur in practice as we demonstrate next.

## 5. Numerical Experiments

To numerically confirm the dire consequences of relative instability for CV, we carry simulation results in the setting of Theorems 3.1, 3.2, 3.3 and 3.4. We sampled the features, the independent noise terms, and the target variables from the linear model (2) with parameter vector $\beta^\star = (3, 1, -5, 3, 0, 0, 0, 0, 0, 0)$ of dimension 10 and noise level $\tau = 10$. We fix $k = 10$. To satisfy the assumptions of Theorems 3.1 and 3.2, we choose $\lambda_n = \sqrt{n}$ for the base level of penalization, and, when comparing algorithms, we set $\delta_n = 1$ for the difference in the penalization parameters. To explore the asymptotic regime in our simulations, we work with $n$ ranging from 90 to 90,000 (so the total sample size $\frac{nk}{k-1}$ ranged from $10^2$ to $10^5$). We used Monte Carlo estimation to compute both $\sigma^2(h_n)$ and $\gamma(h_n)$, leveraging Lemmas M.1 and M.2 proved in Appendix M. We provide Python code replicating all experiments at `https://github.com/alexandre-bayle/ricv` and additional details about the experiments in Appendix M.

We present two types of plots. The first type displays the rates for $\sigma^2(h_n)$, $\gamma(h_n)$, and $r(h_n)$ on the log–log scale by plotting their empirical values for increasing $n$ with dots and plotting lines for the corresponding rates predicted by our theory. We display the values with a $\pm 2$ standard error confidence band with details on how to obtain it for $r(h_n)$ in Appendix M. Thanks to the large number of Monte Carlo replications used, the error bars are very small and thus are not visible. Note that we use the $x$-axis labels $n/900$ to provide a better scale for visualization. $n/900$ goes up to $10^2$, which is consistent with $n$ going up to 90,000. For the second type of plot, using kernel density estimation (KDE), we plot the probability density function across sample sizes of both $\frac{\sqrt{\frac{nk}{k-1}}}{\sigma(h_n)}(\hat{R}_n - R_n)$ and $\frac{\sqrt{\frac{nk}{k-1}}}{\hat{\sigma}_n(h_n)}(\hat{R}_n - R_n)$, where $\hat{\sigma}_n^2(h_n)$ is the within-fold variance estimator, proved to be consistent for $\sigma^2(h_n)$ under the relative loss stability condition in Bayle et al. (2020, Thm. 4). We expect convergence in distribution to $\mathcal{N}(0, 1)$ under the relative loss stability condition thanks to the combination of results of Bayle et al. (2020, Thms. 1, 2, and 4). We therefore shade the area below the curve of the probability density function of $\mathcal{N}(0, 1)$ to make it clear whether the probability density function curves match or not. From its definition (6), note that $R_n$ is straightforward to compute in the simulations thanks to Lemma M.2.

The simulation results for ST are presented in Figure 2. For the single algorithm assessment of ST, the rates of $\sigma^2(h_n^{\mathrm{sing}})$, $\gamma(h_n^{\mathrm{sing}})$, and $r(h_n^{\mathrm{sing}})$ are constant order, $1/n^2$ order, and $1/n$ order, respectively, as stated in Theorem 3.2, and for the algorithm comparison of ST, when $\delta_n = 1$, we have the expected $1/n^2$ rate for $\sigma^2(h_n^{\mathrm{diff}})$ and we actually observe that $\gamma(h_n^{\mathrm{diff}})$ and $r(h_n^{\mathrm{diff}})$ seem to be scaling as $1/(n^2\sqrt{n})$ and $\sqrt{n}$, respectively, even though Theorem 3.1 only established them being $\Omega$ of these rates. As we can see for both choices of the dividing standard deviation in the KDE plots of Figure 2, the asymptotic distribution seems to be Gaussian, but the asymptotic variance does not go to 1 when

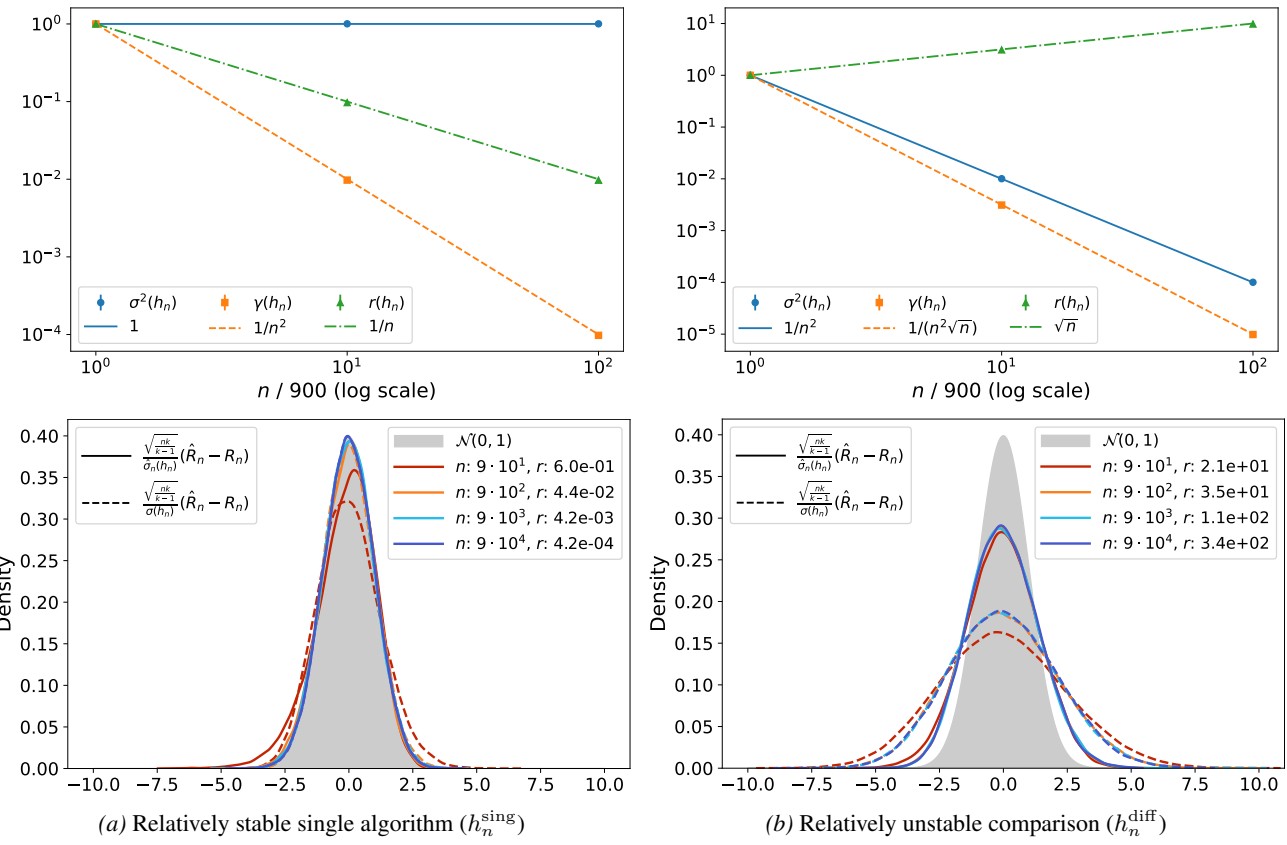

*(a) Relatively stable single algorithm ($h_n^{\mathrm{sing}}$)*    *(b) Relatively unstable comparison ($h_n^{\mathrm{diff}}$)*

*Figure 2.* ST with $\lambda_n = \sqrt{n}$ when $\beta^\star = (3, 1, -5, 3, 0, 0, 0, 0, 0, 0)$. **Top:** $\sigma^2(h_n)$, $\gamma(h_n)$ and $r(h_n)$ all normalized by their values at $n = 900$. **Bottom:** (best viewed in color) KDE plots for $\frac{\sqrt{\frac{nk}{k-1}}}{\hat{\sigma}_n(h_n)}(\hat{R}_n - R_n)$ (solid curves) and $\frac{\sqrt{\frac{nk}{k-1}}}{\sigma(h_n)}(\hat{R}_n - R_n)$ (dashed curves).

the relative loss stability condition does not hold, that is to say in the comparison setting. The variance estimator $\hat{\sigma}_n^2(h_n^{\mathrm{diff}})$ has been proved to be a consistent estimator of the targeted variance of $\sqrt{\frac{nk}{k-1}}(\hat{R}_n - R_n)$ under the loss stability condition in Bayle et al. (2020). In the setting we explored, the condition does not hold for the comparison, and we observed empirically that $\hat{\sigma}_n^2(h_n^{\mathrm{diff}})$ underestimates the targeted variance of $\sqrt{\frac{nk}{k-1}}(\hat{R}_n - R_n)$, and overestimates $\sigma^2(h_n^{\mathrm{diff}})$. While the intervals proposed in Bayle et al. (2020, Eq. 4.1) are valid when the loss stability condition holds, they will not be wide enough when $\hat{\sigma}_n^2(h_n^{\mathrm{diff}})$ underestimates the targeted variance of $\sqrt{\frac{nk}{k-1}}(\hat{R}_n - R_n)$, leading to undercoverage and hence asymptotic invalidity.

Next, in light of our analogous theoretical results for the Lasso, we provide analogous simulations for the Lasso as well, though we make them even more realistic by choosing $\lambda_n$ via inner cross-validation. In particular, we ran simulations for the Lasso with $\lambda_n$ selected via an inner CV (see Appendix M) for each of the $k$ iterations of the CV run, still with constant order $\delta_n = 1$ for the comparison. As mentioned in Section 3, we actually observed in simulations

that the values selected for $\lambda_n$ are concentrated around a constant times $\sqrt{n}$. The results for this setting are displayed in Figure 3 and confirm that the same conclusions hold empirically for the cross-validated Lasso as for ST.

We note that the dichotomy exhibited by ST and Lasso is not universal: there are instances when an algorithm satisfies the relative loss stability condition both in its individual form and in the comparison setting. One example of this is ridge regression and we present the corresponding simulations in Figure 4 in Appendix N. Bousquet & Elisseeff (2002) proved that ridge regression with penalty parameter $\lambda_n = \sqrt{n}$ and bounded targets has $O(1/n)$ uniform stability. This means it has $O(1/n^2)$ loss stability by Kale et al. (2011, Lem. 1) and Kumar et al. (2013, Lem. 2). In the simulations, we see that for individual ridge with isotropic features and no boundedness assumption, loss stability scales as $1/n^2$ and the relative loss stability condition therefore holds since $\sigma^2(h_n^{\mathrm{sing}})$ is of constant order. Meanwhile, in the ridge comparison setting, loss stability scales as $1/n^4$, which, when compared to the observed $1/n^2$ rate of $\sigma^2(h_n^{\mathrm{diff}})$, means the relative loss stability condition also holds for ridge comparisons.

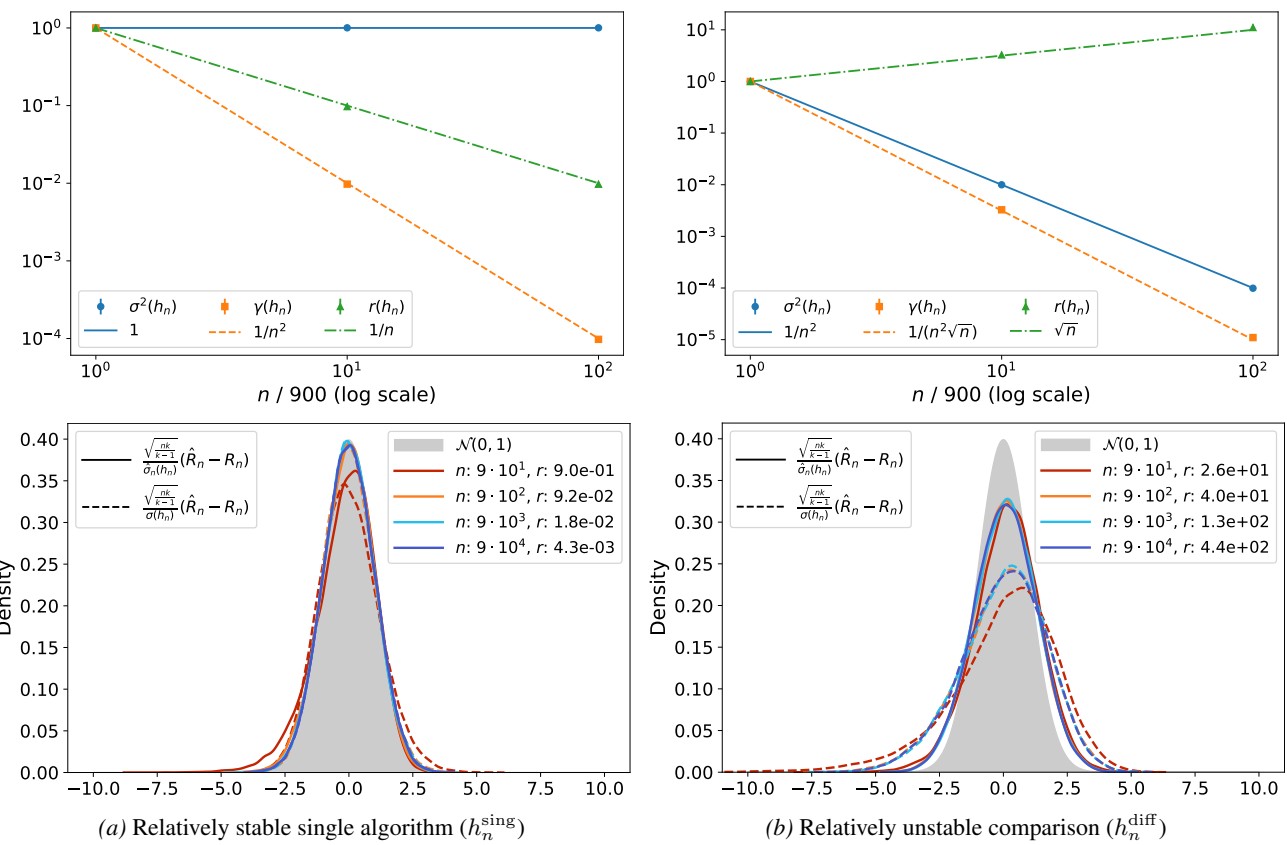

*Figure 3.* Lasso with cross-validated $\lambda_n$ when $\beta^\star = (3, 1, -5, 3, 0, 0, 0, 0, 0, 0)$. **Top:** $\sigma^2(h_n)$, $\gamma(h_n)$ and $r(h_n)$ all normalized by their values at $n = 900$. **Bottom:** (best viewed in color) KDE plots for $\frac{\sqrt{\frac{nk}{k-1}}}{\hat{\sigma}_n(h_n)}(\hat{R}_n - R_n)$ (solid curves) and $\frac{\sqrt{\frac{nk}{k-1}}}{\sigma(h_n)}(\hat{R}_n - R_n)$ (dashed curves).

As a matter of fact, when $\beta^\star$ has no zero coefficients, the ST estimator can also be an example of an algorithm which satisfies the relative loss stability condition in both its individual form and in the comparison setting. The theory sheds light on the importance of the zero coefficients in the true parameter vector. When $\beta^\star$ has no zero coefficients, i.e. $\|\beta^\star\|_0 = p$, ST actually becomes stable for the algorithm comparison setting. The results of the simulations for this setting, with the choice $\beta^\star = (3, 1, -5, 3, 4, -3, 10, 8, 5, 2)$, are presented in Figure 5 in Appendix N and show how the convergence rate of $\gamma(h_n^{\mathrm{diff}})$ changes compared to the $\|\beta^\star\|_0 < p$ setting. It now scales as $1/n^4$, which means that ST satisfies the relative loss stability condition $r(h_n^{\mathrm{diff}}) = o(1)$ in the comparison setting, since $\frac{n^2}{\delta_n^2}\sigma^2(h_n^{\mathrm{diff}})$ still goes to $4\tau^2\|\beta^\star\|_0$ when $\|\beta^\star\|_0 = p$. Nonetheless, we reiterate that even a single zero coefficient in $\beta^\star$ leads to instability for ST and, more generally, for the Lasso in the comparison setting.

## 6. Conclusions and Future Work

Cross-validation is a powerful tool, but given its widespread use for comparing and selecting models, scrutiny of its

statistical properties is critical for safe model deployment. This work highlights the importance of relative stability for CV and the challenges posed by relative instability for model comparison. In particular, we proved that even simple, absolutely-stable learning algorithms can generate relatively unstable comparisons. In practice, this led to invalid and highly misleading confidence intervals for the test error difference with $\sigma^2(h_n^{\mathrm{diff}})$ being well below the targeted variance of $\sqrt{\frac{nk}{k-1}}(\hat{R}_n - R_n)$. Since CV is often used to conduct formal hypothesis tests for an improvement in test error between two learning algorithms (Dieterich, 1998; Lim et al., 2000; Nadeau & Bengio, 2003; Bouckaert & Frank, 2004; Demšar, 2006; Bayle et al., 2020), our work shows that such tests can be misleading even for simple, absolutely stable algorithms and that method developers and consumers should first verify the relative stability of a comparison before applying them.

This paper uses ST and the Lasso to illustrate the dichotomy between algorithm evaluation and comparison when using CV for uncertainty quantification. While it is true that we expect this dichotomy to extend to other ML algorithms as well, we do not attempt to make any claims concerning

other ML algorithms in this work. Importantly, we did not aim to show that the CV central limit theorem (7) is always a poor choice for algorithm comparison. Indeed, Section 5 presented examples (ST with fully dense $\beta^*$ and ridge regression) in which relative comparisons *are* stable. That said, what we have shown is that even a simple ML algorithm, in the linear model setting, applied to very well-behaved data, can fail to satisfy relative stability in the comparison setting, which we hope is enough to at least convince users of CV that they should not expect by default that relative stability holds when comparing two algorithms (even if they are individually stable). This we believe is an important and practical realization that was previously unknown. Our main goal is to increase awareness of the pitfalls of CV, highlighting how simple it is for it to be misleading, especially if not studied through the proper lens of relative stability.

However, this work is not without its limitations. First, our analyses are fairly specific to our particular data distribution and the Lasso and ST models. Establishing broad, easily verified conditions under which an algorithm comparison is or is not relatively stable is an important direction for future work. Second, while we prove the relative instability of ST and Lasso comparisons and demonstrate the invalidity of their CV confidence intervals, we leave open the question of whether relative instability always implies CV invalidity.

The focus of this work is on exposing a surprising failure mode of the commonly used CV procedure and not on identifying the best inference procedure for test error. This is why our experiments, designed principally to corroborate our theory, focused on CV intervals alone. While we have shown that the CV central limit theorem (7) and hence the CV confidence interval construction of Bayle et al. (2020) can break down in the presence of relatively unstable comparisons, our next result presents an alternative (possibly conservative) CI construction that yields asymptotic validity even if the comparison is not relatively stable. Specifically, it yields validity whenever each algorithm is individually relatively stable or, more generally, whenever one can construct a valid interval separately for each algorithm's test loss.

**Proposition 6.1** (Comparison coverage from single algorithm coverage). *Let $\hat{R}_n^{(1)}, R_n^{(1)}$ be the cross-validation error and test error of algorithm $\mathcal{A}_1$, and $\hat{R}_n^{(2)}, R_n^{(2)}$ those of algorithm $\mathcal{A}_2$. To compare $\mathcal{A}_1$ and $\mathcal{A}_2$, if $[L_n^{(1)}, U_n^{(1)}]$ and $[L_n^{(2)}, U_n^{(2)}]$ are asymptotic $(1 - \alpha/2)$-coverage confidence intervals for $R_n^{(1)}$ and $R_n^{(2)}$, respectively, then*

$$[L_n^{(1)} - U_n^{(2)}, U_n^{(1)} - L_n^{(2)}]$$

*will asymptotically cover $R_n^{(1)} - R_n^{(2)}$ with probability at least $1 - \alpha$.*

**Proof**

$$\liminf_{n\to\infty} \mathbb{P}(R_n^{(1)} - R_n^{(2)} \in [L_n^{(1)} - U_n^{(2)}, U_n^{(1)} - L_n^{(2)}])$$
$$\geq 1 - \limsup_{n\to\infty} \mathbb{P}(R_n^{(1)} \notin [L_n^{(1)}, U_n^{(1)}]$$
$$\text{or } R_n^{(2)} \notin [L_n^{(2)}, U_n^{(2)}])$$
$$\geq 1 - \limsup_{n\to\infty} \mathbb{P}(R_n^{(1)} \notin [L_n^{(1)}, U_n^{(1)}])$$
$$- \limsup_{n\to\infty} \mathbb{P}(R_n^{(2)} \notin [L_n^{(2)}, U_n^{(2)}])$$
$$\geq 1 - \alpha/2 - \alpha/2 = 1 - \alpha.$$

□

The Proposition 6.1 approach ensures valid asymptotic coverage under individual algorithm stability without requiring any additional stability assumption on the comparison. However, the interval could also be significantly wider than the interval derived from Bayle et al. (2020), due to strong positive correlations between $\hat{R}_n^{(1)}$ and $\hat{R}_n^{(2)}$ ignored in the construction of Proposition 6.1. Figure 6 in Appendix N illustrates this behavior in the setting used for Figure 2. An open question is whether one can derive tighter confidence intervals for algorithm comparisons when it is only known that each algorithm is individually stable.

## Impact Statement

By highlighting a surprising failure mode of a commonly used procedure for quantifying confidence in the difference between learning algorithms, this paper's potential broader impact is to reduce the overinterpretation of small empirical CV differences between two learning algorithms, helping to more rigorously distinguish legitimate improvements from inconsequential changes.

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

# Appendix Contents

## A. Additional Notation

Let $\xrightarrow{\text{a.s.}}$ denote almost sure convergence. Let $\mathbb{1}[A]$ denote the indicator function of a subset $A$. We will denote by $\Phi$ the cumulative distribution function of the standard Normal and by $\varphi$ its probability density function. We define the sign function as $\mathrm{sign}(x) = \frac{x}{|x|}\mathbb{1}[x \neq 0]$ and the positive part as $x_+ = \max(x, 0)$. We write $\mathbf{M} \sim W_p^{-1}(\Sigma, n)$ to indicate $\mathbf{M}$ follows the inverse-Wishart distribution with $n$ degrees of freedom and scale matrix $\Sigma \in \mathbb{R}^{p \times p}$.

## B. Experimental Details for Figure 1

The experimental setup for Figure 1 is very similar to the one described in the first paragraph of Section 5. We consider the Lasso estimator here, with $\lambda_n = \sqrt{n}$ for the base level of penalization, and when comparing algorithms, $\delta_n = 1$ for the difference in the penalization parameters, where $\beta^\star = (3, 1, -5, 3, 0, 0, 0, 0, 0, 0)$. For the largest sample size under consideration, we are plotting the actual coverage probability of the confidence interval $\hat{R}_n \pm q_{1-\alpha/2}\,\sigma(h_n)/\sqrt{\frac{nk}{k-1}}$, over the full range $\alpha \in [0, 1]$, where $q_{1-\alpha/2}$ is the $(1 - \alpha/2)$-quantile of the standard normal distribution, built from the CV central

limit theorem of Bayle et al. (2020) using the true variance $\sigma^2(h_n)$, against the target coverage, in the single algorithm setting and the comparison setting.

## C. Proof of Lemma 2.3: Lasso-ST proximity

Define the objective functions

$$f_1(\beta) \triangleq \tfrac{1}{2}\|(\mathbf{X}^\top\mathbf{X})^{-1}\mathbf{X}^\top\mathbf{Y} - \beta\|_2^2 + \tfrac{\lambda_n}{n}\|\beta\|_1 \quad \text{and} \quad f_2(\beta) \triangleq \tfrac{1}{2n}\|\mathbf{Y} - \mathbf{X}\beta\|_2^2 + \tfrac{\lambda_n}{n}\|\beta\|_1$$

so that $\hat{\beta}_{\lambda_n} \in \operatorname{argmin}_{\beta\in\mathbb{R}^p} f_1(\beta)$ and $\hat{\beta}_{\lambda_n}^{\mathrm{LASSO}} \in \operatorname{argmin}_{\beta\in\mathbb{R}^p} f_2(\beta)$. For any $\beta \in \mathbb{R}^p$ we have

$$\begin{aligned}
\nabla f_2(\beta) - \nabla f_1(\beta) &= \tfrac{1}{n}\mathbf{X}^\top\mathbf{X}\beta - \tfrac{1}{n}\mathbf{X}^\top\mathbf{Y} - (\beta - (\mathbf{X}^\top\mathbf{X})^{-1}\mathbf{X}^\top\mathbf{Y}) \\
&= (\mathbf{I} - (\mathbf{X}^\top\mathbf{X}/n)^{-1})(\mathbf{X}^\top\mathbf{X}\beta - \mathbf{X}^\top\mathbf{Y})/n \\
&= (\mathbf{X}^\top\mathbf{X}/n - \mathbf{I})(\beta - (\mathbf{X}^\top\mathbf{X})^{-1}\mathbf{X}^\top\mathbf{Y}) = (\mathbf{X}^\top\mathbf{X}/n - \mathbf{I})(\beta - \hat{\beta}_{\mathrm{OLS}}).
\end{aligned}$$

Moreover, by the definitions of the operator norm and $\hat{\beta}_{\lambda_n}$ (Definition 2.2),

$$\begin{aligned}
\|\nabla f_2(\hat{\beta}_{\lambda_n}) - \nabla f_1(\hat{\beta}_{\lambda_n})\|_2 &= \|(\mathbf{X}^\top\mathbf{X}/n - \mathbf{I})(\hat{\beta}_{\lambda_n} - \hat{\beta}_{\mathrm{OLS}})\|_2 \\
&\leq \|\mathbf{X}^\top\mathbf{X}/n - \mathbf{I}\|_{\mathrm{op}}\|\hat{\beta}_{\lambda_n} - \hat{\beta}_{\mathrm{OLS}}\|_2 \leq \|\mathbf{X}^\top\mathbf{X}/n - I\|_{\mathrm{op}}\sqrt{p}\lambda_n/n.
\end{aligned}$$

Finally, since $f_2$ is $\mu_n$ strongly convex, the optimizer comparison lemma of (Wilson et al., 2020, Lem. 1) implies that $\mu_n\|\hat{\beta}_{\lambda_n} - \hat{\beta}_{\lambda_n}^{\mathrm{LASSO}}\|_2^2 \leq \|\hat{\beta}_{\lambda_n} - \hat{\beta}_{\lambda_n}^{\mathrm{LASSO}}\|_2\|\nabla f_2(\hat{\beta}_{\lambda_n}) - \nabla f_1(\hat{\beta}_{\lambda_n})\|_2$, yielding the first result.

Now fix any $q \in \mathbb{N}$, and suppose $X_i \overset{\mathrm{i.i.d.}}{\sim} \mathcal{N}(0, \mathbf{I})$ and $(n - p + 1)/2 > 2q$. Then $V = (\mathbf{X}^\top\mathbf{X})^{-1}$ has an inverse-Wishart distribution with $n$ degrees of freedom, and each diagonal entry $V_{jj}$ has an inverse-gamma distribution with shape $= \frac{n-p+1}{2}$ and scale $= \frac{1}{2}$ (Bodnar et al., 2016, Cor. 1). Therefore, by Jensen's inequality and the moment formula for an inverse-gamma,

$$\begin{aligned}
\mathbb{E}[1/\mu_n^{2q}] = \mathbb{E}[\lambda_{\max}(nV)^{2q}] &\leq \mathbb{E}[\mathrm{tr}(nV)^{2q}] \leq \mathbb{E}[p^{2q-1}\textstyle\sum_{j=1}^p (nV_{jj})^{2q}] = (np)^{2q}\mathbb{E}[V_{11}^{2q}] \\
&= (\tfrac{np}{2})^{2q}\frac{\Gamma(\frac{n-p+1}{2} - 2q)}{\Gamma(\frac{n-p+1}{2})} \leq (\tfrac{n}{n-p+1-4q})^{2q}p^{2q} = O(1).
\end{aligned}$$

Next let $W = \mathbf{X}^\top\mathbf{X}$ so that each entry $W_{jk} = \sum_{i=1}^n X_{ij}X_{ik}$. Then, we may apply Jensen's inequality, the Marcinkiewicz-Zygmund (Rio, 2009, Thm. 2.1) inequality, Jensen's inequality again, and finally the moment formula for a Gaussian random variable to find that

$$\begin{aligned}
\mathbb{E}[\|\mathbf{X}^\top\mathbf{X}/n - \mathbf{I}\|_{\mathrm{op}}^{2q}] = \mathbb{E}[\|\mathbf{X}^\top\mathbf{X}/n - \mathbf{I}\|_F^{2q}] &\leq \mathbb{E}[\tfrac{p^{2q-2}}{n^{2q}}\textstyle\sum_{j=1}^p\sum_{k=1}^p |W_{jk} - \mathbb{E}[W_{jk}]|^{2q}] \\
&\leq \tfrac{p^{2q-2}(2q-1)^q n^q}{n^{2q}}(p\mathbb{E}[|X_{11}^2 - 1|^{2q}] + (p^2 - p)\mathbb{E}[|X_{11}X_{12}|^{2q}]) \\
&\leq \tfrac{p^{2q-2}(2q-1)^q}{n^q}(p2^{2q-1}(1 + \mathbb{E}[|X_{11}|^{4q}]) + (p^2 - p)\mathbb{E}[|X_{11}X_{12}|^{2q}]) \\
&= \tfrac{p^{2q-2}(2q-1)^q}{n^q}(p2^{2q-1}(1 + (4q-1)!!) + (p^2 - p)((2q-1)!!)^2) = O(1/n^q).
\end{aligned}$$

The second advertised result now follows by Cauchy–Schwarz.

## D. Proof of Theorem 3.1: Relative instability of ST comparisons

Theorem 3.1 follows immediately from the following two propositions, proved in Appendices E and F, respectively. Note that the first proposition holds for $\lambda_n = o(n)$ and $\delta_n = o(n)$, and does not require the assumption $\|\beta^\star\|_0 < p$, which makes this proposition a stronger result than what is needed for the proof of Theorem 3.1 assuming $\lambda_n = O(\sqrt{n})$, $\lambda_n = \omega(1)$, $\delta_n = \Theta(1)$ and $\|\beta^\star\|_0 < p$.

**Proposition D.1** (Convergence rate of $\sigma^2(h_n^{\mathrm{diff}})$ for comparison of $\mathrm{ST}(\lambda_n)$ with $\mathrm{ST}(\lambda_n + \delta_n)$)**.** *Assume the linear model* (2)*. If $\lambda_n = o(n)$ and $\delta_n = o(n)$, then $\frac{n^2}{\delta_n^2}\sigma^2(h_n^{\mathrm{diff}}) \to 4\tau^2\|\beta\|_0$.*

**Proposition D.2** (Lower-bounding rate of $\gamma(h_n^{\mathrm{diff}})$ for comparison of $\mathrm{ST}(\lambda_n)$ with $\mathrm{ST}(\lambda_n + \delta_n)$)**.** *Assume the linear model* (2)*, and $\|\beta^\star\|_0 < p$. If $\lambda_n = O(\sqrt{n})$, $\lambda_n = \omega(1)$, and $\delta_n = \Theta(1)$, then $\gamma(h_n^{\mathrm{diff}}) = \Omega(\frac{\delta_n^2}{n^2\sqrt{n}})$.*

# E. Proof of Proposition D.1: Convergence rate of $\sigma^2(h_n^{\text{diff}})$ for comparison of $\text{ST}(\lambda_n)$ with $\text{ST}(\lambda_n + \delta_n)$

We start by introducing a lemma which provides key equations in the comparison setting.

**Lemma E.1** (Useful equations for comparison of two linear predictors). *When defining* $h_n(Z_0, \mathbf{Z}) = (Y_0 - X_0^\top \hat{\beta}^{(1)})^2 - (Y_0 - X_0^\top \hat{\beta}^{(2)})^2$, *we have:*

$$
\begin{aligned}
h_n(Z_0, \mathbf{Z}) &= 2Y_0 X_0^\top (\hat{\beta}^{(2)} - \hat{\beta}^{(1)}) + \text{tr}(X_0 X_0^\top (\hat{\beta}^{(1)} \hat{\beta}^{(1)\top} - \hat{\beta}^{(2)} \hat{\beta}^{(2)\top})) \\
\mathbb{E}[h_n(Z_0, \mathbf{Z}) \mid Z_0] &= 2Y_0 X_0^\top \mathbb{E}[\hat{\beta}^{(2)} - \hat{\beta}^{(1)}] + \text{tr}(X_0 X_0^\top \mathbb{E}[\hat{\beta}^{(1)} \hat{\beta}^{(1)\top} - \hat{\beta}^{(2)} \hat{\beta}^{(2)\top}]) \\
\mathbb{E}[h_n(Z_0, \mathbf{Z}) \mid \mathbf{Z}] &= 2\beta^{\star\top} \mathbb{E}[X_0 X_0^\top](\hat{\beta}^{(2)} - \hat{\beta}^{(1)}) + \text{tr}(\mathbb{E}[X_0 X_0^\top](\hat{\beta}^{(1)} \hat{\beta}^{(1)\top} - \hat{\beta}^{(2)} \hat{\beta}^{(2)\top})) \\
\mathbb{E}[h_n(Z_0, \mathbf{Z})] &= 2\beta^{\star\top} \mathbb{E}[X_0 X_0^\top]\mathbb{E}[\hat{\beta}^{(2)} - \hat{\beta}^{(1)}] + \text{tr}(\mathbb{E}[X_0 X_0^\top]\mathbb{E}[\hat{\beta}^{(1)} \hat{\beta}^{(1)\top} - \hat{\beta}^{(2)} \hat{\beta}^{(2)\top}]) \\
\sigma^2(h_n) &= \mathbb{E}[(2(Y_0 X_0^\top - \beta^{\star\top} \mathbb{E}[X_0 X_0^\top])\mathbb{E}[\hat{\beta}^{(2)} - \hat{\beta}^{(1)}] \\
&\qquad + \text{tr}((X_0 X_0^\top - \mathbb{E}[X_0 X_0^\top])\mathbb{E}[\hat{\beta}^{(1)} \hat{\beta}^{(1)\top} - \hat{\beta}^{(2)} \hat{\beta}^{(2)\top}]))^2] \\
\gamma(h_n) &= \mathbb{E}[(2(Y_0 X_0^\top - \beta^{\star\top} \mathbb{E}[X_0 X_0^\top])(\hat{\beta}^{(2)} - \hat{\beta}^{(1)} - (\hat{\beta}'^{(2)} - \hat{\beta}'^{(1)})) \\
&\qquad + \text{tr}((X_0 X_0^\top - \mathbb{E}[X_0 X_0^\top])(\hat{\beta}^{(1)} \hat{\beta}^{(1)\top} - \hat{\beta}^{(2)} \hat{\beta}^{(2)\top} - (\hat{\beta}'^{(1)} \hat{\beta}'^{(1)\top} - \hat{\beta}'^{(2)} \hat{\beta}'^{(2)\top}))))^2]
\end{aligned}
$$

*where* $\hat{\beta}'^{(1)}$ *and* $\hat{\beta}'^{(2)}$ *are the linear predictor counterparts of* $\hat{\beta}^{(1)}$ *and* $\hat{\beta}^{(2)}$, *but learned on a training set* $\mathbf{Z}'$ *that is the same as* $\mathbf{Z}$ *except for the first point* $Z_1$ *being replaced by an i.i.d copy* $Z_1'$.

**Proof** The first equation follows from the first equation of Lemma I.1. The remaining equations are then derived from there using the same arguments as those mentioned in Lemma I.1. $\square$

We will show that

$$
\frac{n}{\delta_n} \mathbb{E}[\hat{\beta}_{\lambda_n + \delta_n} - \hat{\beta}_{\lambda_n}] \to -\text{sign}(\beta^\star)
$$

and

$$
\frac{n}{\delta_n} \mathbb{E}[\hat{\beta}_{\lambda_n + \delta_n} \hat{\beta}_{\lambda_n + \delta_n}^\top - \hat{\beta}_{\lambda_n} \hat{\beta}_{\lambda_n}^\top] \to -(\text{sign}(\beta^\star)\beta^{\star\top} + \beta^\star \text{sign}(\beta^\star)^\top)
$$

where $\text{sign}(\beta^\star) = (\text{sign}(\beta_i^\star))_{i \in [p]}$, in order to conclude that $\frac{n^2}{\delta_n^2} \sigma^2(h_n^{\text{diff}}) \to 4\tau^2 \|\beta^\star\|_0$.

Indeed, if the convergences of these two expectations hold, starting from the expression of $\sigma^2(h_n)$ in Lemma E.1, since $\frac{n}{\delta_n}\mathbb{E}[\hat{\beta}_{\lambda_n + \delta_n} - \hat{\beta}_{\lambda_n}]$ and $\frac{n}{\delta_n}\mathbb{E}[\hat{\beta}_{\lambda_n + \delta_n}\hat{\beta}_{\lambda_n + \delta_n}^\top - \hat{\beta}_{\lambda_n}\hat{\beta}_{\lambda_n}^\top]$ are non-random, we can expand the square, use linearity of expectation, take the limits and factorize back to obtain the following convergence

$$
\frac{n^2}{\delta_n^2} \sigma^2(h_n^{\text{diff}}) \to \mathbb{E}[(2(Y_0 X_0^\top - \beta^{\star\top} \mathbb{E}[X_0 X_0^\top])(-\text{sign}(\beta^\star)) + \text{tr}((X_0 X_0^\top - \mathbb{E}[X_0 X_0^\top])(\text{sign}(\beta^\star)\beta^{\star\top} + \beta^\star \text{sign}(\beta^\star)^\top)))^2]
$$

where, for $Y_0 = X_0^\top \beta^\star + \varepsilon_0$ with $\mathbb{E}[X_0] = 0$ and $\text{Var}(X_0) = \mathbf{I}$,

$$
\begin{aligned}
&\mathbb{E}[(2(Y_0 X_0^\top - \beta^{\star\top} \mathbb{E}[X_0 X_0^\top])(-\text{sign}(\beta^\star)) + \text{tr}((X_0 X_0^\top - \mathbb{E}[X_0 X_0^\top])(\text{sign}(\beta^\star)\beta^{\star\top} + \beta^\star \text{sign}(\beta^\star)^\top)))^2] \\
&= \mathbb{E}[(-2Y_0 X_0^\top \text{sign}(\beta^\star) + 2\beta^{\star\top}\text{sign}(\beta^\star) + 2X_0^\top \beta^\star X_0^\top \text{sign}(\beta^\star) - 2\beta^{\star\top}\text{sign}(\beta^\star)))^2] \\
&= \mathbb{E}[(-2\varepsilon_0 X_0^\top \text{sign}(\beta^\star))^2] \\
&= 4\mathbb{E}[\varepsilon_0^2]\mathbb{E}[(X_0^\top \text{sign}(\beta^\star))^2] \quad \text{by independence of } \varepsilon_0, X_0 \\
&= 4\tau^2 \|\beta^\star\|_0
\end{aligned}
$$

since

$$
\mathbb{E}[(X_0^\top \text{sign}(\beta^\star))^2] = \text{Var}(\text{sign}(\beta^\star)^\top X_0) = \text{sign}(\beta^\star)^\top \text{Var}(X_0)\text{sign}(\beta^\star) = \text{sign}(\beta^\star)^\top \text{sign}(\beta^\star) = \|\beta^\star\|_0.
$$

We have for $i = 1, \ldots, p$,

$$\hat{\beta}_{\lambda_n+\delta_n,i} - \hat{\beta}_{\lambda_n,i} = \text{sign}(\hat{\beta}_{\text{OLS},i})(|\hat{\beta}_{\text{OLS},i}| - \tfrac{\lambda_n+\delta_n}{n})_+ - \text{sign}(\hat{\beta}_{\text{OLS},i})(|\hat{\beta}_{\text{OLS},i}| - \tfrac{\lambda_n}{n})_+$$

$$= -\text{sign}(\hat{\beta}_{\text{OLS},i})\begin{cases} \frac{\delta_n}{n} & \text{if } |\hat{\beta}_{\text{OLS},i}| > \frac{\lambda_n+\delta_n}{n} \\ |\hat{\beta}_{\text{OLS},i}| - \frac{\lambda_n}{n} & \text{if } |\hat{\beta}_{\text{OLS},i}| \in [\frac{\lambda_n}{n}, \frac{\lambda_n+\delta_n}{n}] . \\ 0 & \text{if } |\hat{\beta}_{\text{OLS},i}| < \frac{\lambda_n}{n} \end{cases}$$

Since $\hat{\beta}_{\text{OLS}} \mid \mathbf{X} \sim \mathcal{N}(\beta^\star, \tau^2(\mathbf{X}^\top\mathbf{X})^{-1})$, we can write $\hat{\beta}_{\text{OLS},i} = \beta_i^\star + \tilde{\tau}_n Z$ where $\tilde{\tau}_n = \frac{\tau}{\sqrt{n}}\sqrt{(\frac{\mathbf{X}^\top\mathbf{X}}{n})_{i,i}^{-1}}$ and $Z \mid \mathbf{X} \sim \mathcal{N}(0,1)$. Note that we could have $i$ as a subscript of $\tilde{\tau}_n$ and $Z$, but we will only consider one $i$ at a time in our computations and we can thus omit this subscript for both of them for the sake of notational simplicity, and we will also omit it for some additional notation we define in the rest of the proof.

We now show that $\frac{n}{\delta_n}\mathbb{E}[\hat{\beta}_{\lambda_n+\delta_n,i} - \hat{\beta}_{\lambda_n,i}] \to -\text{sign}(\beta_i^\star)$.

Using the law of total expectation,

$$\mathbb{E}[\hat{\beta}_{\lambda_n+\delta_n,i} - \hat{\beta}_{\lambda_n,i} \mid \mathbf{X}]$$
$$= -\tfrac{\delta_n}{n}\mathbb{P}(\hat{\beta}_{\text{OLS},i} > \tfrac{\lambda_n+\delta_n}{n} \mid \mathbf{X}) + \tfrac{\delta_n}{n}\mathbb{P}(\hat{\beta}_{\text{OLS},i} < -\tfrac{\lambda_n+\delta_n}{n} \mid \mathbf{X})$$
$$- \mathbb{E}[\hat{\beta}_{\text{OLS},i} - \tfrac{\lambda_n}{n} \mid \hat{\beta}_{\text{OLS},i} \in [\tfrac{\lambda_n}{n}, \tfrac{\lambda_n+\delta_n}{n}], \mathbf{X}]\mathbb{P}(\hat{\beta}_{\text{OLS},i} \in [\tfrac{\lambda_n}{n}, \tfrac{\lambda_n+\delta_n}{n}] \mid \mathbf{X})$$
$$- \mathbb{E}[\hat{\beta}_{\text{OLS},i} + \tfrac{\lambda_n}{n} \mid \hat{\beta}_{\text{OLS},i} \in [-\tfrac{\lambda_n+\delta_n}{n}, -\tfrac{\lambda_n}{n}], \mathbf{X}]\mathbb{P}(\hat{\beta}_{\text{OLS},i} \in [-\tfrac{\lambda_n+\delta_n}{n}, -\tfrac{\lambda_n}{n}] \mid \mathbf{X})$$

Define $\alpha_n^{(1)} = \frac{1}{\tilde{\tau}_n}(\frac{\lambda_n}{n} - \beta_i^\star)$, $\alpha_n^{(2)} = \frac{1}{\tilde{\tau}_n}(\frac{\lambda_n}{n} + \beta_i^\star)$, $\theta_n^{(1)} = \frac{1}{\tilde{\tau}_n}(\frac{\lambda_n+\delta_n}{n} - \beta_i^\star)$ and $\theta_n^{(2)} = \frac{1}{\tilde{\tau}_n}(\frac{\lambda_n+\delta_n}{n} + \beta_i^\star)$.

In the order they appear, the four probabilities above are equal to

$$\mathbb{P}(Z > \theta_n^{(1)} \mid \mathbf{X}) = 1 - \Phi(\theta_n^{(1)}),$$

$$\mathbb{P}(Z < -\theta_n^{(2)} \mid \mathbf{X}) = \Phi(-\theta_n^{(2)}),$$

$$\mathbb{P}(Z \in [\alpha_n^{(1)}, \theta_n^{(1)}] \mid \mathbf{X}) = \Phi(\theta_n^{(1)}) - \Phi(\alpha_n^{(1)}),$$

$$\mathbb{P}(Z \in [-\theta_n^{(2)}, -\alpha_n^{(2)}] \mid \mathbf{X}) = \Phi(-\alpha_n^{(2)}) - \Phi(-\theta_n^{(2)}).$$

Using the first moment of the truncated normal (Johnson et al., 1994), we have

$$\mathbb{E}[\hat{\beta}_{\text{OLS},i} - \tfrac{\lambda_n}{n} \mid \hat{\beta}_{\text{OLS},i} \in [\tfrac{\lambda_n}{n}, \tfrac{\lambda_n+\delta_n}{n}], \mathbf{X}] = \beta_i^\star - \tfrac{\lambda_n}{n} + \tilde{\tau}_n \mathbb{E}[Z \mid Z \in [\alpha_n^{(1)}, \theta_n^{(1)}], \mathbf{X}]$$
$$= \beta_i^\star - \tfrac{\lambda_n}{n} - \tilde{\tau}_n \frac{\varphi(\theta_n^{(1)}) - \varphi(\alpha_n^{(1)})}{\Phi(\theta_n^{(1)}) - \Phi(\alpha_n^{(1)})}$$

and

$$\mathbb{E}[\hat{\beta}_{\text{OLS},i} + \tfrac{\lambda_n}{n} \mid \hat{\beta}_{\text{OLS},i} \in [-\tfrac{\lambda_n+\delta_n}{n}, -\tfrac{\lambda_n}{n}], \mathbf{X}] = \beta_i^\star + \tfrac{\lambda_n}{n} + \tilde{\tau}_n \mathbb{E}[Z \mid Z \in [-\theta_n^{(2)}, -\alpha_n^{(2)}], \mathbf{X}]$$
$$= \beta_i^\star + \tfrac{\lambda_n}{n} - \tilde{\tau}_n \frac{\varphi(-\alpha_n^{(2)}) - \varphi(-\theta_n^{(2)})}{\Phi(-\alpha_n^{(2)}) - \Phi(-\theta_n^{(2)})}.$$

Therefore

$$\mathbb{E}[\hat{\beta}_{\lambda_n+\delta_n,i} - \hat{\beta}_{\lambda_n,i} \mid \mathbf{X}]$$
$$= -\frac{\delta_n}{n}\mathbb{P}(\hat{\beta}_{\mathrm{OLS},i} > \frac{\lambda_n+\delta_n}{n} \mid \mathbf{X}) + \frac{\delta_n}{n}\mathbb{P}(\hat{\beta}_{\mathrm{OLS},i} < -\frac{\lambda_n+\delta_n}{n} \mid \mathbf{X})$$
$$\quad - \mathbb{E}[\hat{\beta}_{\mathrm{OLS},i} - \frac{\lambda_n}{n} \mid \hat{\beta}_{\mathrm{OLS},i} \in [\frac{\lambda_n}{n}, \frac{\lambda_n+\delta_n}{n}], \mathbf{X}]\,\mathbb{P}(\hat{\beta}_{\mathrm{OLS},i} \in [\frac{\lambda_n}{n}, \frac{\lambda_n+\delta_n}{n}] \mid \mathbf{X})$$
$$\quad - \mathbb{E}[\hat{\beta}_{\mathrm{OLS},i} + \frac{\lambda_n}{n} \mid \hat{\beta}_{\mathrm{OLS},i} \in [-\frac{\lambda_n+\delta_n}{n}, -\frac{\lambda_n}{n}], \mathbf{X}]\,\mathbb{P}(\hat{\beta}_{\mathrm{OLS},i} \in [-\frac{\lambda_n+\delta_n}{n}, -\frac{\lambda_n}{n}] \mid \mathbf{X})$$
$$= -\frac{\delta_n}{n}(1 - \Phi(\theta_n^{(1)})) + \frac{\delta_n}{n}\Phi(-\theta_n^{(2)})$$
$$\quad - (\beta_i^\star - \frac{\lambda_n}{n})(\Phi(\theta_n^{(1)}) - \Phi(\alpha_n^{(1)})) + \tilde{\tau}_n(\varphi(\theta_n^{(1)}) - \varphi(\alpha_n^{(1)}))$$
$$\quad - (\beta_i^\star + \frac{\lambda_n}{n})(\Phi(-\alpha_n^{(2)}) - \Phi(-\theta_n^{(2)})) + \tilde{\tau}_n(\varphi(-\alpha_n^{(2)}) - \varphi(-\theta_n^{(2)}))$$
$$= -\frac{\delta_n}{n}(1 - \Phi(\theta_n^{(1)})) + \frac{\delta_n}{n}\Phi(-\theta_n^{(2)})$$
$$\quad - (\beta_i^\star - \frac{\lambda_n}{n})(\theta_n^{(1)} - \alpha_n^{(1)})\Phi'(c_n^{(1)}) + \tilde{\tau}_n(\theta_n^{(1)} - \alpha_n^{(1)})\varphi'(d_n^{(1)})$$
$$\quad - (\beta_i^\star + \frac{\lambda_n}{n})(\theta_n^{(2)} - \alpha_n^{(2)})\Phi'(-c_n^{(2)}) + \tilde{\tau}_n(\theta_n^{(2)} - \alpha_n^{(2)})\varphi'(-d_n^{(2)})$$

where $c_n^{(1)}, d_n^{(1)} \in [\alpha_n^{(1)}, \theta_n^{(1)}]$ and $c_n^{(2)}, d_n^{(2)} \in [\alpha_n^{(2)}, \theta_n^{(2)}]$ using first-order Taylor expansions.

We have $\theta_n^{(1)} - \alpha_n^{(1)} = \theta_n^{(2)} - \alpha_n^{(2)} = \frac{1}{\tilde{\tau}_n}\frac{\delta_n}{n}$, $\Phi' = \varphi$ and $\varphi'(x) = -x\varphi(x)$, thus

$$\mathbb{E}[\hat{\beta}_{\lambda_n+\delta_n,i} - \hat{\beta}_{\lambda_n,i} \mid \mathbf{X}] = -\frac{\delta_n}{n}(1 - \Phi(\theta_n^{(1)})) + \frac{\delta_n}{n}\Phi(-\theta_n^{(2)})$$
$$\quad - (\beta_i^\star - \frac{\lambda_n}{n})\frac{1}{\tilde{\tau}_n}\frac{\delta_n}{n}\varphi(c_n^{(1)}) - \tilde{\tau}_n\frac{1}{\tilde{\tau}_n}\frac{\delta_n}{n}d_n^{(1)}\varphi(d_n^{(1)})$$
$$\quad - (\beta_i^\star + \frac{\lambda_n}{n})\frac{1}{\tilde{\tau}_n}\frac{\delta_n}{n}\varphi(-c_n^{(2)}) - \tilde{\tau}_n\frac{1}{\tilde{\tau}_n}\frac{\delta_n}{n}(-d_n^{(2)}\varphi(-d_n^{(2)}))$$
$$= -\frac{\delta_n}{n}(1 - \Phi(\theta_n^{(1)})) + \frac{\delta_n}{n}\Phi(-\theta_n^{(2)})$$
$$\quad - (\beta_i^\star - \frac{\lambda_n}{n})\frac{1}{\tilde{\tau}_n}\frac{\delta_n}{n}\varphi(c_n^{(1)}) - \frac{\delta_n}{n}d_n^{(1)}\varphi(d_n^{(1)})$$
$$\quad - (\beta_i^\star + \frac{\lambda_n}{n})\frac{1}{\tilde{\tau}_n}\frac{\delta_n}{n}\varphi(-c_n^{(2)}) - \frac{\delta_n}{n}(-d_n^{(2)}\varphi(-d_n^{(2)}))$$
$$= -\frac{\delta_n}{n}(1 - \Phi(\theta_n^{(1)})) + \frac{\delta_n}{n}\Phi(-\theta_n^{(2)})$$
$$\quad + \frac{\delta_n}{n}\alpha_n^{(1)}\varphi(c_n^{(1)}) - \frac{\delta_n}{n}d_n^{(1)}\varphi(d_n^{(1)})$$
$$\quad - \frac{\delta_n}{n}\alpha_n^{(2)}\varphi(-c_n^{(2)}) - \frac{\delta_n}{n}(-d_n^{(2)}\varphi(-d_n^{(2)})).$$

We first consider $\beta_i^\star > 0$.

Since $\lambda_n = o(n)$ and $\delta_n = o(n)$, for $n$ large enough, $\frac{\lambda_n+\delta_n}{n} < \beta_i^\star$, so $\alpha_n^{(1)} \le \theta_n^{(1)} < 0$, thus for $c_n^{(1)} \in [\alpha_n^{(1)}, \theta_n^{(1)}]$, we have $|\alpha_n^{(1)}\varphi(c_n^{(1)})| \le |\alpha_n^{(1)}|\varphi(\theta_n^{(1)}) = |\frac{\alpha_n^{(1)}}{\theta_n^{(1)}}||\theta_n^{(1)}|\varphi(\theta_n^{(1)})$, where the ratio $\frac{\alpha_n^{(1)}}{\theta_n^{(1)}} = \frac{\frac{\lambda_n}{n} - \beta_i^\star}{\frac{\lambda_n+\delta_n}{n} - \beta_i^\star}$ is deterministic and goes to 1.

As $-\theta_n^{(2)} \le -\alpha_n^{(2)} < 0$, for $c_n^{(1)} \in [-\theta_n^{(2)}, -\alpha_n^{(2)}]$, we have $|-\alpha_n^{(2)}\varphi(-c_n^{(2)})| \le |-\alpha_n^{(2)}\varphi(-\alpha_n^{(2)})|$.

Since $\frac{\mathbf{X}^\top\mathbf{X}}{n} \xrightarrow{\text{a.s.}} \mathbb{E}[X_0 X_0^\top]$ (strong law of large numbers), $\lambda_n = o(n)$ and $\delta_n = o(n)$, we have $\tilde{\tau}_n \xrightarrow{\text{a.s.}} 0^+$, and using the continuous mapping theorem, $\alpha_n^{(1)} \xrightarrow{\text{a.s.}} -\infty$, $\theta_n^{(1)} \xrightarrow{\text{a.s.}} -\infty$, $\alpha_n^{(2)} \xrightarrow{\text{a.s.}} +\infty$ and $\theta_n^{(2)} \xrightarrow{\text{a.s.}} +\infty$. We then also have $d_n^{(1)} \xrightarrow{\text{a.s.}} -\infty$ and $d_n^{(2)} \xrightarrow{\text{a.s.}} +\infty$.

$\Phi$ and $x \mapsto x\varphi(x)$ are continuous bounded functions so we get $L^1$ convergence of $\Phi(\theta_n^{(1)})$, $\Phi(-\theta_n^{(2)})$, $\theta_n^{(1)}\varphi(\theta_n^{(1)})$, $-\alpha_n^{(2)}\varphi(-\alpha_n^{(2)})$, $d_n^{(1)}\varphi(d_n^{(1)})$ and $-d_n^{(2)}\varphi(-d_n^{(2)})$ to 0. By putting everything together, we obtain

$$\frac{n}{\delta_n}\mathbb{E}[\hat{\beta}_{\lambda_n+\delta_n,i} - \hat{\beta}_{\lambda_n,i}] = \frac{n}{\delta_n}\mathbb{E}[\mathbb{E}[\hat{\beta}_{\lambda_n+\delta_n,i} - \hat{\beta}_{\lambda_n,i} \mid \mathbf{X}]] \to -1 = -\mathrm{sign}(\beta_i^\star).$$

When $\beta_i^\star < 0$, we show in a similar manner that

$$\frac{n}{\delta_n}\mathbb{E}[\hat{\beta}_{\lambda_n+\delta_n,i} - \hat{\beta}_{\lambda_n,i}] \to 1 = -\mathrm{sign}(\beta_i^\star).$$

If $\beta_i^\star = 0$, $\alpha_n^{(1)} = \alpha_n^{(2)}$ and $\theta_n^{(1)} = \theta_n^{(2)}$ so $1 - \Phi(\alpha_n^{(1)}) = \Phi(-\alpha_n^{(2)})$, $\varphi(\alpha_n^{(1)}) = \varphi(-\alpha_n^{(2)})$, $1 - \Phi(\theta_n^{(1)}) = \Phi(-\theta_n^{(2)})$ and $\varphi(\theta_n^{(1)}) = \varphi(-\theta_n^{(2)})$ which leads to

$$
\begin{aligned}
&\mathbb{E}[\hat{\beta}_{\lambda_n+\delta_n,i} - \hat{\beta}_{\lambda_n,i} \mid \mathbf{X}] \\
&= -\frac{\delta_n}{n}(1 - \Phi(\theta_n^{(1)})) + \frac{\delta_n}{n}\Phi(-\theta_n^{(2)}) \\
&\quad - (\beta_i^\star - \frac{\lambda_n}{n})(\Phi(\theta_n^{(1)}) - \Phi(\alpha_n^{(1)})) + \tilde{\tau}_n(\varphi(\theta_n^{(1)}) - \varphi(\alpha_n^{(1)})) \\
&\quad - (\beta_i^\star + \frac{\lambda_n}{n})(\Phi(-\alpha_n^{(2)}) - \Phi(-\theta_n^{(2)})) + \tilde{\tau}_n(\varphi(-\alpha_n^{(2)}) - \varphi(-\theta_n^{(2)})) \\
&= 0
\end{aligned}
$$

and thus $\mathbb{E}[\hat{\beta}_{\lambda_n+\delta_n,i} - \hat{\beta}_{\lambda_n,i}] = 0 = \text{sign}(\beta_i^\star)$.

Thus, we have convergence component-wise and can conclude $\frac{n}{\delta_n}\mathbb{E}[\hat{\beta}_{\lambda_n+\delta_n} - \hat{\beta}_{\lambda_n}] \to -\text{sign}(\beta^\star)$.

We now show that $\frac{n}{\delta_n}\mathbb{E}[\hat{\beta}_{\lambda_n+\delta_n,i}\hat{\beta}_{\lambda_n+\delta_n,j} - \hat{\beta}_{\lambda_n,i}\hat{\beta}_{\lambda_n,j}] \to -(\text{sign}(\beta_i^\star)\beta_j^\star + \beta_i^\star\text{sign}(\beta_j^\star))$.

Note that

$$
\begin{aligned}
&\mathbb{E}[\frac{n}{\delta_n}(\hat{\beta}_{\lambda_n+\delta_n,i}\hat{\beta}_{\lambda_n+\delta_n,j} - \hat{\beta}_{\lambda_n,i}\hat{\beta}_{\lambda_n,j}) + \text{sign}(\beta_i^\star)\beta_j^\star + \beta_i^\star\text{sign}(\beta_j^\star)] \\
&= \mathbb{E}[\frac{n}{\delta_n}(\hat{\beta}_{\lambda_n+\delta_n,i} - \hat{\beta}_{\lambda_n,i})\hat{\beta}_{\lambda_n+\delta_n,j} + \text{sign}(\beta_i^\star)\beta_j^\star] + \mathbb{E}[\hat{\beta}_{\lambda_n,i}\frac{n}{\delta_n}(\hat{\beta}_{\lambda_n+\delta_n,j} - \hat{\beta}_{\lambda_n,j}) + \beta_i^\star\text{sign}(\beta_j^\star)]
\end{aligned}
$$

with

$$
\begin{aligned}
&\mathbb{E}[\frac{n}{\delta_n}(\hat{\beta}_{\lambda_n+\delta_n,i} - \hat{\beta}_{\lambda_n,i})\hat{\beta}_{\lambda_n+\delta_n,j} + \text{sign}(\beta_i^\star)\beta_j^\star] \\
&= \mathbb{E}[(\frac{n}{\delta_n}(\hat{\beta}_{\lambda_n+\delta_n,i} - \hat{\beta}_{\lambda_n,i}) + \text{sign}(\beta_i^\star))(\hat{\beta}_{\lambda_n+\delta_n,j} - \beta_j^\star)] \\
&\quad + \beta_j^\star\mathbb{E}[\frac{n}{\delta_n}(\hat{\beta}_{\lambda_n+\delta_n,i} - \hat{\beta}_{\lambda_n,i}) + \text{sign}(\beta_i^\star)] - \text{sign}(\beta_i^\star)\mathbb{E}[\hat{\beta}_{\lambda_n+\delta_n,j} - \beta_j^\star]
\end{aligned}
$$

where, using Cauchy–Schwarz,

$$
\begin{aligned}
&\mathbb{E}[(\frac{n}{\delta_n}(\hat{\beta}_{\lambda_n+\delta_n,i} - \hat{\beta}_{\lambda_n,i}) + \text{sign}(\beta_i^\star))(\hat{\beta}_{\lambda_n+\delta_n,j} - \beta_j^\star)] \\
&\leq \sqrt{\mathbb{E}[(\frac{n}{\delta_n}(\hat{\beta}_{\lambda_n+\delta_n,i} - \hat{\beta}_{\lambda_n,i}) + \text{sign}(\beta_i^\star))^2]\mathbb{E}[(\hat{\beta}_{\lambda_n+\delta_n,j} - \beta_j^\star)^2]}.
\end{aligned}
$$

We can do the same with $\mathbb{E}[\hat{\beta}_{\lambda_n,i}\frac{n}{\delta_n}(\hat{\beta}_{\lambda_n+\delta_n,j} - \hat{\beta}_{\lambda_n,j}) + \beta_i^\star\text{sign}(\beta_j^\star)]$.

Therefore, proving $\mathbb{E}[\frac{n}{\delta_n}(\hat{\beta}_{\lambda_n+\delta_n,i}\hat{\beta}_{\lambda_n+\delta_n,j} - \hat{\beta}_{\lambda_n,i}\hat{\beta}_{\lambda_n,j}) + \text{sign}(\beta_i^\star)\beta_j^\star + \beta_i^\star\text{sign}(\beta_j^\star)] \to 0$ for all $i,j$ comes down to proving $\mathbb{E}[(\frac{n}{\delta_n}(\hat{\beta}_{\lambda_n+\delta_n,i} - \hat{\beta}_{\lambda_n,i}) + \text{sign}(\beta_i^\star))^2] = O(1)$ for all $i$ given that we have already shown for all $i$, accounting for the fact that both $\lambda_n$ and $\delta_n$ are $o(n)$,

- $\mathbb{E}[\hat{\beta}_{\lambda_n,i}] \to \beta_i^\star$ and $\mathbb{E}[\hat{\beta}_{\lambda_n+\delta_n,i}] \to \beta_i^\star$,

- $\mathbb{E}[(\hat{\beta}_{\lambda_n,i} - \beta_i^\star)^2] \to 0$ and $\mathbb{E}[(\hat{\beta}_{\lambda_n+\delta_n,i} - \beta_i^\star)^2] \to 0$,

- $\frac{n}{\delta_n}\mathbb{E}[\hat{\beta}_{\lambda_n+\delta_n,i} - \hat{\beta}_{\lambda_n,i}] \to -\text{sign}(\beta_i^\star)$.

The first two bullet points were proved in Appendix I and the third one earlier in this proof.

As a reminder, we have

$$
\hat{\beta}_{\lambda_n+\delta_n,i} - \hat{\beta}_{\lambda_n,i} = -\text{sign}(\hat{\beta}_{\text{OLS},i})\begin{cases} \frac{\delta_n}{n} & \text{if } |\hat{\beta}_{\text{OLS},i}| > \frac{\lambda_n+\delta_n}{n} \\ |\hat{\beta}_{\text{OLS},i}| - \frac{\lambda_n}{n} & \text{if } |\hat{\beta}_{\text{OLS},i}| \in [\frac{\lambda_n}{n}, \frac{\lambda_n+\delta_n}{n}] \\ 0 & \text{if } |\hat{\beta}_{\text{OLS},i}| < \frac{\lambda_n}{n} \end{cases}
$$

thus

$$
(\hat{\beta}_{\lambda_n+\delta_n,i} - \hat{\beta}_{\lambda_n,i})^2 \leq \frac{\delta_n^2}{n^2}
$$

and

$$(\tfrac{n}{\delta_n}(\hat{\beta}_{\lambda_n+\delta_n,i} - \hat{\beta}_{\lambda_n,i}) + \mathrm{sign}(\beta_i^\star))^2 \le 2(\tfrac{n^2}{\delta_n^2}(\hat{\beta}_{\lambda_n+\delta_n,i} - \hat{\beta}_{\lambda_n,i})^2 + \mathrm{sign}(\beta_i^\star)^2) \le 4.$$

Hence, $\mathbb{E}[(\tfrac{n}{\delta_n}(\hat{\beta}_{\lambda_n+\delta_n,i} - \hat{\beta}_{\lambda_n,i}) + \mathrm{sign}(\beta_i^\star))^2] = O(1)$.

Therefore, we get

$$\frac{n}{\delta_n}\mathbb{E}[\hat{\beta}_{\lambda_n+\delta_n}\hat{\beta}_{\lambda_n+\delta_n}^\top - \hat{\beta}_{\lambda_n}\hat{\beta}_{\lambda_n}^\top] \to -(\mathrm{sign}(\beta^\star)\beta^{\star\top} + \beta^\star\mathrm{sign}(\beta^\star)^\top).$$

We can then conclude that $\frac{n^2}{\delta_n^2}\sigma^2(h_n^{\mathrm{diff}}) \to 4\tau^2\|\beta^\star\|_0$ as mentioned earlier in the proof.

# F. Proof of Proposition D.2: Lower-bounding rate of $\gamma(h_n^{\mathrm{diff}})$ for comparison of $\mathrm{ST}(\lambda_n)$ with $\mathrm{ST}(\lambda_n + \delta_n)$

Starting from the expression for $\gamma(h_n)$ stated in Lemma E.1, we have

$$\gamma(h_n^{\mathrm{diff}}) = \mathbb{E}[(2(Y_0 X_0^\top - \beta^{\star\top}\mathbb{E}[X_0 X_0^\top])\nu_n + \mathrm{tr}((X_0 X_0^\top - \mathbb{E}[X_0 X_0^\top])\Psi_n))^2].$$

where

- $\nu_n \triangleq \hat{\beta}_{\lambda_n+\delta_n} - \hat{\beta}_{\lambda_n} - (\hat{\beta}'_{\lambda_n+\delta_n} - \hat{\beta}'_{\lambda_n})$,

- $\Psi_n \triangleq \hat{\beta}_{\lambda_n}\hat{\beta}_{\lambda_n}^\top - \hat{\beta}_{\lambda_n+\delta_n}\hat{\beta}_{\lambda_n+\delta_n}^\top - (\hat{\beta}'_{\lambda_n}\hat{\beta}'^\top_{\lambda_n} - \hat{\beta}'_{\lambda_n+\delta_n}\hat{\beta}'^\top_{\lambda_n+\delta_n})$.

$\mathbb{E}[X_0 X_0^\top] = \mathbf{I}$ since the features are drawn from $\mathcal{N}(0,\mathbf{I})$, and using independence of $Z_0$ from the training points, we have

$$
\begin{aligned}
\gamma(h_n^{\mathrm{diff}}) &= \mathbb{E}[(2\sum_i(Y_0 X_{0,i} - \beta_i^\star)\nu_{n,i} + \sum_{i,j}(X_{0,i}X_{0,j} - \mathbb{1}[i=j])\Psi_{n,i,j}))^2] \\
&= 4\sum_i \mathbb{E}[(Y_0 X_{0,i} - \beta_i^\star)^2]\mathbb{E}[\nu_{n,i}^2] \\
&\quad + 4\sum_{i\ne j}\mathbb{E}[(Y_0 X_{0,i} - \beta_i^\star)(Y_0 X_{0,j} - \beta_j^\star)]\mathbb{E}[\nu_{n,i}\nu_{n,j}] \\
&\quad + 4\sum_{i,j,k}\mathbb{E}[(Y_0 X_{0,i} - \beta_i^\star)(X_{0,j}X_{0,k} - \mathbb{1}[j=k])]\mathbb{E}[\nu_{n,i}\Psi_{n,j,k}] \\
&\quad + \sum_{i,j,k,l}\mathbb{E}[(X_{0,i}X_{0,j} - \mathbb{1}[i=j])(X_{0,k}X_{0,l} - \mathbb{1}[k=l])]\mathbb{E}[\Psi_{n,i,j}\Psi_{n,k,l}].
\end{aligned}
$$

Since $Y_0 = X_0^\top\beta^\star + \varepsilon_0 = \sum_k X_{0,k}\beta_k^\star + \varepsilon_0$ with $X_0 \sim \mathcal{N}(0,\mathbf{I})$ and $\varepsilon_0 \perp\!\!\!\perp X_0$, we have

$$\mathbb{E}[Y_0 X_{0,i}] = \beta_i^\star\mathbb{E}[X_{0,i}^2] + \sum_{k\ne i}\beta_k^\star\mathbb{E}[X_{0,i}X_{0,k}] + \mathbb{E}[\varepsilon_0 X_{0,i}] = \beta_i^\star$$

and $Y_0^2 = \sum_{k,l}X_{0,k}X_{0,l}\beta_k^\star\beta_l^\star + 2\varepsilon_0\sum_k X_{0,k}\beta_k^\star + \varepsilon_0^2$, so for $i \ne j$,

$$\mathbb{E}[Y_0^2 X_{0,i}X_{0,j}] = \sum_{k,l}\mathbb{E}[X_{0,i}X_{0,j}X_{0,k}X_{0,l}]\beta_k^\star\beta_l^\star + 2\sum_k\mathbb{E}[\varepsilon_0 X_{0,i}X_{0,j}X_{0,k}]\beta_k^\star + \mathbb{E}[\varepsilon_0^2 X_{0,i}X_{0,j}] = 2\beta_i^\star\beta_j^\star$$

since the expectation in the first sum is equal to 1 when $k=i, l=j$ or $k=j, l=i$, and equal to 0 otherwise, and thus, for $i \ne j$,

$$\mathbb{E}[(Y_0 X_{0,i} - \beta_i^\star)(Y_0 X_{0,j} - \beta_j^\star)] = \mathbb{E}[Y_0^2 X_{0,i}X_{0,j}] - \beta_i^\star\mathbb{E}[Y_0 X_{0,j}] - \beta_j^\star\mathbb{E}[Y_0 X_{0,i}] + \beta_i^\star\beta_j^\star = \beta_i^\star\beta_j^\star.$$

For the case $i = j$,

$$
\begin{aligned}
\mathbb{E}[Y_0^2 X_{0,i}^2] &= \sum_{k,l}\mathbb{E}[X_{0,i}^2 X_{0,k}X_{0,l}]\beta_k^\star\beta_l^\star + 2\sum_k\mathbb{E}[\varepsilon_0 X_{0,i}^2 X_{0,k}]\beta_k^\star + \mathbb{E}[\varepsilon_0^2 X_{0,i}^2] \\
&= \mathbb{E}[X_{0,i}^4]\beta_i^{\star2} + \sum_{k\ne i}\mathbb{E}[X_{0,i}^2 X_{0,k}^2]\beta_k^{\star2} + \tau^2 \\
&= \mathbb{E}[X_{0,i}^4]\beta_i^{\star2} + \sum_{k\ne i}\beta_k^{\star2} + \tau^2
\end{aligned}
$$

and then, for $\beta_i^\star = 0$,

$$\mathbb{E}[(Y_0 X_{0,i} - \beta_i^\star)^2] = \mathbb{E}[Y_0^2 X_{0,i}^2] = \sum_{k\ne i}\beta_k^{\star2} + \tau^2 \ge \tau^2 > 0.$$

Therefore

$$\begin{aligned}
\gamma(h_n^{\mathrm{diff}}) = \ &4\sum_{i,\beta_i^\star=0}\mathbb{E}[Y_0^2 X_{0,i}^2]\mathbb{E}[\nu_{n,i}^2]\\
&+ 4\sum_{i,\beta_i^\star\neq0}\mathbb{E}[(Y_0 X_{0,i} - \beta_i^\star)^2]\mathbb{E}[\nu_{n,i}^2]\\
&+ 4\sum_{i\neq j,\beta_i^\star\neq0,\beta_j^\star\neq0}\beta_i^\star\beta_j^\star\mathbb{E}[\nu_{n,i}\nu_{n,j}]\\
&+ 4\sum_{i,j,k}\mathbb{E}[(Y_0 X_{0,i} - \beta_i^\star)(X_{0,j}X_{0,k} - \mathbb{1}[j=k])]\mathbb{E}[\nu_{n,i}\Psi_{n,j,k}]\\
&+ \sum_{i,j,k,l}\mathbb{E}[(X_{0,i}X_{0,j} - \mathbb{1}[i=j])(X_{0,k}X_{0,l} - \mathbb{1}[k=l])]\mathbb{E}[\Psi_{n,i,j}\Psi_{n,k,l}].
\end{aligned}$$

where importantly we were able to remove the $i,j$ terms in the third sum when $\beta_i^\star = 0$ or $\beta_j^\star = 0$.

We will now prove the following results:

- $\mathbb{E}[\nu_{n,i}^2] = O(\frac{\delta_n^2}{n^2})$ for all $i$,

- $\mathbb{E}[\nu_{n,i}^2] = \Omega(\frac{\delta_n^2}{n^2\sqrt{n}})$ for all $i$ such that $\beta_i^\star = 0$,

- $\mathbb{E}[\nu_{n,i}^2] = o(\frac{\delta_n^2}{n^2\sqrt{n}})$ for all $i$ such that $\beta_i^\star \neq 0$,

- $\mathbb{E}[\Psi_{n,i,j}^2] = O(\frac{\delta_n^2}{n^4})$ for all $i,j$.

Once we prove these, Cauchy–Schwarz will yield the following upper-bounding rates for terms appearing in the expression of $\gamma(h_n^{\mathrm{diff}})$:

- for $i,j$ such that $\beta_i^\star \neq 0$ and $\beta_j^\star \neq 0$, $|\mathbb{E}[\nu_{n,i}\nu_{n,j}]| \leq \sqrt{\mathbb{E}[\nu_{n,i}^2]\mathbb{E}[\nu_{n,j}^2]} = o(\frac{\delta_n^2}{n^2\sqrt{n}})$,

- $|\mathbb{E}[\nu_{n,i}\Psi_{n,j,k}]| \leq \sqrt{\mathbb{E}[\nu_{n,i}^2]\mathbb{E}[\Psi_{n,j,k}^2]} = O(\sqrt{\frac{\delta_n^2}{n^2}\frac{\delta_n^2}{n^4}}) = O(\frac{\delta_n^2}{n^3}) = o(\frac{\delta_n^2}{n^2\sqrt{n}})$,

- $|\mathbb{E}[\Psi_{n,i,j}\Psi_{n,k,l}]| \leq \sqrt{\mathbb{E}[\Psi_{n,i,j}^2]\mathbb{E}[\Psi_{n,k,l}^2]} = O(\sqrt{\frac{\delta_n^2}{n^4}\frac{\delta_n^2}{n^4}}) = O(\frac{\delta_n^2}{n^4}) = o(\frac{\delta_n^2}{n^2\sqrt{n}})$,

and it will therefore be clear that $\gamma(h_n^{\mathrm{diff}}) = \Omega(\frac{\delta_n^2}{n^2\sqrt{n}})$ as the terms of leading order in $\gamma(h_n^{\mathrm{diff}})$ will be the $\mathbb{E}[\nu_{n,i}^2]$ terms for $i$ such that $\beta_i^\star = 0$.

We will now prove the first result $\mathbb{E}[\nu_{n,i}^2] = O(\frac{\delta_n^2}{n^2})$ for all $i$.

We have

$$\begin{aligned}
\nu_{n,i} = \ &\hat{\beta}_{\lambda_n+\delta_n,i} - \hat{\beta}_{\lambda_n,i} - (\hat{\beta}'_{\lambda_n+\delta_n,i} - \hat{\beta}'_{\lambda_n,i})\\
= \ &\mathrm{sign}(\hat{\beta}_{\mathrm{OLS},i})(|\hat{\beta}_{\mathrm{OLS},i}| - \tfrac{\lambda_n+\delta_n}{n})_+ - \mathrm{sign}(\hat{\beta}_{\mathrm{OLS},i})(|\hat{\beta}_{\mathrm{OLS},i}| - \tfrac{\lambda_n}{n})_+\\
&- (\mathrm{sign}(\hat{\beta}'_{\mathrm{OLS},i})(|\hat{\beta}'_{\mathrm{OLS},i}| - \tfrac{\lambda_n+\delta_n}{n})_+ - \mathrm{sign}(\hat{\beta}'_{\mathrm{OLS},i})(|\hat{\beta}'_{\mathrm{OLS},i}| - \tfrac{\lambda_n}{n})_+)\\
= \ &\mathrm{sign}(\hat{\beta}_{\mathrm{OLS},i})\begin{cases}-\frac{\delta_n}{n} & \text{if } |\hat{\beta}_{\mathrm{OLS},i}| > \frac{\lambda_n+\delta_n}{n}\\ \frac{\lambda_n}{n} - |\hat{\beta}_{\mathrm{OLS},i}| & \text{if } |\hat{\beta}_{\mathrm{OLS},i}| \in [\frac{\lambda_n}{n}, \frac{\lambda_n+\delta_n}{n}]\\ 0 & \text{if } |\hat{\beta}_{\mathrm{OLS},i}| < \frac{\lambda_n}{n}\end{cases}\\
&- \mathrm{sign}(\hat{\beta}'_{\mathrm{OLS},i})\begin{cases}-\frac{\delta_n}{n} & \text{if } |\hat{\beta}'_{\mathrm{OLS},i}| > \frac{\lambda_n+\delta_n}{n}\\ \frac{\lambda_n}{n} - |\hat{\beta}'_{\mathrm{OLS},i}| & \text{if } |\hat{\beta}'_{\mathrm{OLS},i}| \in [\frac{\lambda_n}{n}, \frac{\lambda_n+\delta_n}{n}].\\ 0 & \text{if } |\hat{\beta}'_{\mathrm{OLS},i}| < \frac{\lambda_n}{n}\end{cases}
\end{aligned}$$

We can observe that both $|\hat{\beta}_{\lambda_n+\delta_n,i} - \hat{\beta}_{\lambda_n,i}|$ and $|\hat{\beta}'_{\lambda_n+\delta_n,i} - \hat{\beta}'_{\lambda_n,i}|$ are upper-bounded by $\frac{\delta_n}{n}$ and thus $\nu_{n,i}^2 \leq 4\frac{\delta_n^2}{n^2}$, which implies $\mathbb{E}[\nu_{n,i}^2] = O(\frac{\delta_n^2}{n^2})$ for all $i$.

We will now prove the second result $\mathbb{E}[\nu_{n,i}^2] = \Omega(\frac{\delta_n^2}{n^2\sqrt{n}})$ for all $i$ such that $\beta_i^\star = 0$.

Based on the previous expression, we can further detail $\nu_{n,i}$ as follows

$$\nu_{n,i} = \begin{cases} -\frac{\delta_n}{n} & \text{if } \hat{\beta}_{\text{OLS},i} > \frac{\lambda_n + \delta_n}{n} \\ \frac{\delta_n}{n} & \text{if } \hat{\beta}_{\text{OLS},i} < -\frac{\lambda_n + \delta_n}{n} \\ \frac{\lambda_n}{n} - \hat{\beta}_{\text{OLS},i} & \text{if } \hat{\beta}_{\text{OLS},i} \in [\frac{\lambda_n}{n}, \frac{\lambda_n + \delta_n}{n}] \\ -\frac{\lambda_n}{n} - \hat{\beta}_{\text{OLS},i} & \text{if } \hat{\beta}_{\text{OLS},i} \in [-\frac{\lambda_n + \delta_n}{n}, -\frac{\lambda_n}{n}] \\ 0 & \text{if } |\hat{\beta}_{\text{OLS},i}| < \frac{\lambda_n}{n} \end{cases} - \begin{cases} -\frac{\delta_n}{n} & \text{if } \hat{\beta}'_{\text{OLS},i} > \frac{\lambda_n + \delta_n}{n} \\ \frac{\delta_n}{n} & \text{if } \hat{\beta}'_{\text{OLS},i} < -\frac{\lambda_n + \delta_n}{n} \\ \frac{\lambda_n}{n} - \hat{\beta}'_{\text{OLS},i} & \text{if } \hat{\beta}'_{\text{OLS},i} \in [\frac{\lambda_n}{n}, \frac{\lambda_n + \delta_n}{n}] \\ -\frac{\lambda_n}{n} - \hat{\beta}'_{\text{OLS},i} & \text{if } \hat{\beta}'_{\text{OLS},i} \in [-\frac{\lambda_n + \delta_n}{n}, -\frac{\lambda_n}{n}] \\ 0 & \text{if } |\hat{\beta}'_{\text{OLS},i}| < \frac{\lambda_n}{n} \end{cases}$$

which means there are 25 possible cases that form a partition and we can write $\nu_{n,i}$ as the sum of 25 terms that are of the form: an indicator of one of the 25 events multiplied by the value of $\nu_{n,i}$ for this event. We can then similarly write $\nu_{n,i}^2$ as the sum of 25 terms that are of the form: an indicator of one of the 25 events multiplied by the value of $\nu_{n,i}^2$ for this event.

We can then lower-bound $\mathbb{E}[\nu_{n,i}^2]$ by the expectation of any one of the 25 terms since they are all non-negative. In particular, we can do it using the term coming from the combination of the first case on the left side and the last case on the right side

$$\mathbb{E}[\nu_{n,i}^2] \geq \mathbb{E}[\tfrac{\delta_n^2}{n^2}\mathbb{1}[\hat{\beta}_{\text{OLS},i} > \tfrac{\lambda_n + \delta_n}{n}, |\hat{\beta}'_{\text{OLS},i}| < \tfrac{\lambda_n}{n}]]$$
$$= \tfrac{\delta_n^2}{n^2}\mathbb{P}(\hat{\beta}_{\text{OLS},i} > \tfrac{\lambda_n + \delta_n}{n}, |\hat{\beta}'_{\text{OLS},i}| < \tfrac{\lambda_n}{n}).$$

Since $\lambda_n = \omega(1)$ and $\delta_n = \Theta(1)$, $\frac{\lambda_n}{n} - \frac{\delta_n}{n} > 0$ for $n$ large enough, and we then have $\{\hat{\beta}_{\text{OLS},i} > \hat{\beta}'_{\text{OLS},i} + 2\frac{\delta_n}{n}, \hat{\beta}'_{\text{OLS},i} \in [\frac{\lambda_n}{n} - \frac{\delta_n}{n}, \frac{\lambda_n}{n}]\} \subseteq \{\hat{\beta}_{\text{OLS},i} > \frac{\lambda_n + \delta_n}{n}, |\hat{\beta}'_{\text{OLS},i}| < \frac{\lambda_n}{n}\}$, therefore

$$\mathbb{P}(\hat{\beta}_{\text{OLS},i} > \tfrac{\lambda_n + \delta_n}{n}, |\hat{\beta}'_{\text{OLS},i}| < \tfrac{\lambda_n}{n})$$
$$\geq \mathbb{P}(\hat{\beta}_{\text{OLS},i} > \hat{\beta}'_{\text{OLS},i} + 2\tfrac{\delta_n}{n}, \hat{\beta}'_{\text{OLS},i} \in [\tfrac{\lambda_n}{n} - \tfrac{\delta_n}{n}, \tfrac{\lambda_n}{n}])$$
$$= \mathbb{P}(n(\hat{\beta}'_{\text{OLS},i} - \hat{\beta}_{\text{OLS},i}) < -2\delta_n, \hat{\beta}'_{\text{OLS},i} \in [\tfrac{\lambda_n}{n} - \tfrac{\delta_n}{n}, \tfrac{\lambda_n}{n}]).$$

We have

$$\text{Cov}(\hat{\beta}_{\text{OLS},i}, \hat{\beta}'_{\text{OLS},i} \mid \mathbf{X}, \mathbf{X}') = \text{Cov}(\beta^\star + (\mathbf{X}^\top\mathbf{X})^{-1}\mathbf{X}^\top\varepsilon, \beta^\star + (\mathbf{X}'^\top\mathbf{X}')^{-1}\mathbf{X}'^\top\varepsilon' \mid \mathbf{X}, \mathbf{X}')$$
$$= (\mathbf{X}^\top\mathbf{X})^{-1}\mathbf{X}^\top\text{Cov}(\varepsilon, \varepsilon')\mathbf{X}'(\mathbf{X}'^\top\mathbf{X}')^{-1}$$
$$= \tau^2(\mathbf{X}^\top\mathbf{X})^{-1}\tilde{\mathbf{X}}^\top\tilde{\mathbf{X}}(\mathbf{X}'^\top\mathbf{X}')^{-1}$$

where $\tilde{\mathbf{X}} \triangleq (X_2, \ldots, X_n)^\top$ is the matrix of regressors for the training points except for the first one that is being changed, since $\text{Cov}(\varepsilon_i, \varepsilon'_j)$ is equal to $\tau^2$ if $i = j \geq 2$ and 0 otherwise. Then

$$\text{Cov}(\hat{\beta}'_{\text{OLS},i} - \hat{\beta}_{\text{OLS},i}, \hat{\beta}'_{\text{OLS},i} \mid \mathbf{X}, \mathbf{X}') = \tau^2(\mathbf{X}'^\top\mathbf{X}')^{-1} - \tau^2(\mathbf{X}^\top\mathbf{X})^{-1}\tilde{\mathbf{X}}^\top\tilde{\mathbf{X}}(\mathbf{X}'^\top\mathbf{X}')^{-1}$$
$$= \tau^2(\mathbf{I} - (\mathbf{X}^\top\mathbf{X})^{-1}\tilde{\mathbf{X}}^\top\tilde{\mathbf{X}})(\mathbf{X}'^\top\mathbf{X}')^{-1}.$$

Hence, the bivariate normal vector $(\hat{\beta}'_{\text{OLS},i} - \hat{\beta}_{\text{OLS},i}, \hat{\beta}'_{\text{OLS},i})$ has uncorrelated components in the limit, with zero correlation being equivalent to independence for multivariate normal vectors. Since $n(\hat{\beta}'_{\text{OLS}} - \hat{\beta}_{\text{OLS}}) \xrightarrow{\text{a.s.}} V \triangleq (Y'_1 - X'^\top_1\beta^\star)X'_1 - (Y_1 - X_1^\top\beta^\star)X_1$, proved in Appendix J, and $\delta_n = \Theta(1)$, we have

$$\mathbb{P}(n(\hat{\beta}'_{\text{OLS},i} - \hat{\beta}_{\text{OLS},i}) < -2\delta_n, \hat{\beta}'_{\text{OLS},i} \in [\tfrac{\lambda_n}{n} - \tfrac{\delta_n}{n}, \tfrac{\lambda_n}{n}]) = \Theta(\mathbb{P}(\hat{\beta}'_{\text{OLS},i} \in [\tfrac{\lambda_n}{n} - \tfrac{\delta_n}{n}, \tfrac{\lambda_n}{n}])).$$

We can then focus on the rate of $\mathbb{P}(\hat{\beta}'_{\text{OLS},i} \in [\frac{\lambda_n}{n} - \frac{\delta_n}{n}, \frac{\lambda_n}{n}])$.

$$\mathbb{P}(\hat{\beta}'_{\text{OLS},i} \in [\tfrac{\lambda_n}{n} - \tfrac{\delta_n}{n}, \tfrac{\lambda_n}{n}]) = \mathbb{E}[\mathbb{P}(\hat{\beta}'_{\text{OLS},i} \in [\tfrac{\lambda_n}{n} - \tfrac{\delta_n}{n}, \tfrac{\lambda_n}{n}] \mid \mathbf{X}')]$$

where, using $\beta_i^\star = 0$ and $\hat{\beta}'_{\text{OLS}} \mid \mathbf{X} \sim \mathcal{N}(\beta^\star, \tau^2(\mathbf{X}'^\top\mathbf{X}')^{-1})$ and defining $\tilde{\tau}'_n = \frac{\tau}{\sqrt{n}}\sqrt{(\frac{\mathbf{X}'^\top\mathbf{X}'}{n})_{i,i}^{-1}}$,

$$\mathbb{P}(\hat{\beta}'_{\text{OLS},i} \in [\tfrac{\lambda_n}{n} - \tfrac{\delta_n}{n}, \tfrac{\lambda_n}{n}] \mid \mathbf{X}') = \Phi(\tfrac{1}{\tilde{\tau}'_n}\tfrac{\lambda_n}{n}) - \Phi(\tfrac{1}{\tilde{\tau}'_n}(\tfrac{\lambda_n}{n} - \tfrac{\delta_n}{n}))$$
$$= \tfrac{1}{\tilde{\tau}'_n}\tfrac{\delta_n}{n}\varphi(\tfrac{1}{\tilde{\tau}'_n}\tfrac{\lambda_n}{n}) - \tfrac{1}{2}\tfrac{1}{\tilde{\tau}'^2_n}\tfrac{\delta_n^2}{n^2}\varphi'(c_n)$$

for $c_n \in [\frac{1}{\tilde{\tau}'_n}(\frac{\lambda_n}{n} - \frac{\delta_n}{n}), \frac{1}{\tilde{\tau}'_n}\frac{\lambda_n}{n}]$ by a second-order Taylor expansion. We have

$$\frac{1}{\tilde{\tau}'_n}\frac{\delta_n}{n}\varphi(\frac{1}{\tilde{\tau}'_n}\frac{\lambda_n}{n}) = \frac{\delta_n}{\sqrt{n}}\frac{1}{\tilde{\tau}'_n}\frac{\sqrt{n}}{n}\varphi(\frac{1}{\tilde{\tau}'_n}\frac{\lambda_n}{n})$$

whose expectation is $\Theta(\frac{1}{\sqrt{n}})$ since $\lambda_n = O(\sqrt{n})$ and $\delta_n = \Theta(1)$ yield $\frac{\delta_n}{\sqrt{n}} = \Theta(\frac{1}{\sqrt{n}})$ and $\mathbb{E}[\frac{1}{\tilde{\tau}'_n}\frac{\sqrt{n}}{n}\varphi(\frac{1}{\tilde{\tau}'_n}\frac{\lambda_n}{n})] = \Theta(1)$.

As for the second part of the Taylor expansion, its expectation is a $o(\frac{1}{\sqrt{n}})$ since $\varphi'$ is bounded, $\delta_n = \Theta(1)$ and we have

$$\mathbb{E}[\frac{1}{\tilde{\tau}'^2_n}] = \frac{1}{\tau^2}\mathbb{E}[((\mathbf{X'}^\top\mathbf{X'})^{-1}_{i,i})^{-1}] = n - p + 1$$

using the fact that for $X_i \overset{\text{i.i.d.}}{\sim} \mathcal{N}(0, \mathbf{I})$, we know $(\mathbf{X}^\top\mathbf{X})^{-1} \sim W_p^{-1}(\mathbf{I}, n)$ and then the diagonal element $(\mathbf{X}^\top\mathbf{X})^{-1}_{i,i}$ follows an inverse gamma distribution with shape parameter $\frac{n-p+1}{2}$ and scale parameter $\frac{1}{2}$, and the expectation of the reciprocal of an inverse gamma distributed variable is the ratio of the shape and the scale.

We can then conclude that

$$\mathbb{P}(\hat{\beta}'_{\text{OLS},i} \in [\frac{\lambda_n}{n} - \frac{\delta_n}{n}, \frac{\lambda_n}{n}]) = \mathbb{E}[\mathbb{P}(\hat{\beta}'_{\text{OLS},i} \in [\frac{\lambda_n}{n} - \frac{\delta_n}{n}, \frac{\lambda_n}{n}] \mid \mathbf{X'})] = \Theta(\frac{1}{\sqrt{n}})$$

and thus $\mathbb{E}[\nu^2_{n,i}] = \Omega(\frac{\delta^2_n}{n^2\sqrt{n}})$.

We will now prove the third result $\mathbb{E}[\nu^2_{n,i}] = o(\frac{\delta^2_n}{n^2\sqrt{n}})$ for all $i$ such that $\beta^\star_i \neq 0$.

Consider $i$ such that $\beta^\star_i > 0$, since the combination of the first case on the left side and the first case on the right side in the expression of $\nu_{n,i}$ corresponds to a value of 0 for $\nu_{n,i}$, we can write $\nu^2_{n,i}$ as the sum of 24 terms that are of the form: an indicator of one of the 24 other events multiplied by the value of $\nu^2_{n,i}$ for this event. Since $\nu^2_{n,i} \leq 4\frac{\delta^2_n}{n^2}$, we can upper-bound $\nu^2_{n,i}$ by $4\frac{\delta^2_n}{n^2}$ multiplied by the sum of the 24 indicators and we then need to show that all 24 indicators have an expectation which is $o(\frac{1}{\sqrt{n}})$. Using $\mathbb{E}[\mathbb{1}[A]] = \mathbb{P}(A)$, $\mathbb{P}(A \cap B) \leq \min(\mathbb{P}(A), \mathbb{P}(B))$ and the fact that $\hat{\beta}_{\text{OLS}}$ and $\hat{\beta}'_{\text{OLS}}$ have the same unconditional distribution, we can upper-bound all 24 indicator expectations by one of the following four probabilities

- $\mathbb{P}(\hat{\beta}_{\text{OLS},i} < -\frac{\lambda_n+\delta_n}{n}) = \mathbb{E}[\mathbb{P}(\hat{\beta}_{\text{OLS},i} < -\frac{\lambda_n+\delta_n}{n} \mid \mathbf{X})]$,

- $\mathbb{P}(\hat{\beta}_{\text{OLS},i} \in [\frac{\lambda_n}{n}, \frac{\lambda_n+\delta_n}{n}]) = \mathbb{E}[\mathbb{P}(\hat{\beta}_{\text{OLS},i} \in [\frac{\lambda_n}{n}, \frac{\lambda_n+\delta_n}{n}] \mid \mathbf{X})]$,

- $\mathbb{P}(\hat{\beta}_{\text{OLS},i} \in [-\frac{\lambda_n+\delta_n}{n}, -\frac{\lambda_n}{n}]) = \mathbb{E}[\mathbb{P}(\hat{\beta}_{\text{OLS},i} \in [-\frac{\lambda_n+\delta_n}{n}, -\frac{\lambda_n}{n}] \mid \mathbf{X})]$,

- $\mathbb{P}(|\hat{\beta}_{\text{OLS},i}| < \frac{\lambda_n}{n}) = \mathbb{E}[\mathbb{P}(|\hat{\beta}_{\text{OLS},i}| < \frac{\lambda_n}{n} \mid \mathbf{X})]$.

Since $\hat{\beta}_{\text{OLS}} \mid \mathbf{X} \sim \mathcal{N}(\beta^\star, \tau^2(\mathbf{X}^\top\mathbf{X})^{-1})$, we can write $\hat{\beta}_{\text{OLS},i} = \beta^\star_i + \tilde{\tau}_n Z$ where $\tilde{\tau}_n = \frac{\tau}{\sqrt{n}}\sqrt{(\frac{\mathbf{X}^\top\mathbf{X}}{n})^{-1}_{i,i}}$ and $Z \mid \mathbf{X} \sim \mathcal{N}(0, 1)$. Note that we could have $i$ as a subscript of $\tilde{\tau}_n$ and $Z$, but we will only consider one $i$ at a time in our computations and we can thus omit this subscript for both of them for the sake of notational simplicity, and we will also omit it for some additional notation we define in the rest of the proof.

Define $\alpha_n^{(1)} = \frac{1}{\tilde{\tau}_n}(\frac{\lambda_n}{n} - \beta^\star_i)$, $\alpha_n^{(2)} = \frac{1}{\tilde{\tau}_n}(\frac{\lambda_n}{n} + \beta^\star_i)$, $\theta_n^{(1)} = \frac{1}{\tilde{\tau}_n}(\frac{\lambda_n+\delta_n}{n} - \beta^\star_i)$ and $\theta_n^{(2)} = \frac{1}{\tilde{\tau}_n}(\frac{\lambda_n+\delta_n}{n} + \beta^\star_i)$.

In the order they appear, the four conditional probabilities above are equal to

$$\mathbb{P}(Z < -\theta_n^{(2)} \mid \mathbf{X}) = \Phi(-\theta_n^{(2)}),$$

$$\mathbb{P}(Z \in [\alpha_n^{(1)}, \theta_n^{(1)}] \mid \mathbf{X}) = \Phi(\theta_n^{(1)}) - \Phi(\alpha_n^{(1)}),$$

$$\mathbb{P}(Z \in [-\theta_n^{(2)}, -\alpha_n^{(2)}] \mid \mathbf{X}) = \Phi(-\alpha_n^{(2)}) - \Phi(-\theta_n^{(2)}),$$

$$\mathbb{P}(Z \in [-\alpha_n^{(2)}, \alpha_n^{(1)}] \mid \mathbf{X}) = \Phi(\alpha_n^{(1)}) - \Phi(-\alpha_n^{(2)}).$$

Since $\frac{\mathbf{X}^\top\mathbf{X}}{n} \xrightarrow{\text{a.s.}} \mathbb{E}[X_0 X_0^\top]$ (strong law of large numbers), $\lambda_n = o(n)$ and $\delta_n = o(n)$, we have $\tilde{\tau}_n \xrightarrow{\text{a.s.}} 0^+$, and using the continuous mapping theorem, $\alpha_n^{(1)} \xrightarrow{\text{a.s.}} -\infty$, $\theta_n^{(1)} \xrightarrow{\text{a.s.}} -\infty$, $\alpha_n^{(2)} \xrightarrow{\text{a.s.}} +\infty$ and $\theta_n^{(2)} \xrightarrow{\text{a.s.}} +\infty$ as $\beta^\star_i > 0$.

If we show that $\sqrt{n}\,\Phi(\alpha_n^{(1)})$ goes to 0 in $L^1$, then all other similar convergences will follow and we will get that all four unconditional probabilities listed above are $o(\frac{1}{\sqrt{n}})$ and thus $\mathbb{E}[\nu_{n,i}^2] = o(\frac{\delta_n^2}{n^2\sqrt{n}})$ for all $i$ such that $\beta_i^\star > 0$.

We have

$$\sqrt{n}\,\Phi(\alpha_n^{(1)}) = \frac{\sqrt{n}}{\alpha_n^{(1)}} \cdot \alpha_n^{(1)}\Phi(\alpha_n^{(1)})$$

thus, by Cauchy–Schwarz,

$$\mathbb{E}[|\sqrt{n}\,\Phi(\alpha_n^{(1)})|] \leq \sqrt{\mathbb{E}\left[\frac{n}{(\alpha_n^{(1)})^2}\right]\mathbb{E}\left[(\alpha_n^{(1)}\Phi(\alpha_n^{(1)}))^2\right]}.$$

$\alpha_n^{(1)} \xrightarrow{\text{a.s.}} -\infty$ so $\alpha_n^{(1)}\Phi(\alpha_n^{(1)}) \xrightarrow{\text{a.s.}} 0$. This comes from the fact that $x\,\Phi(x) \to 0$ for $x \to -\infty$, as we notice that for $x < 0$, we have $0 < -x\,\Phi(x) = -x\,(1 - \Phi(-x)) = -x \int_{-x}^{+\infty} \varphi(t)dt \leq \int_{-x}^{+\infty} t\varphi(t)dt$ where this last expression goes to 0 when $x \to -\infty$.

Since $\lambda_n = o(n)$ and $\delta_n = o(n)$, for $n$ large enough, $\frac{\lambda_n + \delta_n}{n} < \beta_i^\star$, so $\alpha_n^{(1)} \leq \theta_n^{(1)} < 0$. Since the function $x \mapsto x\,\Phi(x)$ is continuous bounded for $x < 0$, we get $L^1$ convergence of $(\alpha_n^{(1)}\Phi(\alpha_n^{(1)}))^2$ to 0.

Moreover, $\frac{n}{(\alpha_n^{(1)})^2} = \frac{n\tilde{\tau}_n^2}{(\frac{\lambda_n}{n} - \beta_i^\star)^2} = \frac{\tau^2}{(\frac{\lambda_n}{n} - \beta_i^\star)^2}(\frac{\mathbf{X}^\top\mathbf{X}}{n})_{i,i}^{-1}$ and it is thus sufficient to have $\mathbb{E}[(\frac{\mathbf{X}^\top\mathbf{X}}{n})_{i,i}^{-1}] = O(1)$, which is the case for features drawn i.i.d. from $\mathcal{N}(0, \mathbf{I})$ as $\mathbb{E}[(\frac{\mathbf{X}^\top\mathbf{X}}{n})_{i,i}^{-1}] = \frac{n}{n-p-1}$.

Hence, $\sqrt{n}\,\Phi(\alpha_n^{(1)})$ goes to 0 in $L^1$.

The proof is similar for $i$ such that $\beta_i^\star < 0$.

Finally, we show the fourth result $\mathbb{E}[\Psi_{n,i,j}^2] = O(\frac{\delta_n^2}{n^4})$ for all $i, j$ or equivalently $\mathbb{E}[\Psi_{n,i,j}^2] = O(\frac{\delta_n^4}{n^4})$ since $\delta_n = \Theta(1)$.

Similarly to previous computations and upper-bounding with Cauchy–Schwarz, we can upper-bound $\mathbb{E}[\Psi_{n,i,j}^2]$ using products of $\mathbb{E}[\nu_{n,i}^4]$ and the fourth moment of $\hat{\beta}_{\lambda_n,i}$, $\hat{\beta}_{\lambda_n+\delta_n,i}$, $\hat{\beta}'_{\lambda_n,i}$ or $\hat{\beta}'_{\lambda_n+\delta_n,i}$ and their counterparts for $j$. Since $\nu_{n,i}^2 \leq 4\frac{\delta_n^2}{n^2}$, we have $\mathbb{E}[\nu_{n,i}^4] = O(\frac{\delta_n^4}{n^4})$. Additionally, the fourth moments are bounded as we showed the $L^4$ consistency of soft-thresholding for $\beta^\star$. This gives us $\mathbb{E}[\Psi_{n,i,j}^2] = O(\frac{\delta_n^4}{n^4})$.

With the four results proved, we can conclude that $\gamma(h_n^{\text{diff}}) = \Omega(\frac{\delta_n^2}{n^2\sqrt{n}})$.

## G. Proof of Theorem 3.2: Relative stability of ST

Theorem 3.2 follows immediately from the following two propositions, proved in Appendices I and J, respectively.

**Proposition G.1** (Convergence of $\sigma^2(h_n^{\text{sing}})$ for ST$(\lambda_n)$). *Assume the linear model* (2). *If* $\lambda_n = o(n)$, *then* $\sigma^2(h_n^{\text{sing}}) \to 2\tau^4$.

**Proposition G.2** (Convergence rate of $\gamma(h_n^{\text{sing}})$ for ST$(\lambda_n)$). *Assume the linear model* (2). *If* $\lambda_n = o(n)$, *then* $\gamma(h_n^{\text{sing}}) \sim \frac{C}{n^2}$ *for a constant* $C > 0$ *whose explicit expression is given in* (8).

## H. Proof of Lemma 4.1: Convergence of $\sigma^2(h_n^{\text{sing}})$ and $\sigma^2(h_n^{\text{diff}})$

Let $Y_0 = X_0^\top \beta^\star + \varepsilon_0$ be the response variable with $\text{Var}(\varepsilon_0) = \tau^2$. Using the information on the distribution of $X_0$ and the independence of $X_0$ and $\varepsilon_0$, note that

$$\mathbb{E}[Y_0^2] = \text{Var}(Y_0) + \mathbb{E}[Y_0]^2 = \text{Var}(X_0^\top \beta^\star + \varepsilon_0) + 0 = \beta^{\star\top}\text{Var}(X_0)\beta^\star + \text{Var}(\varepsilon_0) = \beta^{\star\top}\beta^\star + \tau^2.$$

For the single linear predictor, starting from the expression of $\sigma^2(h_n)$ in Lemma I.1, since $\mathbb{E}[\hat{\beta}_n]$ and $\mathbb{E}[\hat{\beta}_n\hat{\beta}_n^\top]$ are non-random, we can expand the square, use linearity of expectation, take the limits and factorize back to obtain the convergence

$$
\begin{aligned}
\sigma^2(h_n^{\mathrm{sing}}) &= \mathbb{E}[(Y_0^2 - \mathbb{E}[Y_0^2] - 2(Y_0 X_0^\top - \beta^{\star\top}\mathbb{E}[X_0 X_0^\top])\mathbb{E}[\hat{\beta}_n] + \mathrm{tr}((X_0 X_0^\top - \mathbb{E}[X_0 X_0^\top])\mathbb{E}[\hat{\beta}_n\hat{\beta}_n^\top]))^2] \\
&\to \mathbb{E}[(Y_0^2 - \mathbb{E}[Y_0^2] - 2(Y_0 X_0^\top - \beta^{\star\top}\mathbb{E}[X_0 X_0^\top])\beta^\star + \mathrm{tr}((X_0 X_0^\top - \mathbb{E}[X_0 X_0^\top])\beta^\star\beta^{\star\top}))^2] \\
&= \mathbb{E}[(Y_0^2 - \beta^{\star\top}\beta^\star - \tau^2 - 2Y_0 X_0^\top\beta^\star + 2\beta^{\star\top}\beta^\star + \mathrm{tr}((X_0 X_0^\top\beta^\star\beta^{\star\top} - \beta^\star\beta^{\star\top}))^2] \\
&= \mathbb{E}[((X_0^\top\beta^\star + \varepsilon_0)^2 - \beta^{\star\top}\beta^\star - \tau^2 - 2(X_0^\top\beta^\star + \varepsilon_0)X_0^\top\beta^\star + 2\beta^{\star\top}\beta^\star + (X_0^\top\beta^\star)^2 - \beta^{\star\top}\beta^\star)^2] \\
&= \mathbb{E}[(\varepsilon_0^2 - \tau^2)^2] = \mathrm{Var}(\varepsilon_0^2) = \mathbb{E}[\varepsilon_0^4] - \mathbb{E}[\varepsilon_0^2]^2 = 3\tau^4 - \tau^4 = 2\tau^4.
\end{aligned}
$$

Similarly, we derive the second result with two linear predictors by starting from the expression of $\sigma^2(h_n)$ in Lemma E.1.

# I. Proof of Proposition G.1: Convergence of $\sigma^2(h_n^{\mathrm{sing}})$ for $\mathrm{ST}(\lambda_n)$

We start by introducing a lemma which provides equations that will prove useful in the single algorithm setting.

**Lemma I.1** (Useful equations for single linear predictor). *When defining $h_n(Z_0, \mathbf{Z}) = (Y_0 - X_0^\top\hat{\beta})^2$, we have:*

$$
\begin{aligned}
h_n(Z_0, \mathbf{Z}) &= Y_0^2 - 2Y_0 X_0^\top\hat{\beta} + \mathrm{tr}(X_0 X_0^\top\hat{\beta}\hat{\beta}^\top) \\
\mathbb{E}[h_n(Z_0, \mathbf{Z}) \mid Z_0] &= Y_0^2 - 2Y_0 X_0^\top\mathbb{E}[\hat{\beta}] + \mathrm{tr}(X_0 X_0^\top\mathbb{E}[\hat{\beta}\hat{\beta}^\top]) \\
\mathbb{E}[h_n(Z_0, \mathbf{Z}) \mid \mathbf{Z}] &= \mathbb{E}[Y_0^2] - 2\beta^{\star\top}\mathbb{E}[X_0 X_0^\top]\hat{\beta} + \mathrm{tr}(\mathbb{E}[X_0 X_0^\top]\hat{\beta}\hat{\beta}^\top) \\
\mathbb{E}[h_n(Z_0, \mathbf{Z})] &= \mathbb{E}[Y_0^2] - 2\beta^{\star\top}\mathbb{E}[X_0 X_0^\top]\mathbb{E}[\hat{\beta}] + \mathrm{tr}(\mathbb{E}[X_0 X_0^\top]\mathbb{E}[\hat{\beta}\hat{\beta}^\top]) \\
\sigma^2(h_n) &= \mathbb{E}[(Y_0^2 - \mathbb{E}[Y_0^2] - 2(Y_0 X_0^\top - \beta^{\star\top}\mathbb{E}[X_0 X_0^\top])\mathbb{E}[\hat{\beta}] \\
&\quad + \mathrm{tr}((X_0 X_0^\top - \mathbb{E}[X_0 X_0^\top])\mathbb{E}[\hat{\beta}\hat{\beta}^\top]))^2] \\
\gamma(h_n) &= \mathbb{E}[(2(Y_0 X_0^\top - \beta^{\star\top}\mathbb{E}[X_0 X_0^\top])(\hat{\beta}' - \hat{\beta}) \\
&\quad + \mathrm{tr}((X_0 X_0^\top - \mathbb{E}[X_0 X_0^\top])(\hat{\beta}\hat{\beta}^\top - \hat{\beta}'\hat{\beta}'^\top)))^2]
\end{aligned}
$$

*where $\hat{\beta}'$ is the linear predictor learned on a training set $\mathbf{Z}'$ that is the same as $\mathbf{Z}$ except for the first point $Z_1$ being replaced by an i.i.d copy $Z_1'$.*

**Proof**

$$
\begin{aligned}
h_n(Z_0, \mathbf{Z}) &= (Y_0 - X_0^\top\hat{\beta})^2 \\
&= Y_0^2 - 2Y_0 X_0^\top\hat{\beta} + (X_0^\top\hat{\beta})^2 \\
&= Y_0^2 - 2Y_0 X_0^\top\hat{\beta} + X_0^\top\hat{\beta}\hat{\beta}^\top X_0 \\
&= Y_0^2 - 2Y_0 X_0^\top\hat{\beta} + \mathrm{tr}(X_0 X_0^\top\hat{\beta}\hat{\beta}^\top)
\end{aligned}
$$

Note that $\mathbb{E}[Y_0 X_0^\top] = \mathbb{E}[\mathbb{E}[Y_0 \mid X_0]X_0^\top] = \mathbb{E}[X_0^\top\beta^\star X_0^\top] = \beta^{\star\top}\mathbb{E}[X_0 X_0^\top]$.

Since $\hat{\beta}$ is only a function of $\mathbf{Z}$, the independence of $Z_0$ and $\mathbf{Z}$ yields the next three equations.

The fifth equation comes from noticing

$$
\sigma^2(h_n) = \mathrm{Var}(\mathbb{E}[h_n(Z_0, \mathbf{Z}) \mid Z_0]) = \mathbb{E}[(\mathbb{E}[h_n(Z_0, \mathbf{Z}) \mid Z_0] - \mathbb{E}[h_n(Z_0, \mathbf{Z})])^2].
$$

And the last one comes from the definition of $\gamma(h_n)$ as

$$
\gamma(h_n) = \mathbb{E}[(h(Z_0, \mathbf{Z}) - h(Z_0, \mathbf{Z}') - (\mathbb{E}[h(Z_0, \mathbf{Z}) \mid \mathbf{Z}] - \mathbb{E}[h(Z_0, \mathbf{Z}') \mid \mathbf{Z}']))^2].
$$

$\square$

We will show that $\mathbb{E}[\hat{\beta}_{\lambda_n}] \to \beta^\star$ and $\mathbb{E}[\hat{\beta}_{\lambda_n}\hat{\beta}_{\lambda_n}^\top] \to \beta^\star\beta^{\star\top}$ in order to obtain the convergence of $\sigma^2(h_n^{\mathrm{sing}})$ as an application of Lemma 4.1.

We have for $i = 1, \ldots, p$,

$$
\begin{aligned}
\hat{\beta}_{\lambda_n,i} &= \text{sign}(\hat{\beta}_{\text{OLS},i})(|\hat{\beta}_{\text{OLS},i}| - \tfrac{\lambda_n}{n})_+ \\
&= \text{sign}(\hat{\beta}_{\text{OLS},i})
\begin{cases}
|\hat{\beta}_{\text{OLS},i}| - \tfrac{\lambda_n}{n} & \text{if } |\hat{\beta}_{\text{OLS},i}| \geq \tfrac{\lambda_n}{n} \\
0 & \text{if } |\hat{\beta}_{\text{OLS},i}| < \tfrac{\lambda_n}{n}
\end{cases}.
\end{aligned}
$$

A classic result for the OLS estimator is $\hat{\beta}_{\text{OLS}} \mid \mathbf{X} \sim \mathcal{N}(\beta^\star, \tau^2(\mathbf{X}^\top\mathbf{X})^{-1})$. We can write $\hat{\beta}_{\text{OLS},i} = \beta_i^\star + \tilde{\tau}_n Z$ where $\tilde{\tau}_n = \frac{\tau}{\sqrt{n}}\sqrt{(\frac{\mathbf{X}^\top\mathbf{X}}{n})^{-1}_{i,i}}$ and $Z \mid \mathbf{X} \sim \mathcal{N}(0,1)$. Note that we could have $i$ as a subscript of $\tilde{\tau}_n$ and $Z$, but we will only consider one $i$ at a time in our computations and we can thus omit this subscript for both of them for the sake of notational simplicity, and we will also omit it for some additional notation we define in the rest of the proof.

We now show that $\mathbb{E}[\hat{\beta}_{\lambda_n,i}] \to \beta_i^\star$.

Using the law of total expectation,

$$
\begin{aligned}
\mathbb{E}[\hat{\beta}_{\lambda_n,i} \mid \mathbf{X}] &= \mathbb{E}[\hat{\beta}_{\text{OLS},i} - \tfrac{\lambda_n}{n} \mid \hat{\beta}_{\text{OLS},i} \geq \tfrac{\lambda_n}{n}, \mathbf{X}] \, \mathbb{P}(\hat{\beta}_{\text{OLS},i} \geq \tfrac{\lambda_n}{n} \mid \mathbf{X}) \\
&\quad + \mathbb{E}[\hat{\beta}_{\text{OLS},i} + \tfrac{\lambda_n}{n} \mid \hat{\beta}_{\text{OLS},i} \leq -\tfrac{\lambda_n}{n}, \mathbf{X}] \, \mathbb{P}(\hat{\beta}_{\text{OLS},i} \leq -\tfrac{\lambda_n}{n} \mid \mathbf{X}).
\end{aligned}
$$

Define $\alpha_n^{(1)} = \frac{1}{\tilde{\tau}_n}(\frac{\lambda_n}{n} - \beta_i^\star)$ and $\alpha_n^{(2)} = \frac{1}{\tilde{\tau}_n}(\frac{\lambda_n}{n} + \beta_i^\star)$.

The first probability is equal to

$$
\mathbb{P}(Z \geq \alpha_n^{(1)} \mid \mathbf{X}) = 1 - \Phi(\alpha_n^{(1)})
$$

and the second probability to

$$
\mathbb{P}(Z \leq -\alpha_n^{(2)} \mid \mathbf{X}) = \Phi(-\alpha_n^{(2)}) = 1 - \Phi(\alpha_n^{(2)}).
$$

Using the first moment of the truncated normal (Johnson et al., 1994), we have

$$
\begin{aligned}
\mathbb{E}[\hat{\beta}_{\text{OLS},i} - \tfrac{\lambda_n}{n} \mid \hat{\beta}_{\text{OLS},i} \geq \tfrac{\lambda_n}{n}, \mathbf{X}] &= \beta_i^\star - \tfrac{\lambda_n}{n} + \tilde{\tau}_n \mathbb{E}[Z \mid Z \geq \alpha_n^{(1)}, \mathbf{X}] \\
&= \beta_i^\star - \tfrac{\lambda_n}{n} + \tilde{\tau}_n \frac{\varphi(\alpha_n^{(1)})}{1 - \Phi(\alpha_n^{(1)})}
\end{aligned}
$$

and

$$
\begin{aligned}
\mathbb{E}[\hat{\beta}_{\text{OLS},i} + \tfrac{\lambda_n}{n} \mid \hat{\beta}_{\text{OLS},i} \leq -\tfrac{\lambda_n}{n}, \mathbf{X}] &= \beta_i^\star + \tfrac{\lambda_n}{n} + \tilde{\tau}_n \mathbb{E}[Z \mid Z \leq -\alpha_n^{(2)}] \\
&= \beta_i^\star + \tfrac{\lambda_n}{n} - \tilde{\tau}_n \frac{\varphi(-\alpha_n^{(2)})}{\Phi(-\alpha_n^{(2)})}.
\end{aligned}
$$

Therefore

$$
\begin{aligned}
\mathbb{E}[\hat{\beta}_{\lambda_n,i} \mid \mathbf{X}] &= \mathbb{E}[\hat{\beta}_{\text{OLS},i} - \tfrac{\lambda_n}{n} \mid \hat{\beta}_{\text{OLS},i} \geq \tfrac{\lambda_n}{n}, \mathbf{X}] \, \mathbb{P}(\hat{\beta}_{\text{OLS},i} \geq \tfrac{\lambda_n}{n} \mid \mathbf{X}) \\
&\quad + \mathbb{E}[\hat{\beta}_{\text{OLS},i} + \tfrac{\lambda_n}{n} \mid \hat{\beta}_{\text{OLS},i} \leq -\tfrac{\lambda_n}{n}, \mathbf{X}] \, \mathbb{P}(\hat{\beta}_{\text{OLS},i} \leq -\tfrac{\lambda_n}{n} \mid \mathbf{X}) \\
&= (\beta_i^\star - \tfrac{\lambda_n}{n})(1 - \Phi(\alpha_n^{(1)})) + \tilde{\tau}_n \varphi(\alpha_n^{(1)}) + (\beta_i^\star + \tfrac{\lambda_n}{n})\Phi(-\alpha_n^{(2)}) - \tilde{\tau}_n\varphi(-\alpha_n^{(2)}) \\
&= (\beta_i^\star - \tfrac{\lambda_n}{n})(1 - \Phi(\alpha_n^{(1)})) + (\beta_i^\star + \tfrac{\lambda_n}{n})\Phi(-\alpha_n^{(2)}) + \tilde{\tau}_n(\varphi(\alpha_n^{(1)}) - \varphi(-\alpha_n^{(2)})).
\end{aligned}
$$

Note that $\varphi'(x) = -x\varphi(x)$. A straightforward study of the behavior of the function $x \mapsto x\varphi(x)$ shows it is bounded. We denote the maximum of its absolute value by $M$.

Using the mean value inequality for $\varphi$, we have

$$
\begin{aligned}
|\tilde{\tau}_n(\varphi(\alpha_n^{(1)}) - \varphi(-\alpha_n^{(2)}))| &\leq \tilde{\tau}_n|\alpha_n^{(1)} - (-\alpha_n^{(2)})| \cdot \max_{[-\alpha_n^{(2)}, \alpha_n^{(1)}]}|\varphi'| \\
&\leq M\tilde{\tau}_n|\alpha_n^{(1)} - (-\alpha_n^{(2)})| \\
&= M\tilde{\tau}_n \tfrac{1}{\tilde{\tau}_n}(\tfrac{\lambda_n}{n} - \beta_i^\star + \tfrac{\lambda_n}{n} + \beta_i^\star) \\
&= 2M\tfrac{\lambda_n}{n}.
\end{aligned}
$$

Therefore, since $\lambda_n = o(n)$, $\tilde{\tau}_n(\varphi(\alpha_n^{(1)}) - \varphi(-\alpha_n^{(2)}))$ goes to 0 in $L^1$.

We first consider $\beta_i^\star > 0$.

Since $\frac{\mathbf{X}^\top \mathbf{X}}{n} \xrightarrow{\text{a.s.}} \mathbb{E}[X_0 X_0^\top]$ (strong law of large numbers), and $\lambda_n = o(n)$, we have $\tilde{\tau}_n \xrightarrow{\text{a.s.}} 0^+$, and using the continuous mapping theorem, $\alpha_n^{(1)} \xrightarrow{\text{a.s.}} -\infty$ and $\alpha_n^{(2)} \xrightarrow{\text{a.s.}} +\infty$. $\Phi$ is continuous bounded so we get $L^1$ convergence of $\Phi(\alpha_n^{(1)})$ and $\Phi(-\alpha_n^{(2)})$ to 0. By putting everything together, we obtain

$$\mathbb{E}[\hat{\beta}_{\lambda_n,i}] = \mathbb{E}[\mathbb{E}[\hat{\beta}_{\lambda_n,i} \mid \mathbf{X}]] \to \beta_i^\star.$$

When $\beta_i^\star < 0$, we show in a similar manner that $\mathbb{E}[\hat{\beta}_{\lambda_n,i}] \to \beta_i^\star$.

If $\beta_i^\star = 0$, $\alpha_n^{(1)} = \alpha_n^{(2)}$ so $1 - \Phi(\alpha_n^{(1)}) = \Phi(-\alpha_n^{(2)})$ and $\varphi(\alpha_n^{(1)}) = \varphi(-\alpha_n^{(2)})$ which leads to $\mathbb{E}[\hat{\beta}_{\lambda_n,i} \mid \mathbf{X}] = 0$ and thus $\mathbb{E}[\hat{\beta}_{\lambda_n,i}] = 0$.

Thus, we have convergence component-wise and can conclude $\mathbb{E}[\hat{\beta}_{\lambda_n}] \to \beta^\star$.

We now show that $\mathbb{E}[\hat{\beta}_{\lambda_n,i}\hat{\beta}_{\lambda_n,j}] \to \beta_i^\star\beta_j^\star$.

Note that

$$\mathbb{E}[\hat{\beta}_{\lambda_n,i}\hat{\beta}_{\lambda_n,j} - \beta_i^\star\beta_j^\star] = \mathbb{E}[(\hat{\beta}_{\lambda_n,i} - \beta_i^\star)\hat{\beta}_{\lambda_n,j}] + \beta_i^\star\mathbb{E}[\hat{\beta}_{\lambda_n,j} - \beta_j^\star]$$

where, using Cauchy–Schwarz and the fact that $(a+b)^2 \le 2(a^2 + b^2)$,

$$|\mathbb{E}[(\hat{\beta}_{\lambda_n,i} - \beta_i^\star)\hat{\beta}_{\lambda_n,j}]| \le \sqrt{\mathbb{E}[(\hat{\beta}_{\lambda_n,i} - \beta_i^\star)^2]\mathbb{E}[\hat{\beta}_{\lambda_n,j}^2]} \le \sqrt{\mathbb{E}[(\hat{\beta}_{\lambda_n,i} - \beta_i^\star)^2]2(\mathbb{E}[(\hat{\beta}_{\lambda_n,j} - \beta_j^\star)^2] + \beta_j^{\star 2})}.$$

Therefore, proving $\mathbb{E}[\hat{\beta}_{\lambda_n,i}\hat{\beta}_{\lambda_n,j}] \to \beta_i^\star\beta_j^\star$ for all $i, j$ comes down to proving $\mathbb{E}[(\hat{\beta}_{\lambda_n,i} - \beta_i^\star)^2] \to 0$ for all $i$ given that we have already shown $\mathbb{E}[\hat{\beta}_{\lambda_n,i}] \to \beta_i^\star$ for all $i$.

As a reminder, we have

$$\hat{\beta}_{\lambda_n,i} = \text{sign}(\hat{\beta}_{\text{OLS},i})\begin{cases} |\hat{\beta}_{\text{OLS},i}| - \frac{\lambda_n}{n} & \text{if } |\hat{\beta}_{\text{OLS},i}| \ge \frac{\lambda_n}{n} \\ 0 & \text{if } |\hat{\beta}_{\text{OLS},i}| < \frac{\lambda_n}{n} \end{cases}$$

thus

$$\mathbb{E}[(\hat{\beta}_{\lambda_n,i} - \beta_i^\star)^2 \mid \mathbf{X}]$$
$$= \mathbb{E}[(\hat{\beta}_{\text{OLS},i} - \beta_i^\star - \frac{\lambda_n}{n})^2 \mid \hat{\beta}_{\text{OLS},i} \ge \frac{\lambda_n}{n}, \mathbf{X}]\,\mathbb{P}(\hat{\beta}_{\text{OLS},i} \ge \frac{\lambda_n}{n} \mid \mathbf{X})$$
$$+ \mathbb{E}[(\hat{\beta}_{\text{OLS},i} - \beta_i^\star + \frac{\lambda_n}{n})^2 \mid \hat{\beta}_{\text{OLS},i} \le -\frac{\lambda_n}{n}, \mathbf{X}]\,\mathbb{P}(\hat{\beta}_{\text{OLS},i} \le -\frac{\lambda_n}{n} \mid \mathbf{X}).$$

We introduce an intermediate lemma where we derive the second and fourth moments of the truncated normal. The second moment we will use right now and the fourth moment will be used later.

**Lemma I.2** (Moments of the truncated normal). *Let $X \sim \mathcal{N}(0,1)$, and $m_k = \mathbb{E}[X^k \mid a \le X \le b]$ for $k \in \mathbb{N}$, where $-\infty \le a < b \le \infty$. Then $m_2 = 1 + \frac{a\varphi(a) - b\varphi(b)}{\Phi(b) - \Phi(a)}$ and $m_4 = 3 + \frac{(a^3 + 3a)\varphi(a) - (b^3 + 3b)\varphi(b)}{\Phi(b) - \Phi(a)}$.*

**Proof** First, we can derive a recursive formula on the $m_k$'s using integration by parts with the fact that $\varphi'(x) = -x\varphi(x)$. For $k \in \mathbb{N}$, we have

$$m_{k+2} = \int_a^b \frac{x^{k+2}\varphi(x)}{\Phi(b) - \Phi(a)} dx = [\frac{-x^{k+1}\varphi(x)}{\Phi(b) - \Phi(a)}]_a^b + (k+1)\int_a^b \frac{x^k\varphi(x)}{\Phi(b) - \Phi(a)} dx$$
$$= \frac{a^{k+1}\varphi(a) - b^{k+1}\varphi(b)}{\Phi(b) - \Phi(a)} + (k+1)m_k.$$

Since $m_0 = \int_a^b \frac{\varphi(x)}{\Phi(b) - \Phi(a)} dx = 1$, we immediately obtain $m_2 = 1 + \frac{a\varphi(a) - b\varphi(b)}{\Phi(b) - \Phi(a)}$. And consequently, we have

$$m_4 = \frac{a^3\varphi(a) - b^3\varphi(b)}{\Phi(b) - \Phi(a)} + 3m_2 = \frac{a^3\varphi(a) - b^3\varphi(b)}{\Phi(b) - \Phi(a)} + 3(1 + \frac{a\varphi(a) - b\varphi(b)}{\Phi(b) - \Phi(a)})$$
$$= 3 + \frac{(a^3 + 3a)\varphi(a) - (b^3 + 3b)\varphi(b)}{\Phi(b) - \Phi(a)}.$$

$\square$

Using the second moment derived in Lemma I.2, we have

$$\mathbb{E}[(\hat{\beta}_{\mathrm{OLS},i} - \beta_i^\star - \tfrac{\lambda_n}{n})^2 \mid \hat{\beta}_{\mathrm{OLS},i} \geq \tfrac{\lambda_n}{n}, \mathbf{X}]$$
$$= \mathbb{E}[(\tilde{\tau}_n Z - \tfrac{\lambda_n}{n})^2 \mid Z \geq \alpha_n^{(1)}, \mathbf{X}]$$
$$= \tilde{\tau}_n^2 \mathbb{E}[Z^2 \mid Z \geq \alpha_n^{(1)}, \mathbf{X}] - 2\tilde{\tau}_n \tfrac{\lambda_n}{n} \mathbb{E}[Z \mid Z \geq \alpha_n^{(1)}, \mathbf{X}] + \tfrac{\lambda_n^2}{n^2}$$
$$= \tilde{\tau}_n^2 (1 + \tfrac{\alpha_n^{(1)}\varphi(\alpha_n^{(1)})}{1-\Phi(\alpha_n^{(1)})}) - 2\tilde{\tau}_n \tfrac{\lambda_n}{n} \tfrac{\varphi(\alpha_n^{(1)})}{1-\Phi(\alpha_n^{(1)})} + \tfrac{\lambda_n^2}{n^2}$$

and

$$\mathbb{E}[(\hat{\beta}_{\mathrm{OLS},i} - \beta_i^\star + \tfrac{\lambda_n}{n})^2 \mid \hat{\beta}_{\mathrm{OLS},i} \leq -\tfrac{\lambda_n}{n}, \mathbf{X}]$$
$$= \tilde{\tau}_n^2 \mathbb{E}[Z^2 \mid Z \leq -\alpha_n^{(2)}, \mathbf{X}] + 2\tilde{\tau}_n \tfrac{\lambda_n}{n} \mathbb{E}[Z \mid Z \leq -\alpha_n^{(2)}, \mathbf{X}] + \tfrac{\lambda_n^2}{n^2}$$
$$= \tilde{\tau}_n^2 (1 + \tfrac{\alpha_n^{(2)}\varphi(-\alpha_n^{(2)})}{\Phi(-\alpha_n^{(2)})}) - 2\tilde{\tau}_n \tfrac{\lambda_n}{n} \tfrac{\varphi(-\alpha_n^{(2)})}{\Phi(-\alpha_n^{(2)})} + \tfrac{\lambda_n^2}{n^2}.$$

Thus

$$\mathbb{E}[(\hat{\beta}_{\lambda_n,i} - \beta_i^\star)^2 \mid \mathbf{X}]$$
$$= \mathbb{E}[(\hat{\beta}_{\mathrm{OLS},i} - \beta_i^\star - \tfrac{\lambda_n}{n})^2 \mid \hat{\beta}_{\mathrm{OLS},i} \geq \tfrac{\lambda_n}{n}, \mathbf{X}] \, \mathbb{P}(\hat{\beta}_{\mathrm{OLS},i} \geq \tfrac{\lambda_n}{n} \mid \mathbf{X})$$
$$\quad + \mathbb{E}[(\hat{\beta}_{\mathrm{OLS},i} - \beta_i^\star + \tfrac{\lambda_n}{n})^2 \mid \hat{\beta}_{\mathrm{OLS},i} \leq -\tfrac{\lambda_n}{n}, \mathbf{X}] \, \mathbb{P}(\hat{\beta}_{\mathrm{OLS},i} \leq -\tfrac{\lambda_n}{n} \mid \mathbf{X})$$
$$= \tilde{\tau}_n^2 (1 - \Phi(\alpha_n^{(1)}) + \alpha_n^{(1)}\varphi(\alpha_n^{(1)})) - 2\tilde{\tau}_n \tfrac{\lambda_n}{n} \varphi(\alpha_n^{(1)}) + \tfrac{\lambda_n^2}{n^2}(1 - \Phi(\alpha_n^{(1)}))$$
$$\quad + \tilde{\tau}_n^2 (\Phi(-\alpha_n^{(2)}) + \alpha_n^{(2)}\varphi(-\alpha_n^{(2)})) - 2\tilde{\tau}_n \tfrac{\lambda_n}{n} \varphi(-\alpha_n^{(2)}) + \tfrac{\lambda_n^2}{n^2}\Phi(-\alpha_n^{(2)}).$$

For $X_i \overset{\text{i.i.d.}}{\sim} \mathcal{N}(0, \mathbf{I})$, we know $(\mathbf{X}^\top \mathbf{X})^{-1} \sim W_p^{-1}(\mathbf{I}, n)$, therefore $\mathbb{E}[(\mathbf{X}^\top \mathbf{X})^{-1}] = \tfrac{\mathbf{I}}{n-p-1}$ and $\mathbb{E}[(\tfrac{\mathbf{X}^\top \mathbf{X}}{n})_{i,i}^{-1}] = \tfrac{n}{n-p-1} = o(n)$.

Thus, using Jensen's inequality, $\mathbb{E}[\sqrt{(\tfrac{\mathbf{X}^\top \mathbf{X}}{n})_{i,i}^{-1}}] \leq \sqrt{\mathbb{E}[(\tfrac{\mathbf{X}^\top \mathbf{X}}{n})_{i,i}^{-1}]} = \sqrt{\tfrac{n}{n-p-1}} = o(\sqrt{n})$.

As a reminder, $\tilde{\tau}_n = \tfrac{\tau}{\sqrt{n}}\sqrt{(\tfrac{\mathbf{X}^\top \mathbf{X}}{n})_{i,i}^{-1}}$. We then have $L^1$ convergence of both $\tilde{\tau}_n$ and $\tilde{\tau}_n^2$ to 0. As previously mentioned, the function $x \mapsto x\varphi(x)$ is bounded. Since $\Phi$ and $\varphi$ are also bounded, and $\lambda_n = o(n)$, then

$$\mathbb{E}[(\hat{\beta}_{\lambda_n,i} - \beta_i^\star)^2] = \mathbb{E}[\mathbb{E}[(\hat{\beta}_{\lambda_n,i} - \beta_i^\star)^2 \mid \mathbf{X}]] \to 0.$$

Therefore, we get

$$\mathbb{E}[\hat{\beta}_{\lambda_n} \hat{\beta}_{\lambda_n}^\top] \to \beta^\star \beta^{\star\top}.$$

We can then conclude that $\sigma^2(h_n^{\mathrm{sing}}) \to 2\tau^4$ by Lemma 4.1.

## J. Proof of Proposition G.2: Convergence rate of $\gamma(h_n^{\mathrm{sing}})$ for $\mathrm{ST}(\lambda_n)$

As a reminder, to study the loss stability, we consider $Z_1' = (X_1', Y_1')$ an i.i.d. copy of $Z_1 = (X_1, Y_1)$ used as replacement for the first point of the training set.

Define the vector $V \triangleq (Y_1' - X_1'^\top \beta^\star)X_1' - (Y_1 - X_1^\top \beta^\star)X_1$ and the symmetric matrix $M \triangleq -(V\beta^{\star\top} + \beta^\star V^\top)$.

Starting from the expression of $\gamma(h_n)$ in Lemma I.1 and using the fact that $X_0 \sim \mathcal{N}(0, \mathbf{I})$, we have

$$\gamma(h_n^{\mathrm{sing}}) = \mathbb{E}[(2(Y_0 X_0^\top - \beta^{\star\top})(\hat{\beta}_{\lambda_n}' - \hat{\beta}_{\lambda_n}) + \mathrm{tr}((X_0 X_0^\top - \mathbf{I})(\hat{\beta}_{\lambda_n}\hat{\beta}_{\lambda_n}^\top - \hat{\beta}_{\lambda_n}'\hat{\beta}_{\lambda_n}'^\top)))^2].$$

We will show that

$$\gamma(h_n^{\text{sing}}) \sim \tfrac{1}{n^2}\mathbb{E}[(2(Y_0 X_0^\top - \beta^{\star\top})V + \operatorname{tr}((X_0 X_0^\top - \mathbf{I})M))^2].$$

by proving that the difference

$$W_n \triangleq (2(Y_0 X_0^\top - \beta^{\star\top})(\hat{\beta}'_{\lambda_n} - \hat{\beta}_{\lambda_n}) + \operatorname{tr}((X_0 X_0^\top - \mathbf{I})(\hat{\beta}_{\lambda_n}\hat{\beta}_{\lambda_n}^\top - \hat{\beta}'_{\lambda_n}\hat{\beta}'^\top_{\lambda_n})))^2$$
$$- (2(Y_0 X_0^\top - \beta^{\star\top})\tfrac{V}{n} + \operatorname{tr}((X_0 X_0^\top - \mathbf{I})\tfrac{M}{n}))^2$$

goes to $0$ in $L^1$.

Since $a^2 - b^2 = (a - b)(a + b)$, we have

$$W_n = (D_{n,1} + D_{n,2})(S_{n,1} + S_{n,2}).$$

where

$$
\begin{aligned}
D_{n,1} &\triangleq 2(Y_0 X_0^\top - \beta^{\star\top})(\hat{\beta}'_{\lambda_n} - \hat{\beta}_{\lambda_n} - \tfrac{V}{n}), \\
D_{n,2} &\triangleq \operatorname{tr}((X_0 X_0^\top - \mathbf{I})(\hat{\beta}_{\lambda_n}\hat{\beta}_{\lambda_n}^\top - \hat{\beta}'_{\lambda_n}\hat{\beta}'^\top_{\lambda_n} - \tfrac{M}{n})), \\
S_{n,1} &\triangleq 2(Y_0 X_0^\top - \beta^{\star\top})(\hat{\beta}'_{\lambda_n} - \hat{\beta}_{\lambda_n} + \tfrac{V}{n}), \\
S_{n,2} &\triangleq \operatorname{tr}((X_0 X_0^\top - \mathbf{I})(\hat{\beta}_{\lambda_n}\hat{\beta}_{\lambda_n}^\top - \hat{\beta}'_{\lambda_n}\hat{\beta}'^\top_{\lambda_n} + \tfrac{M}{n})).
\end{aligned}
$$

Using Cauchy–Schwarz and the fact that $(a + b)^2 \leq 2(a^2 + b^2)$,

$$\mathbb{E}[|W_n|] \leq \sqrt{\mathbb{E}[(D_{n,1} + D_{n,2})^2]\,\mathbb{E}[(S_{n,1} + S_{n,2})^2]}$$
$$\leq 2\sqrt{\mathbb{E}[D_{n,1}^2 + D_{n,2}^2]\,\mathbb{E}[S_{n,1}^2 + S_{n,2}^2]}.$$

To obtain convergence of $W_n$ to $0$ in $L^1$, we will thus prove that $\mathbb{E}[D_{n,1}^2] \to 0$, $\mathbb{E}[S_{n,1}^2] = O(1)$, $\mathbb{E}[D_{n,2}^2] \to 0$ and $\mathbb{E}[S_{n,2}^2] = O(1)$.

We have

$$
\begin{aligned}
\mathbb{E}[D_{n,1}^2] &= \mathbb{E}[4(Y_0 X_0^\top - \beta^{\star\top})(\hat{\beta}'_{\lambda_n} - \hat{\beta}_{\lambda_n} - \tfrac{V}{n})(\hat{\beta}'_{\lambda_n} - \hat{\beta}_{\lambda_n} - \tfrac{V}{n})^\top(Y_0 X_0 - \beta^\star)] \\
&= \mathbb{E}[4\operatorname{tr}((Y_0 X_0^\top - \beta^{\star\top})(\hat{\beta}'_{\lambda_n} - \hat{\beta}_{\lambda_n} - \tfrac{V}{n})(\hat{\beta}'_{\lambda_n} - \hat{\beta}_{\lambda_n} - \tfrac{V}{n})^\top(Y_0 X_0 - \beta^\star))] \\
&= \mathbb{E}[4\operatorname{tr}((Y_0 X_0 - \beta^\star)(Y_0 X_0^\top - \beta^{\star\top})(\hat{\beta}'_{\lambda_n} - \hat{\beta}_{\lambda_n} - \tfrac{V}{n})(\hat{\beta}'_{\lambda_n} - \hat{\beta}_{\lambda_n} - \tfrac{V}{n})^\top)] \\
&= 4\operatorname{tr}(\mathbb{E}[(Y_0 X_0 - \beta^\star)(Y_0 X_0^\top - \beta^{\star\top})(\hat{\beta}'_{\lambda_n} - \hat{\beta}_{\lambda_n} - \tfrac{V}{n})(\hat{\beta}'_{\lambda_n} - \hat{\beta}_{\lambda_n} - \tfrac{V}{n})^\top]) \\
&= 4\operatorname{tr}(\mathbb{E}[(Y_0 X_0 - \beta^\star)(Y_0 X_0^\top - \beta^{\star\top})]\mathbb{E}[(\hat{\beta}'_{\lambda_n} - \hat{\beta}_{\lambda_n} - \tfrac{V}{n})(\hat{\beta}'_{\lambda_n} - \hat{\beta}_{\lambda_n} - \tfrac{V}{n})^\top])
\end{aligned}
$$

as $\hat{\beta}'_{\lambda_n} - \hat{\beta}_{\lambda_n} - \tfrac{V}{n}$ is a function of the training points and using independence of $Z_0$ from the training points.

By Cauchy–Schwarz, for all $i, j$,

$$\mathbb{E}[|(\hat{\beta}'_{\lambda_n,i} - \hat{\beta}_{\lambda_n,i} - \tfrac{V_i}{n})(\hat{\beta}'_{\lambda_n,j} - \hat{\beta}_{\lambda_n,j} - \tfrac{V_j}{n})|] \leq \sqrt{\mathbb{E}[(\hat{\beta}'_{\lambda_n,i} - \hat{\beta}_{\lambda_n,i} - \tfrac{V_i}{n})^2]\mathbb{E}[(\hat{\beta}'_{\lambda_n,j} - \hat{\beta}_{\lambda_n,j} - \tfrac{V_j}{n})^2]}$$

thus, if we show $\mathbb{E}[(\hat{\beta}'_{\lambda_n,i} - \hat{\beta}_{\lambda_n,i} - \tfrac{V_i}{n})^2] \to 0$ for all $i$, then we obtain

$$\mathbb{E}[(\hat{\beta}'_{\lambda_n} - \hat{\beta}_{\lambda_n} - \tfrac{V}{n})(\hat{\beta}'_{\lambda_n} - \hat{\beta}_{\lambda_n} - \tfrac{V}{n})^\top] \to 0$$

and therefore $\mathbb{E}[D_{n,1}^2] \to 0$. We are going to hold off on proving $\mathbb{E}[(\hat{\beta}'_{\lambda_n,i} - \hat{\beta}_{\lambda_n,i} - \tfrac{V_i}{n})^2] \to 0$ as we will actually show the stronger convergence $\mathbb{E}[(\hat{\beta}'_{\lambda_n,i} - \hat{\beta}_{\lambda_n,i} - \tfrac{V_i}{n})^4] \to 0$ in the context of proving $\mathbb{E}[D_{n,2}^2] \to 0$.

With similar computations and upper-bounding, we can show that $\mathbb{E}[S_{n,1}^2] = O(1)$ if we prove that for all $i$, $\mathbb{E}[(\hat{\beta}'_{\lambda_n,i} - \hat{\beta}_{\lambda_n,i} + \tfrac{V_i}{n})^2] = O(1)$.

As we have shown in Appendix I that the soft-thresholding Lasso estimator is consistent for $\beta^\star$ in $L^2$ when $\lambda_n = o(n)$, both $\mathbb{E}[\hat{\beta}_{\lambda_n,i}^2]$ and $\mathbb{E}[\hat{\beta}_{\lambda_n,i}'^2]$ are bounded and thus $\mathbb{E}[(\hat{\beta}_{\lambda_n,i}' - \hat{\beta}_{\lambda_n,i} + \frac{V_i}{n})^2] = O(1)$ since $(\hat{\beta}_{\lambda_n,i}' - \hat{\beta}_{\lambda_n,i} + \frac{V_i}{n})^2 \leq 3(\hat{\beta}_{\lambda_n,i}'^2 + \hat{\beta}_{\lambda_n,i}^2 + \frac{V_i^2}{n^2})$ by Cauchy–Schwarz.

We now focus on proving $\mathbb{E}[D_{n,2}^2] \to 0$.

We have

$$
\begin{aligned}
D_{n,2} &= \operatorname{tr}((X_0 X_0^\top - \mathbf{I})(\hat{\beta}_{\lambda_n}\hat{\beta}_{\lambda_n}^\top - \hat{\beta}_{\lambda_n}'\hat{\beta}_{\lambda_n}'^\top - \tfrac{M}{n})) \\
&= X_0^\top(\hat{\beta}_{\lambda_n}\hat{\beta}_{\lambda_n}^\top - \hat{\beta}_{\lambda_n}'\hat{\beta}_{\lambda_n}'^\top - \tfrac{M}{n})X_0 - \operatorname{tr}(\hat{\beta}_{\lambda_n}\hat{\beta}_{\lambda_n}^\top - \hat{\beta}_{\lambda_n}'\hat{\beta}_{\lambda_n}'^\top - \tfrac{M}{n}) \\
&= \sum_{i,j}(X_{0,i}X_{0,j} - \mathbb{1}[i=j])(\hat{\beta}_{\lambda_n,i}\hat{\beta}_{\lambda_n,j} - \hat{\beta}_{\lambda_n,i}'\hat{\beta}_{\lambda_n,j}' - \tfrac{M_{i,j}}{n}) \\
&= \sum_{i,j} U_{i,j}(\hat{\beta}_{\lambda_n,i}\hat{\beta}_{\lambda_n,j} - \hat{\beta}_{\lambda_n,i}'\hat{\beta}_{\lambda_n,j}' - \tfrac{M_{i,j}}{n})
\end{aligned}
$$

where $U_{i,j} \triangleq X_{0,i}X_{0,j} - \mathbb{1}[i=j]$, and thus

$$
D_{n,2}^2 = \sum_{i,j,k,l} U_{i,j}U_{k,l}(\hat{\beta}_{\lambda_n,i}\hat{\beta}_{\lambda_n,j} - \hat{\beta}_{\lambda_n,i}'\hat{\beta}_{\lambda_n,j}' - \tfrac{M_{i,j}}{n})(\hat{\beta}_{\lambda_n,k}\hat{\beta}_{\lambda_n,l} - \hat{\beta}_{\lambda_n,k}'\hat{\beta}_{\lambda_n,l}' - \tfrac{M_{k,l}}{n}).
$$

Using independence of $Z_0$ and the training points, we have

$$
\mathbb{E}[D_{n,2}^2] = \sum_{i,j,k,l} \mathbb{E}[U_{i,j}U_{k,l}]\mathbb{E}[(\hat{\beta}_{\lambda_n,i}\hat{\beta}_{\lambda_n,j} - \hat{\beta}_{\lambda_n,i}'\hat{\beta}_{\lambda_n,j}' - \tfrac{M_{i,j}}{n})(\hat{\beta}_{\lambda_n,k}\hat{\beta}_{\lambda_n,l} - \hat{\beta}_{\lambda_n,k}'\hat{\beta}_{\lambda_n,l}' - \tfrac{M_{k,l}}{n})]
$$

where, using Cauchy–Schwarz,

$$
\begin{aligned}
&\mathbb{E}[|(\hat{\beta}_{\lambda_n,i}\hat{\beta}_{\lambda_n,j} - \hat{\beta}_{\lambda_n,i}'\hat{\beta}_{\lambda_n,j}' - \tfrac{M_{i,j}}{n})(\hat{\beta}_{\lambda_n,k}\hat{\beta}_{\lambda_n,l} - \hat{\beta}_{\lambda_n,k}'\hat{\beta}_{\lambda_n,l}' - \tfrac{M_{k,l}}{n})|] \\
&\leq \sqrt{\mathbb{E}[(\hat{\beta}_{\lambda_n,i}\hat{\beta}_{\lambda_n,j} - \hat{\beta}_{\lambda_n,i}'\hat{\beta}_{\lambda_n,j}' - \tfrac{M_{i,j}}{n})^2]\mathbb{E}[(\hat{\beta}_{\lambda_n,k}\hat{\beta}_{\lambda_n,l} - \hat{\beta}_{\lambda_n,k}'\hat{\beta}_{\lambda_n,l}' - \tfrac{M_{k,l}}{n})^2]}.
\end{aligned}
$$

We thus want to show $\mathbb{E}[(\hat{\beta}_{\lambda_n,i}\hat{\beta}_{\lambda_n,j} - \hat{\beta}_{\lambda_n,i}'\hat{\beta}_{\lambda_n,j}' - \tfrac{M_{i,j}}{n})^2] \to 0$ for all $i,j$.

Since $M = -(V\beta^{\star\top} + \beta^\star V^\top)$, we have $M_{i,j} = -V_i\beta_j^\star - \beta_i^\star V_j$ and then

$$
\begin{aligned}
&\hat{\beta}_{\lambda_n,i}\hat{\beta}_{\lambda_n,j} - \hat{\beta}_{\lambda_n,i}'\hat{\beta}_{\lambda_n,j}' - \tfrac{M_{i,j}}{n} \\
&= \hat{\beta}_{\lambda_n,i}\hat{\beta}_{\lambda_n,j} - \hat{\beta}_{\lambda_n,i}'\hat{\beta}_{\lambda_n,j}' + \tfrac{V_i}{n}\beta_j^\star + \beta_i^\star\tfrac{V_j}{n} \\
&= -(\hat{\beta}_{\lambda_n,i}' - \hat{\beta}_{\lambda_n,i} - \tfrac{V_i}{n})\hat{\beta}_{\lambda_n,j} - \hat{\beta}_{\lambda_n,i}'(\hat{\beta}_{\lambda_n,j}' - \hat{\beta}_{\lambda_n,j} - \tfrac{V_j}{n}) - \tfrac{V_i}{n}(\hat{\beta}_{\lambda_n,j} - \beta_j^\star) - (\hat{\beta}_{\lambda_n,i}' - \beta_i^\star)\tfrac{V_j}{n}.
\end{aligned}
$$

By Cauchy–Schwarz,

$$
\begin{aligned}
&(\hat{\beta}_{\lambda_n,i}\hat{\beta}_{\lambda_n,j} - \hat{\beta}_{\lambda_n,i}'\hat{\beta}_{\lambda_n,j}' - \tfrac{M_{i,j}}{n})^2 \\
&= ((\hat{\beta}_{\lambda_n,i}' - \hat{\beta}_{\lambda_n,i} - \tfrac{V_i}{n})\hat{\beta}_{\lambda_n,j} + \hat{\beta}_{\lambda_n,i}'(\hat{\beta}_{\lambda_n,j}' - \hat{\beta}_{\lambda_n,j} - \tfrac{V_j}{n}) + \tfrac{V_i}{n}(\hat{\beta}_{\lambda_n,j} - \beta_j^\star) + (\hat{\beta}_{\lambda_n,i}' - \beta_i^\star)\tfrac{V_j}{n})^2 \\
&\leq 4((\hat{\beta}_{\lambda_n,i}' - \hat{\beta}_{\lambda_n,i} - \tfrac{V_i}{n})^2\hat{\beta}_{\lambda_n,j}^2 + \hat{\beta}_{\lambda_n,i}'^2(\hat{\beta}_{\lambda_n,j}' - \hat{\beta}_{\lambda_n,j} - \tfrac{V_j}{n})^2 + \tfrac{V_i^2}{n^2}(\hat{\beta}_{\lambda_n,j} - \beta_j^\star)^2 + (\hat{\beta}_{\lambda_n,i}' - \beta_i^\star)^2\tfrac{V_j^2}{n^2})
\end{aligned}
$$

and the probability version of Cauchy–Schwarz yields

$$
\begin{aligned}
&\mathbb{E}[(\hat{\beta}_{\lambda_n,i}\hat{\beta}_{\lambda_n,j} - \hat{\beta}_{\lambda_n,i}'\hat{\beta}_{\lambda_n,j}' - \tfrac{M_{i,j}}{n})^2] \\
&\leq 4(\sqrt{\mathbb{E}[(\hat{\beta}_{\lambda_n,i}' - \hat{\beta}_{\lambda_n,i} - \tfrac{V_i}{n})^4]\mathbb{E}[\hat{\beta}_{\lambda_n,j}^4]} + \sqrt{\mathbb{E}[\hat{\beta}_{\lambda_n,i}'^4]\mathbb{E}[(\hat{\beta}_{\lambda_n,j}' - \hat{\beta}_{\lambda_n,j} - \tfrac{V_j}{n})^4]} \\
&\quad + \sqrt{\tfrac{\mathbb{E}[V_i^4]}{n^4}\mathbb{E}[(\hat{\beta}_{\lambda_n,j} - \beta_j^\star)^4]} + \sqrt{\mathbb{E}[(\hat{\beta}_{\lambda_n,i}' - \beta_i^\star)^4]\tfrac{\mathbb{E}[V_j^4]}{n^4}}).
\end{aligned}
$$

Hence, we will get $\mathbb{E}[D_{n,2}^2] \to 0$ if we prove that for all $i$

- $\mathbb{E}[(\hat{\beta}_{\lambda_n,i} - \beta_i^\star)^4] \to 0$, the proof will be the same for $\mathbb{E}[(\hat{\beta}_{\lambda_n,i}' - \beta_i^\star)^4] \to 0$,

- $\mathbb{E}[(\hat{\beta}'_{\lambda_n,i} - \hat{\beta}_{\lambda_n,i} - \frac{V_i}{n})^4] \to 0$.

Note that we will automatically get $L^2$ convergence of $\hat{\beta}'_{\lambda_n,i} - \hat{\beta}_{\lambda_n,i} - \frac{V_i}{n}$ to 0 for all $i$, which implies $\mathbb{E}[D^2_{n,1}] \to 0$ as mentioned earlier.

We now introduce a lemma that will allow us to upper-bound quantities of interest.

**Lemma J.1** (Hölder corollary). *For integers $k, \ell \geq 2$, for $(a_1, \ldots, a_k) \in \mathbb{R}^k$, we have the following inequality*

$$(\textstyle\sum_{i=1}^k |a_i|)^\ell \leq k^{\ell-1} \sum_{i=1}^k |a_i|^\ell.$$

**Proof**  For $(x_1, \ldots, x_k), (y_1, \ldots, y_k) \in \mathbb{R}^k$ and $p, q \in (1, +\infty)$ such that $\frac{1}{p} + \frac{1}{q} = 1$, Hölder's inequality gives us

$$\textstyle\sum_{i=1}^k |x_i y_i| \leq (\sum_{i=1}^k |x_i|^p)^{\frac{1}{p}} (\sum_{i=1}^k |y_i|^q)^{\frac{1}{q}}$$

and therefore the lemma is an application of it with $x_i = a_i, y_i = 1, p = \ell$. $\qquad\square$

Combining Lemma J.1 for $\ell = 4$ with similar computations and upper-bounding as above, we can show that $\mathbb{E}[S^2_{n,2}]$ is bounded if for all $i$, $\mathbb{E}[\hat{\beta}^4_{\lambda_n,i}]$ and $\mathbb{E}[\hat{\beta}'^4_{\lambda_n,i}]$ are bounded, which automatically comes from the $L^4$ convergence of the soft-thresholding Lasso estimator to $\beta^\star$ needed for $\mathbb{E}[D^2_{n,2}] \to 0$.

We start by showing $\mathbb{E}[(\hat{\beta}_{\lambda_n,i} - \beta^\star_j)^4] \to 0$.

As a reminder, we have

$$\hat{\beta}_{\lambda_n,i} = \text{sign}(\hat{\beta}_{\text{OLS},i}) \begin{cases} |\hat{\beta}_{\text{OLS},i}| - \frac{\lambda_n}{n} & \text{if } |\hat{\beta}_{\text{OLS},i}| \geq \frac{\lambda_n}{n} \\ 0 & \text{if } |\hat{\beta}_{\text{OLS},i}| < \frac{\lambda_n}{n} \end{cases}$$

thus, using $(a+b)^4 \leq 8(a^4 + b^4)$, which is an application of Lemma J.1 for $\ell = 4$,

$$\begin{aligned} &\mathbb{E}[(\hat{\beta}_{\lambda_n,i} - \beta^\star_i)^4 \mid \mathbf{X}] \\ &= \mathbb{E}[(\hat{\beta}_{\text{OLS},i} - \beta^\star_i - \tfrac{\lambda_n}{n})^4 \mid \hat{\beta}_{\text{OLS},i} \geq \tfrac{\lambda_n}{n}, \mathbf{X}] \mathbb{P}(\hat{\beta}_{\text{OLS},i} \geq \tfrac{\lambda_n}{n} \mid \mathbf{X}) \\ &\quad + \mathbb{E}[(\hat{\beta}_{\text{OLS},i} - \beta^\star_i + \tfrac{\lambda_n}{n})^4 \mid \hat{\beta}_{\text{OLS},i} \leq -\tfrac{\lambda_n}{n}, \mathbf{X}] \mathbb{P}(\hat{\beta}_{\text{OLS},i} \leq -\tfrac{\lambda_n}{n} \mid \mathbf{X}) \\ &\leq 8(\mathbb{E}[(\hat{\beta}_{\text{OLS},i} - \beta^\star_i)^4 \mid \hat{\beta}_{\text{OLS},i} \geq \tfrac{\lambda_n}{n}, \mathbf{X}] + \tfrac{\lambda_n^4}{n^4}) \mathbb{P}(\hat{\beta}_{\text{OLS},i} \geq \tfrac{\lambda_n}{n} \mid \mathbf{X}) \\ &\quad + 8(\mathbb{E}[(\hat{\beta}_{\text{OLS},i} - \beta^\star_i)^4 \mid \hat{\beta}_{\text{OLS},i} \leq -\tfrac{\lambda_n}{n}, \mathbf{X}] + \tfrac{\lambda_n^4}{n^4}) \mathbb{P}(\hat{\beta}_{\text{OLS},i} \leq -\tfrac{\lambda_n}{n} \mid \mathbf{X}). \end{aligned}$$

Since $\hat{\beta}_{\text{OLS}} \mid \mathbf{X} \sim \mathcal{N}(\beta^\star, \tau^2(\mathbf{X}^\top \mathbf{X})^{-1})$, we can write $\hat{\beta}_{\text{OLS},i} = \beta^\star_i + \tilde{\tau}_n Z$ where $\tilde{\tau}_n = \frac{\tau}{\sqrt{n}}\sqrt{(\frac{\mathbf{X}^\top \mathbf{X}}{n})^{-1}_{i,i}}$ and $Z \mid \mathbf{X} \sim \mathcal{N}(0,1)$. Note that we could have $i$ as a subscript of $\tilde{\tau}_n$ and $Z$, but we will only consider one $i$ at a time in our computations and we can thus omit this subscript for both of them for the sake of notational simplicity, and we will also omit it for some additional notation we define in the rest of the proof.

Define $\alpha^{(1)}_n = \frac{1}{\tilde{\tau}_n}(\frac{\lambda_n}{n} - \beta^\star_i)$ and $\alpha^{(2)}_n = \frac{1}{\tilde{\tau}_n}(\frac{\lambda_n}{n} + \beta^\star_i)$.

Using the fourth moment derived in Lemma I.2, we have

$$\begin{aligned} &\mathbb{E}[(\hat{\beta}_{\text{OLS},i} - \beta^\star_i)^4 \mid \hat{\beta}_{\text{OLS},i} \geq \tfrac{\lambda_n}{n}, \mathbf{X}] \\ &= \mathbb{E}[(\tilde{\tau}_n Z)^4 \mid Z \geq \alpha^{(1)}_n, \mathbf{X}] \\ &= \tilde{\tau}_n^4 \mathbb{E}[Z^4 \mid Z \geq \alpha^{(1)}_n, \mathbf{X}] \\ &= \tilde{\tau}_n^4(3 + \tfrac{((\alpha^{(1)}_n)^3 + 3\alpha^{(1)}_n)\varphi(\alpha^{(1)}_n)}{1 - \Phi(\alpha^{(1)}_n)}) \end{aligned}$$

and

$$\begin{aligned} &\mathbb{E}[(\hat{\beta}_{\text{OLS},i} - \beta^\star_i)^4 \mid \hat{\beta}_{\text{OLS},i} \leq -\tfrac{\lambda_n}{n}, \mathbf{X}] \\ &= \tilde{\tau}_n^4 \mathbb{E}[Z^4 \mid Z \leq -\alpha^{(2)}_n, \mathbf{X}] \\ &= \tilde{\tau}_n^4(3 + \tfrac{((\alpha^{(2)}_n)^3 + \alpha^{(2)}_n)\varphi(-\alpha^{(2)}_n)}{\Phi(-\alpha^{(2)}_n)}). \end{aligned}$$

Since $\mathbb{P}(\hat{\beta}_{\mathrm{OLS},i} \geq \frac{\lambda_n}{n} \mid \mathbf{X}) = 1 - \Phi(\alpha_n^{(1)})$ and $\mathbb{P}(\hat{\beta}_{\mathrm{OLS},i} \leq -\frac{\lambda_n}{n} \mid \mathbf{X}) = \Phi(-\alpha_n^{(2)})$,

$$
\begin{aligned}
&\mathbb{E}[(\hat{\beta}_{\lambda_n,i} - \beta_i^\star)^4 \mid \mathbf{X}] \\
&\leq 8(\mathbb{E}[(\hat{\beta}_{\mathrm{OLS},i} - \beta_i^\star)^4 \mid \hat{\beta}_{\mathrm{OLS},i} \geq \tfrac{\lambda_n}{n}, \mathbf{X}] + \tfrac{\lambda_n^4}{n^4})\mathbb{P}(\hat{\beta}_{\mathrm{OLS},i} \geq \tfrac{\lambda_n}{n} \mid \mathbf{X}) \\
&\quad + 8(\mathbb{E}[(\hat{\beta}_{\mathrm{OLS},i} - \beta_i^\star)^4 \mid \hat{\beta}_{\mathrm{OLS},i} \leq -\tfrac{\lambda_n}{n}, \mathbf{X}] + \tfrac{\lambda_n^4}{n^4})\mathbb{P}(\hat{\beta}_{\mathrm{OLS},i} \leq -\tfrac{\lambda_n}{n} \mid \mathbf{X}) \\
&= 8(3\tilde{\tau}_n^4(1 - \Phi(\alpha_n^{(1)})) + \tilde{\tau}_n^4((\alpha_n^{(1)})^3 + 3\alpha_n^{(1)})\varphi(\alpha_n^{(1)}) + \tfrac{\lambda_n^4}{n^4}(1 - \Phi(\alpha_n^{(1)}))) \\
&\quad + 8(3\tilde{\tau}_n^4\Phi(-\alpha_n^{(2)}) + \tilde{\tau}_n^4((\alpha_n^{(2)})^3 + \alpha_n^{(2)})\varphi(-\alpha_n^{(2)}) + \tfrac{\lambda_n^4}{n^4}\Phi(-\alpha_n^{(2)})).
\end{aligned}
$$

For $X_i \overset{\text{i.i.d.}}{\sim} \mathcal{N}(0, \mathbf{I})$, we know $(\mathbf{X}^\top \mathbf{X})^{-1} \sim W_p^{-1}(\mathbf{I}, n)$ and then the diagonal element $(\mathbf{X}^\top \mathbf{X})_{i,i}^{-1}$ follows an inverse gamma distribution with shape parameter $\frac{n-p+1}{2}$ and scale parameter $\frac{1}{2}$. Therefore, $\mathbb{E}[((\mathbf{X}^\top \mathbf{X})_{i,i}^{-1})^2] = \frac{1}{(n-p-1)(n-p-3)}$ and $\mathbb{E}[((\frac{\mathbf{X}^\top \mathbf{X}}{n})_{i,i}^{-1})^2] = \frac{n^2}{(n-p-1)(n-p-3)} = o(n^2)$.

As a reminder, $\tilde{\tau}_n = \frac{\tau}{\sqrt{n}}\sqrt{(\frac{\mathbf{X}^\top \mathbf{X}}{n})_{i,i}^{-1}}$. We then have $L^1$ convergence of $\tilde{\tau}_n^4$ to 0. As previously mentioned, the function $x \mapsto x\varphi(x)$ is bounded. Similarly, a straightforward study of the behavior of the function $x \mapsto x^3\varphi(x)$ shows it is bounded. Since $\Phi$ is also bounded, and $\lambda_n = o(n)$, then

$$
\mathbb{E}[(\hat{\beta}_{\lambda_n,i} - \beta_i^\star)^4] = \mathbb{E}[\mathbb{E}[(\hat{\beta}_{\lambda_n,i} - \beta_i^\star)^4 \mid \mathbf{X}]] \to 0.
$$

We now show that $\mathbb{E}[(\hat{\beta}'_{\lambda_n,i} - \hat{\beta}_{\lambda_n,i} - \frac{V_i}{n})^4] \to 0$.

We have

$$
\begin{aligned}
&\hat{\beta}'_{\lambda_n,i} - \hat{\beta}_{\lambda_n,i} \\
&= \mathrm{sign}(\hat{\beta}'_{\mathrm{OLS},i})(|\hat{\beta}'_{\mathrm{OLS},i}| - \tfrac{\lambda_n}{n})_+ - \mathrm{sign}(\hat{\beta}_{\mathrm{OLS},i})(|\hat{\beta}_{\mathrm{OLS},i}| - \tfrac{\lambda_n}{n})_+ \\
&= \mathrm{sign}(\hat{\beta}'_{\mathrm{OLS},i})\begin{cases} |\hat{\beta}'_{\mathrm{OLS},i}| - \tfrac{\lambda_n}{n} & \text{if } |\hat{\beta}'_{\mathrm{OLS},i}| \geq \tfrac{\lambda_n}{n} \\ 0 & \text{if } |\hat{\beta}'_{\mathrm{OLS},i}| < \tfrac{\lambda_n}{n} \end{cases} - \mathrm{sign}(\hat{\beta}_{\mathrm{OLS},i})\begin{cases} |\hat{\beta}_{\mathrm{OLS},i}| - \tfrac{\lambda_n}{n} & \text{if } |\hat{\beta}_{\mathrm{OLS},i}| \geq \tfrac{\lambda_n}{n} \\ 0 & \text{if } |\hat{\beta}_{\mathrm{OLS},i}| < \tfrac{\lambda_n}{n} \end{cases} \\
&= \begin{cases} \hat{\beta}'_{\mathrm{OLS},i} - \tfrac{\lambda_n}{n} & \text{if } \hat{\beta}'_{\mathrm{OLS},i} \geq \tfrac{\lambda_n}{n} \\ \hat{\beta}'_{\mathrm{OLS},i} + \tfrac{\lambda_n}{n} & \text{if } \hat{\beta}'_{\mathrm{OLS},i} \leq -\tfrac{\lambda_n}{n} \\ 0 & \text{if } |\hat{\beta}'_{\mathrm{OLS},i}| < \tfrac{\lambda_n}{n} \end{cases} - \begin{cases} \hat{\beta}_{\mathrm{OLS},i} - \tfrac{\lambda_n}{n} & \text{if } \hat{\beta}_{\mathrm{OLS},i} \geq \tfrac{\lambda_n}{n} \\ \hat{\beta}_{\mathrm{OLS},i} + \tfrac{\lambda_n}{n} & \text{if } \hat{\beta}_{\mathrm{OLS},i} \leq -\tfrac{\lambda_n}{n} \\ 0 & \text{if } |\hat{\beta}_{\mathrm{OLS},i}| < \tfrac{\lambda_n}{n} \end{cases}.
\end{aligned}
$$

As an intermediate step, we need to show $\hat{\beta}'_{\mathrm{OLS}} - \hat{\beta}_{\mathrm{OLS}} - \frac{V}{n} \xrightarrow{\text{a.s.}} 0$.

Let $\tilde{\mathbf{X}} \triangleq (X_2, \ldots, X_n)^\top$ be the matrix of regressors for the training points except for the first one that is being changed.

We have

$$
\begin{aligned}
&\hat{\beta}'_{\mathrm{OLS}} - \hat{\beta}_{\mathrm{OLS}} \\
&= (\mathbf{X}'^\top \mathbf{X}')^{-1}\mathbf{X}'^\top \mathbf{Y}' - (\mathbf{X}^\top \mathbf{X})^{-1}\mathbf{X}^\top \mathbf{Y} \\
&= (\tilde{\mathbf{X}}^\top \tilde{\mathbf{X}} + X_1'X_1'^\top)^{-1}(\tilde{\mathbf{X}}^\top \tilde{\mathbf{Y}} + Y_1'X_1') - (\tilde{\mathbf{X}}^\top \tilde{\mathbf{X}} + X_1X_1^\top)^{-1}(\tilde{\mathbf{X}}^\top \tilde{\mathbf{Y}} + Y_1X_1) \\
&= [(\tilde{\mathbf{X}}^\top \tilde{\mathbf{X}} + X_1'X_1'^\top)^{-1} - (\tilde{\mathbf{X}}^\top \tilde{\mathbf{X}} + X_1X_1^\top)^{-1}]\tilde{\mathbf{X}}^\top \tilde{\mathbf{Y}} \\
&\quad + (\tilde{\mathbf{X}}^\top \tilde{\mathbf{X}} + X_1'X_1'^\top)^{-1}Y_1'X_1' - (\tilde{\mathbf{X}}^\top \tilde{\mathbf{X}} + X_1X_1^\top)^{-1}Y_1X_1.
\end{aligned}
$$

Using the Sherman–Morrison–Woodbury formula,

$$
\begin{aligned}
(\tilde{\mathbf{X}}^\top \tilde{\mathbf{X}} + X_1X_1^\top)^{-1} &= (\tilde{\mathbf{X}}^\top \tilde{\mathbf{X}})^{-1} - (\tilde{\mathbf{X}}^\top \tilde{\mathbf{X}})^{-1}X_1(\mathbf{I} + X_1^\top(\tilde{\mathbf{X}}^\top \tilde{\mathbf{X}})^{-1}X_1)^{-1}X_1^\top(\tilde{\mathbf{X}}^\top \tilde{\mathbf{X}})^{-1} \\
&= \tfrac{1}{n}(\tfrac{\tilde{\mathbf{X}}^\top \tilde{\mathbf{X}}}{n})^{-1} - \tfrac{1}{n^2}(\tfrac{\tilde{\mathbf{X}}^\top \tilde{\mathbf{X}}}{n})^{-1}X_1(\mathbf{I} + \tfrac{1}{n}X_1^\top(\tfrac{\tilde{\mathbf{X}}^\top \tilde{\mathbf{X}}}{n})^{-1}X_1)^{-1}X_1^\top(\tfrac{\tilde{\mathbf{X}}^\top \tilde{\mathbf{X}}}{n})^{-1} \\
&= \tfrac{1}{n}A_n - \tfrac{1}{n^2}B_n
\end{aligned}
$$

where, by the strong law of large numbers,

- $A_n \triangleq (\frac{\tilde{\mathbf{X}}^\top \tilde{\mathbf{X}}}{n})^{-1} \xrightarrow{\text{a.s.}} \mathbb{E}[X_0 X_0^\top]^{-1} = \mathbf{I},$

- $B_n \triangleq (\frac{\tilde{\mathbf{X}}^\top \tilde{\mathbf{X}}}{n})^{-1} X_1 (\mathbf{I} + \frac{1}{n} X_1^\top (\frac{\tilde{\mathbf{X}}^\top \tilde{\mathbf{X}}}{n})^{-1} X_1)^{-1} X_1^\top (\frac{\tilde{\mathbf{X}}^\top \tilde{\mathbf{X}}}{n})^{-1} \xrightarrow{\text{a.s.}} X_1 X_1^\top.$

Similarly,

$$(\tilde{\mathbf{X}}^\top \tilde{\mathbf{X}} + X_1' X_1'^\top)^{-1} = \frac{1}{n} A_n - \frac{1}{n^2} B_n'$$

with

$$B_n' \triangleq (\frac{\tilde{\mathbf{X}}^\top \tilde{\mathbf{X}}}{n})^{-1} X_1' (\mathbf{I} + \frac{1}{n} X_1'^\top (\frac{\tilde{\mathbf{X}}^\top \tilde{\mathbf{X}}}{n})^{-1} X_1')^{-1} X_1'^\top (\frac{\tilde{\mathbf{X}}^\top \tilde{\mathbf{X}}}{n})^{-1} \xrightarrow{\text{a.s.}} X_1' X_1'^\top.$$

Then

$$\hat{\beta}'_{\text{OLS}} - \hat{\beta}_{\text{OLS}} = \frac{1}{n^2}(B_n - B_n')\tilde{\mathbf{X}}^\top \tilde{\mathbf{Y}} + (\frac{1}{n} A_n - \frac{1}{n^2} B_n') Y_1' X_1' - (\frac{1}{n} A_n - \frac{1}{n^2} B_n) Y_1 X_1$$
$$= \frac{1}{n}(B_n - B_n')\frac{\tilde{\mathbf{X}}^\top \tilde{\mathbf{Y}}}{n} + (\frac{1}{n} A_n - \frac{1}{n^2} B_n') Y_1' X_1' - (\frac{1}{n} A_n - \frac{1}{n^2} B_n) Y_1 X_1$$

where $\frac{\tilde{\mathbf{X}}^\top \tilde{\mathbf{Y}}}{n} \xrightarrow{\text{a.s.}} \mathbb{E}[Y_0 X_0] = \beta^\star$, by the strong law of large numbers.

Therefore,

$$n(\hat{\beta}'_{\text{OLS}} - \hat{\beta}_{\text{OLS}}) \xrightarrow{\text{a.s.}} (X_1 X_1^\top - X_1' X_1'^\top)\beta^\star + Y_1' X_1' - Y_1 X_1$$
$$= (Y_1' - X_1'^\top \beta^\star) X_1' - (Y_1 - X_1^\top \beta^\star) X_1$$
$$= V.$$

We can write

$$(\hat{\beta}'_{\lambda_n,i} - \hat{\beta}_{\lambda_n,i} - \tfrac{V_i}{n})^4 = (\hat{\beta}'_{\text{OLS},i} - \hat{\beta}_{\text{OLS},i} - \tfrac{V_i}{n})^4 \mathbb{1}\left[\hat{\beta}_{\text{OLS},i} \geq \tfrac{\lambda_n}{n}, \hat{\beta}'_{\text{OLS},i} \geq \tfrac{\lambda_n}{n}\right]$$
$$+ (\hat{\beta}'_{\text{OLS},i} - \hat{\beta}_{\text{OLS},i} - \tfrac{V_i}{n})^4 \mathbb{1}\left[\hat{\beta}_{\text{OLS},i} \leq -\tfrac{\lambda_n}{n}, \hat{\beta}'_{\text{OLS},i} \leq -\tfrac{\lambda_n}{n}\right]$$
$$+ (\hat{\beta}'_{\text{OLS},i} - \hat{\beta}_{\text{OLS},i} - 2\tfrac{\lambda_n}{n} - \tfrac{V_i}{n})^4 \mathbb{1}\left[\hat{\beta}_{\text{OLS},i} \leq -\tfrac{\lambda_n}{n}, \hat{\beta}'_{\text{OLS},i} \geq \tfrac{\lambda_n}{n}\right]$$
$$+ (\hat{\beta}'_{\text{OLS},i} - \hat{\beta}_{\text{OLS},i} + 2\tfrac{\lambda_n}{n} - \tfrac{V_i}{n})^4 \mathbb{1}\left[\hat{\beta}_{\text{OLS},i} \geq \tfrac{\lambda_n}{n}, \hat{\beta}'_{\text{OLS},i} \leq -\tfrac{\lambda_n}{n}\right]$$
$$+ (\hat{\beta}'_{\text{OLS},i} - \tfrac{\lambda_n}{n} - \tfrac{V_i}{n})^4 \mathbb{1}\left[|\hat{\beta}_{\text{OLS},i}| < \tfrac{\lambda_n}{n}, \hat{\beta}'_{\text{OLS},i} \geq \tfrac{\lambda_n}{n}\right]$$
$$+ (\hat{\beta}'_{\text{OLS},i} + \tfrac{\lambda_n}{n} - \tfrac{V_i}{n})^4 \mathbb{1}\left[|\hat{\beta}_{\text{OLS},i}| < \tfrac{\lambda_n}{n}, \hat{\beta}'_{\text{OLS},i} \leq -\tfrac{\lambda_n}{n}\right]$$
$$+ (\hat{\beta}_{\text{OLS},i} - \tfrac{\lambda_n}{n} + \tfrac{V_i}{n})^4 \mathbb{1}\left[\hat{\beta}_{\text{OLS},i} \geq \tfrac{\lambda_n}{n}, |\hat{\beta}'_{\text{OLS},i}| < \tfrac{\lambda_n}{n}\right]$$
$$+ (\hat{\beta}_{\text{OLS},i} + \tfrac{\lambda_n}{n} + \tfrac{V_i}{n})^4 \mathbb{1}\left[\hat{\beta}_{\text{OLS},i} \leq -\tfrac{\lambda_n}{n}, |\hat{\beta}'_{\text{OLS},i}| < \tfrac{\lambda_n}{n}\right]$$
$$+ (\tfrac{V_i}{n})^4 \mathbb{1}\left[|\hat{\beta}_{\text{OLS},i}| < \tfrac{\lambda_n}{n}, |\hat{\beta}'_{\text{OLS},i}| < \tfrac{\lambda_n}{n}\right]$$

and we have a similar expression for $(\hat{\beta}'_{\lambda_n,i} - \hat{\beta}_{\lambda_n,i} - \tfrac{V_i}{n})^6$ with terms taken to the sixth power.

Since $\hat{\beta}_{\text{OLS}} \mid \mathbf{X} \sim \mathcal{N}(\beta^\star, \tau^2 (\mathbf{X}^\top \mathbf{X})^{-1})$ and we can bound the central moments of a Normal with the powers of its variance, there exists $C > 0$ such that $\mathbb{E}[(\hat{\beta}_{\text{OLS},i} - \beta_i^\star)^6 \mid \mathbf{X}] \leq C(\tau^2 (\mathbf{X}^\top \mathbf{X})_{i,i}^{-1})^3 = C\tau^6 ((\mathbf{X}^\top \mathbf{X})_{i,i}^{-1})^3.$

For $X_i \overset{\text{i.i.d.}}{\sim} \mathcal{N}(0, \mathbf{I})$, we know $(\mathbf{X}^\top \mathbf{X})^{-1} \sim W_p^{-1}(\mathbf{I}, n)$ and then the diagonal element $(\mathbf{X}^\top \mathbf{X})_{i,i}^{-1}$ follows an inverse gamma distribution with shape parameter $\frac{n-p+1}{2}$ and scale parameter $\frac{1}{2}$. Therefore, $\mathbb{E}[((\mathbf{X}^\top \mathbf{X})_{i,i}^{-1})^3] = \frac{1}{(n-p-1)(n-p-3)(n-p-5)}$, which means $\mathbb{E}[(\hat{\beta}_{\text{OLS},i} - \beta_i^\star)^6]$ and thus $\mathbb{E}[\hat{\beta}_{\text{OLS},i}^6]$, by an application of Lemma J.1 for $\ell = 6$, are bounded. Similarly, $\mathbb{E}[\hat{\beta}_{\text{OLS},i}'^6]$ is bounded.

Consequently, since $\lambda_n = o(n)$ and $\mathbb{E}[\hat{\beta}_{\mathrm{OLS},i}^6]$ and $\mathbb{E}[\hat{\beta}_{\mathrm{OLS},i}'^6]$ are bounded, the almost sure convergence of the fourth moment turns into $L^1$ convergence to 0.

Therefore,

$$
\begin{aligned}
\gamma(h_n^{\mathrm{sing}}) &\sim \tfrac{1}{n^2}\mathbb{E}[(2(Y_0 X_0^\top - \beta^{\star\top})V + \mathrm{tr}((X_0 X_0^\top - \mathbf{I})M))^2] \\
&= \tfrac{1}{n^2}\mathbb{E}[(2Y_0 X_0^\top V - 2\beta^{\star\top}V - \mathrm{tr}((X_0 X_0^\top - \mathbf{I})(V\beta^{\star\top} + \beta^\star V^\top)))^2] \\
&= \tfrac{1}{n^2}\mathbb{E}[(2Y_0 X_0^\top V - 2X_0^\top \beta^\star X_0^\top V)^2] \\
&= \tfrac{1}{n^2}\mathbb{E}[(2(Y_0 - X_0^\top \beta^\star)X_0^\top V)^2]
\end{aligned}
\tag{8}
$$

where $V = (Y_1' - X_1'^\top \beta^\star)X_1' - (Y_1 - X_1^\top \beta^\star)X_1$.

## K. Proof of Theorem 3.3: Relative instability of Lasso comparisons

Instantiate the ST notation of Theorem 3.1, and define the shorthand

$$
\begin{aligned}
V &\triangleq 2(Y_0 X_0^\top - \beta^{\star\top}\mathbb{E}[X_0 X_0^\top])^\top, \quad M \triangleq (X_0 X_0^\top - \mathbb{E}[X_0 X_0^\top]), \\
A &\triangleq V^\top \mathbb{E}[\hat{\beta}_{\lambda_n+\delta_n}^{\mathrm{LASSO}} - \hat{\beta}_{\lambda_n}^{\mathrm{LASSO}}] + \mathrm{tr}(M, \mathbb{E}[\hat{\beta}_{\lambda_n}^{\mathrm{LASSO}}\hat{\beta}_{\lambda_n}^{\mathrm{LASSO}\top} - \hat{\beta}_{\lambda_n+\delta_n}^{\mathrm{LASSO}}\hat{\beta}_{\lambda_n+\delta_n}^{\mathrm{LASSO}\top}]), \quad \text{and} \\
B &\triangleq V^\top \mathbb{E}[\hat{\beta}_{\lambda_n+\delta_n} - \hat{\beta}_{\lambda_n}] + \mathrm{tr}(M, \mathbb{E}[\hat{\beta}_{\lambda_n}\hat{\beta}_{\lambda_n}^\top - \hat{\beta}_{\lambda_n+\delta_n}\hat{\beta}_{\lambda_n+\delta_n}^\top]).
\end{aligned}
$$

We will establish the $\sigma^2(\tilde{h}_n^{\mathrm{diff}})$ upper bound in Appendix K.1 and the $\gamma(\tilde{h}_n^{\mathrm{diff}})$ lower bound in Appendix K.2.

### K.1. $\sigma^2(\tilde{h}_n^{\mathrm{diff}})$ upper bound

By Lemma E.1 and Cauchy-Schwarz, we have

$$
\begin{aligned}
\sigma^2(\tilde{h}_n^{\mathrm{diff}}) - \sigma^2(h_n^{\mathrm{diff}}) &= \mathbb{E}[A^2 - B^2] = \mathbb{E}[(A-B)^2] + \mathbb{E}[(A-B)2B] \\
&\leq \mathbb{E}[(A-B)^2] + 2\sqrt{\mathbb{E}[(A-B)^2]\mathbb{E}[B^2]} = \mathbb{E}[(A-B)^2] + 2\sqrt{\mathbb{E}[(A-B)^2]\sigma^2(h_n^{\mathrm{diff}})}.
\end{aligned}
\tag{9}
$$

Meanwhile, Cauchy-Schwarz, the triangle inequality, and the definition of the operator norm imply

$$
\begin{aligned}
|A - B| &= |V^\top \mathbb{E}[\hat{\beta}_{\lambda_n+\delta_n}^{\mathrm{LASSO}} - \hat{\beta}_{\lambda_n+\delta_n} + \hat{\beta}_{\lambda_n} - \hat{\beta}_{\lambda_n}^{\mathrm{LASSO}}] \\
&\quad + \mathrm{tr}(M, \mathbb{E}[\hat{\beta}_{\lambda_n}^{\mathrm{LASSO}}\hat{\beta}_{\lambda_n}^{\mathrm{LASSO}\top} - \hat{\beta}_{\lambda_n}\hat{\beta}_{\lambda_n}^\top + \hat{\beta}_{\lambda_n+\delta_n}^{\mathrm{LASSO}}\hat{\beta}_{\lambda_n+\delta_n}^{\mathrm{LASSO}\top} - \hat{\beta}_{\lambda_n+\delta_n}\hat{\beta}_{\lambda_n+\delta_n}^\top])| \\
&\leq \|V\|_2 \mathbb{E}[\|\hat{\beta}_{\lambda_n+\delta_n}^{\mathrm{LASSO}} - \hat{\beta}_{\lambda_n+\delta_n}\|_2 + \|\hat{\beta}_{\lambda_n} - \hat{\beta}_{\lambda_n}^{\mathrm{LASSO}}\|_2] \\
&\quad + \|M\|_{\mathrm{op}} \mathbb{E}[\|\hat{\beta}_{\lambda_n}^{\mathrm{LASSO}} - \hat{\beta}_{\lambda_n}\|_2 (\|\hat{\beta}_{\lambda_n}^{\mathrm{LASSO}} - \hat{\beta}_{\lambda_n}\|_2 + 2\|\hat{\beta}_{\lambda_n}\|_2) \\
&\quad + \|\hat{\beta}_{\lambda_n+\delta_n}^{\mathrm{LASSO}} - \hat{\beta}_{\lambda_n+\delta_n}\|_2 (\|\hat{\beta}_{\lambda_n+\delta_n}^{\mathrm{LASSO}} - \hat{\beta}_{\lambda_n+\delta_n}\|_2 + 2\|\hat{\beta}_{\lambda_n+\delta_n}\|_2)]
\end{aligned}
$$

Now, since $\mathbb{E}[\|\hat{\beta}_{\lambda_n}^{\mathrm{LASSO}} - \hat{\beta}_{\lambda_n}\|_2^2] = O(\frac{\lambda_n^2}{n^3})$ by Lemma 2.3, $\mathbb{E}[\|\hat{\beta}_{\lambda_n}\|_2^2] \leq \mathbb{E}[\|\hat{\beta}_{\mathrm{OLS}}\|_2^2] = O(1)$ by (Afendras & Markatou, 2016, Thm. 1), and $\delta_n = O(\lambda_n)$, we have $\mathbb{E}[(A-B)^2] = O(\frac{\lambda_n^2 + (\lambda_n + \delta_n)^2}{n^3}) = O(\frac{\lambda_n^2}{n^3})$. Therefore, since $\sigma^2(h_n^{\mathrm{diff}}) = O(\frac{1}{n^2})$ by Theorem 3.1 and $\lambda_n = O(\sqrt{n})$, we can conclude from inequality (9) that $\sigma^2(\tilde{h}_n^{\mathrm{diff}}) = O(\frac{1}{n^2})$ as well.

### K.2. $\gamma(\tilde{h}_n^{\mathrm{diff}})$ lower bound

Let $(A', B')$ be an exchangeable copy of $(A, B)$ in which the first datapoint $Z_1$ has been replaced by an i.i.d. copy $Z_1'$. Then, by the triangle inequality and exchangeability,

$$
\begin{aligned}
\sqrt{\gamma(\tilde{h}_n^{\mathrm{diff}})} - \sqrt{\gamma(h_n^{\mathrm{diff}})} &= \sqrt{\mathbb{E}[(A-A')^2]} - \sqrt{\mathbb{E}[(B-B')^2]} \\
&\geq -\sqrt{\mathbb{E}[(A-B)^2]} - \sqrt{\mathbb{E}[(A'-B')^2]} = -2\sqrt{\mathbb{E}[(A-B)^2]}.
\end{aligned}
\tag{10}
$$

Since $\sqrt{\gamma(h_n^{\mathrm{diff}})} = \Omega(\frac{1}{n^{5/4}})$ by Theorem 3.1, $\lambda_n = O(\sqrt{n})$ by assumption, and $\sqrt{\mathbb{E}[(A-B)^2]} = O(\frac{\lambda_n}{n^{3/2}}) = O(\frac{1}{n})$ by Appendix K.1, we also have $\sqrt{\gamma(\tilde{h}_n^{\mathrm{diff}})} = \Omega(\frac{1}{n^{5/4}})$ by (10).

## L. Proof of Theorem 3.4: Relative stability of the Lasso

Instantiate the ST notation of Theorem 3.2, and define the shorthand

$$V \triangleq 2(Y_0 X_0^\top - \beta^{\star\top} \mathbb{E}[X_0 X_0^\top])^\top, \quad M \triangleq (X_0 X_0^\top - \mathbb{E}[X_0 X_0^\top]),$$
$$B \triangleq -V^\top \mathbb{E}[\hat{\beta}_{\lambda_n}^{\mathrm{LASSO}}] + \mathrm{tr}(M, \mathbb{E}[\hat{\beta}_{\lambda_n}^{\mathrm{LASSO}} \hat{\beta}_{\lambda_n}^{\mathrm{LASSO}\top}]), \quad \text{and}$$
$$A \triangleq -V^\top \mathbb{E}[\hat{\beta}_{\lambda_n}] + \mathrm{tr}(M, \mathbb{E}[\hat{\beta}_{\lambda_n} \hat{\beta}_{\lambda_n}^\top]).$$

We will establish the $\sigma^2(\tilde{h}_n^{\mathrm{sing}})$ lower bound in Appendix L.1 and the $\gamma(\tilde{h}_n^{\mathrm{sing}})$ upper bound in Appendix L.2.

### L.1. $\sigma^2(\tilde{h}_n^{\mathrm{sing}})$ lower bound

By Lemma I.1 and Cauchy-Schwarz, we have

$$\sigma^2(h_n^{\mathrm{sing}}) - \sigma^2(\tilde{h}_n^{\mathrm{sing}}) = \mathbb{E}[A^2 - B^2] = \mathbb{E}[(A - B)2A] - \mathbb{E}[(A-B)^2] \tag{11}$$
$$\leq 2\sqrt{\mathbb{E}[(A-B)^2]\mathbb{E}[A^2]} = 2\sqrt{\mathbb{E}[(A-B)^2]\sigma^2(h_n^{\mathrm{sing}})}.$$

Meanwhile, Cauchy-Schwarz, the triangle inequality, and the definition of the operator norm imply

$$|A - B| = |V^\top \mathbb{E}[\hat{\beta}_{\lambda_n} - \hat{\beta}_{\lambda_n}^{\mathrm{LASSO}}] + \mathrm{tr}(M, \mathbb{E}[\hat{\beta}_{\lambda_n}^{\mathrm{LASSO}} \hat{\beta}_{\lambda_n}^{\mathrm{LASSO}\top} - \hat{\beta}_{\lambda_n} \hat{\beta}_{\lambda_n}^\top])|$$
$$\leq \|V\|_2 \mathbb{E}[\|\hat{\beta}_{\lambda_n} - \hat{\beta}_{\lambda_n}^{\mathrm{LASSO}}\|_2] + \|M\|_{\mathrm{op}} \mathbb{E}[\|\hat{\beta}_{\lambda_n}^{\mathrm{LASSO}} - \hat{\beta}_{\lambda_n}\|_2 (\|\hat{\beta}_{\lambda_n}^{\mathrm{LASSO}} - \hat{\beta}_{\lambda_n}\|_2 + 2\|\hat{\beta}_{\lambda_n}\|_2).$$

Now, since $\mathbb{E}[\|\hat{\beta}_{\lambda_n}^{\mathrm{LASSO}} - \hat{\beta}_{\lambda_n}\|_2^2] = O(\frac{\lambda_n^2}{n^3})$ by Lemma 2.3 and $\mathbb{E}[\|\hat{\beta}_{\lambda_n}\|_2^2] \leq \mathbb{E}[\|\hat{\beta}_{\mathrm{OLS}}\|_2^2] = O(1)$ by (Afendras & Markatou, 2016, Thm. 1), we have $\mathbb{E}[(A-B)^2] = O(\frac{\lambda_n^2}{n^3})$. Therefore, since $\sigma^2(h_n^{\mathrm{sing}}) = \Theta(1)$ by Theorem 3.2 and $\lambda_n = o(n^{3/2})$, we can conclude from inequality (11) that $\sigma^2(\tilde{h}_n^{\mathrm{sing}}) = \Omega(1)$ as well.

### L.2. $\gamma(\tilde{h}_n^{\mathrm{sing}})$ upper bound

Let $(A', B')$ be an exchangeable copy of $(A, B)$ in which the first datapoint $Z_1$ has been replaced by an i.i.d. copy $Z_1'$. Then, by Lemma I.1, Cauchy-Schwarz, and exchangeability,

$$\gamma(\tilde{h}_n^{\mathrm{sing}}) - \gamma(h_n^{\mathrm{sing}}) = \mathbb{E}[(B - B')^2 - (A - A')^2] \tag{12}$$
$$= \mathbb{E}[(B - A + A' - B')^2] + \mathbb{E}[(B - A + A' - B')2(A - A')]$$
$$\leq 4\mathbb{E}[(A-B)^2] + 4\sqrt{\mathbb{E}[(A-B)^2]\mathbb{E}[(A-A')^2]} = 4\mathbb{E}[(A-B)^2] + 4\sqrt{\mathbb{E}[(A-B)^2]\gamma(h_n^{\mathrm{sing}})}.$$

Since $\gamma(h_n^{\mathrm{sing}}) = O(\frac{1}{n^2})$ by Theorem 3.1, $\sqrt{\mathbb{E}[(A-B)^2]} = O(\frac{\lambda_n}{n^{3/2}})$ by Appendix L.1, and $\lambda_n = o(n)$, we conclude from inequality (12) that $\gamma(\tilde{h}_n^{\mathrm{sing}}) = O(\frac{\lambda_n^2}{n^3} + \frac{\lambda_n}{n^{7/2}}) = o(\frac{1}{n})$.

## M. Experimental Setup Details

We provide additional details about the numerical experiments presented in Section 5.

In our simulations, we work with the following sample sizes for the full set size $\frac{nk}{k-1}$: 100, 1,000, 10,000, 100,000, which means $n$ takes the following values: 90, 900, 9,000, 90,000.

For simulations with the Lasso estimator, we used the implementation from `scikit-learn`. For the KDE plots, we called `kdeplot` from the `seaborn` library.

We perform 50,000 replications to sample from $\frac{\sqrt{\frac{nk}{k-1}}}{\sigma(h_n)}(\hat{R}_n - R_n)$ and $\frac{\sqrt{\frac{nk}{k-1}}}{\hat{\sigma}_n(h_n)}(\hat{R}_n - R_n)$. We ensured reproducibility by setting random seeds at the start of all replications.

Regarding the inner cross-validation used to determine $\lambda_n$ in each iteration of the outer cross-validation, we performed an adaptive grid search via $(k-1)$-fold cross-validation on the training set of size $n$, based on the initial split of the cross-validation on the full set of size $\frac{nk}{k-1}$. For the adaptive grid search scheme, we started with powers of 10, identified

the best choice of penalization, subdivided around this choice with 10 values with an exponential scaling, and did so 3 additional times to identify the optimal penalization with precision.

We now introduce two lemmas that allow us to properly estimate $\sigma^2(h_n)$, $\gamma(h_n)$ and $R_n$.

**Lemma M.1** ($\sigma^2(h_n)$ rewriting for Monte Carlo estimation)**.**

$$\sigma^2(h_n) = \mathbb{E}[h_n(Z_0, \mathbf{Z})(h_n(Z_0, \tilde{\mathbf{Z}}) - h_n(\tilde{Z}_0, \tilde{\mathbf{Z}}))]$$

*where $\tilde{Z}_0$ and $\tilde{\mathbf{Z}}$ are independent draws from the same distribution as $Z_0$ and $\mathbf{Z}$, respectively.*

**Proof**

$$\begin{aligned}
\sigma^2(h_n) &= \mathrm{Var}(\mathbb{E}[h_n(Z_0, \mathbf{Z}) \mid Z_0]) \\
&= \mathbb{E}[\mathbb{E}[h_n(Z_0, \mathbf{Z}) \mid Z_0]^2] - \mathbb{E}[h_n(Z_0, \mathbf{Z})]^2 \\
&= \mathbb{E}[\mathbb{E}[h_n(Z_0, \mathbf{Z})h_n(Z_0, \tilde{\mathbf{Z}}) \mid Z_0]] - \mathbb{E}[h_n(Z_0, \mathbf{Z})h_n(\tilde{Z}_0, \tilde{\mathbf{Z}})] \\
&= \mathbb{E}[h_n(Z_0, \mathbf{Z})h_n(Z_0, \tilde{\mathbf{Z}})] - \mathbb{E}[h_n(Z_0, \mathbf{Z})h_n(\tilde{Z}_0, \tilde{\mathbf{Z}})] \\
&= \mathbb{E}[h_n(Z_0, \mathbf{Z})(h_n(Z_0, \tilde{\mathbf{Z}}) - h_n(\tilde{Z}_0, \tilde{\mathbf{Z}}))]
\end{aligned}$$

$\square$

**Lemma M.2** (Conditional expectation and $\gamma(h_n)$ rewriting for Monte Carlo estimation)**.** *If the features are drawn from a distribution with mean 0 and identity covariance matrix, we have*

$$\mathbb{E}[h_n^{\mathrm{sing}}(Z_0, \mathbf{Z}) \mid \mathbf{Z}] = \tau^2 + \|\beta^\star - \hat{\beta}\|_2^2,$$

*and thus*

$$\mathbb{E}[h_n^{\mathrm{diff}}(Z_0, \mathbf{Z}) \mid \mathbf{Z}] = \|\beta^\star - \hat{\beta}_1\|_2^2 - \|\beta^\star - \hat{\beta}^{(2)}\|_2^2,$$

$$\gamma(h_n^{\mathrm{sing}}) = \mathbb{E}[(h_n^{\mathrm{sing}}(Z_0, \mathbf{Z}) - \|\beta^\star - \hat{\beta}\|_2^2 - (h_n^{\mathrm{sing}}(Z_0, \mathbf{Z}') - \|\beta^\star - \hat{\beta}'\|_2^2))^2],$$

*and*

$$\gamma(h_n^{\mathrm{diff}}) = \mathbb{E}[(h_n^{\mathrm{diff}}(Z_0, \mathbf{Z}) - \|\beta^\star - \hat{\beta}_1\|_2^2 + \|\beta^\star - \hat{\beta}^{(2)}\|_2^2 - (h_n^{\mathrm{diff}}(Z_0, \mathbf{Z}') - \|\beta^\star - \hat{\beta}_1'\|_2^2 + \|\beta^\star - \hat{\beta}'^{(2)}\|_2^2))^2].$$

**Proof**    Starting from the expression of $\mathbb{E}[h_n(Z_0, \mathbf{Z}) \mid \mathbf{Z}]$ in Lemma I.1, we have

$$\begin{aligned}
\mathbb{E}[h_n^{\mathrm{sing}}(Z_0, \mathbf{Z}) \mid \mathbf{Z}] &= \mathbb{E}[Y_0^2] - 2\beta^{\star\top}\mathbb{E}[X_0 X_0^\top]\hat{\beta} + \mathrm{tr}(\mathbb{E}[X_0 X_0^\top]\hat{\beta}\hat{\beta}^\top) \\
&= \mathrm{Var}(Y_0) + \mathbb{E}[Y_0]^2 - 2\beta^{\star\top}\mathbb{E}[X_0 X_0^\top]\hat{\beta} + \mathrm{tr}(\mathbb{E}[X_0 X_0^\top]\hat{\beta}\hat{\beta}^\top) \\
&= \beta^{\star\top}\mathrm{Var}(X_0)\beta^\star + \tau^2 + (\mathbb{E}[X_0]^\top\beta^\star)^2 - 2\beta^{\star\top}\mathbb{E}[X_0 X_0^\top]\hat{\beta} + \mathrm{tr}(\mathbb{E}[X_0 X_0^\top]\hat{\beta}\hat{\beta}^\top) \\
&= \beta^{\star\top}\mathrm{Var}(X_0)\beta^\star + \tau^2 + \beta^{\star\top}\mathbb{E}[X_0]\mathbb{E}[X_0]^\top\beta^\star - 2\beta^{\star\top}\mathbb{E}[X_0 X_0^\top]\hat{\beta} + \hat{\beta}^\top\mathbb{E}[X_0 X_0^\top]\hat{\beta} \\
&= \tau^2 + \beta^{\star\top}\mathbb{E}[X_0 X_0^\top]\beta^\star + \hat{\beta}^\top\mathbb{E}[X_0 X_0^\top]\hat{\beta} - 2\beta^{\star\top}\mathbb{E}[X_0 X_0^\top]\hat{\beta} \\
&= \tau^2 + (\beta^\star - \hat{\beta})^\top\mathbb{E}[X_0 X_0^\top](\beta^\star - \hat{\beta}) \\
&= \tau^2 + \|\beta^\star - \hat{\beta}\|_2^2
\end{aligned}$$

since the features are drawn from a distribution with mean 0 and identity covariance matrix. The other three expressions follow from the definition of the quantities.    $\square$

The Monte Carlo estimation of $\sigma^2(h_n)$ and $\gamma(h_n)$ is based on 5,000,000 replications when using deterministic $\lambda_n$, but on 1,000,000 when $\lambda_n$ is selected via inner cross-validation due to computational complexity. Based on the Monte Carlo standard errors obtained for $\sigma^2(h_n)$ and $\gamma(h_n)$, we applied the Delta method as follows to obtain a standard error

for $r(h_n) = \frac{n \cdot \gamma(h_n)}{\sigma^2(h_n)}$. We define $f(x,y) = \frac{nx}{y}$ and we denote by $M$ the number of Monte Carlo replications used to estimate $\sigma^2(h_n)$ and $\gamma(h_n)$. Starting from the Monte Carlo standard errors $\frac{\sigma_x}{\sqrt{M}}$ of $\sigma^2(h_n)$ and $\frac{\sigma_y}{\sqrt{M}}$ of $\gamma(h_n)$, and using $\nabla f = (\frac{n}{y}, -\frac{nx}{y^2})$, we get to a standard error for $r(h_n)$ by computing

$$\nabla f(x,y)^\top \mathrm{diag}(\sigma_x^2, \sigma_y^2) \nabla f(x,y) = \frac{n^2\sigma_x^2}{y^2} + \frac{n^2 x^2 \sigma_y^2}{y^4}.$$

Denoting the Monte Carlo estimates of $\sigma^2(h_n)$ and $\gamma(h_n)$ by $\hat{x}$ and $\hat{y}$, respectively, the standard error we use for $r(h_n)$ is then

$$\frac{1}{\sqrt{M}} \sqrt{\frac{n^2\sigma_x^2}{\hat{y}^2} + \frac{n^2\hat{x}^2\sigma_y^2}{\hat{y}^4}}.$$

# N. Additional Experimental Results

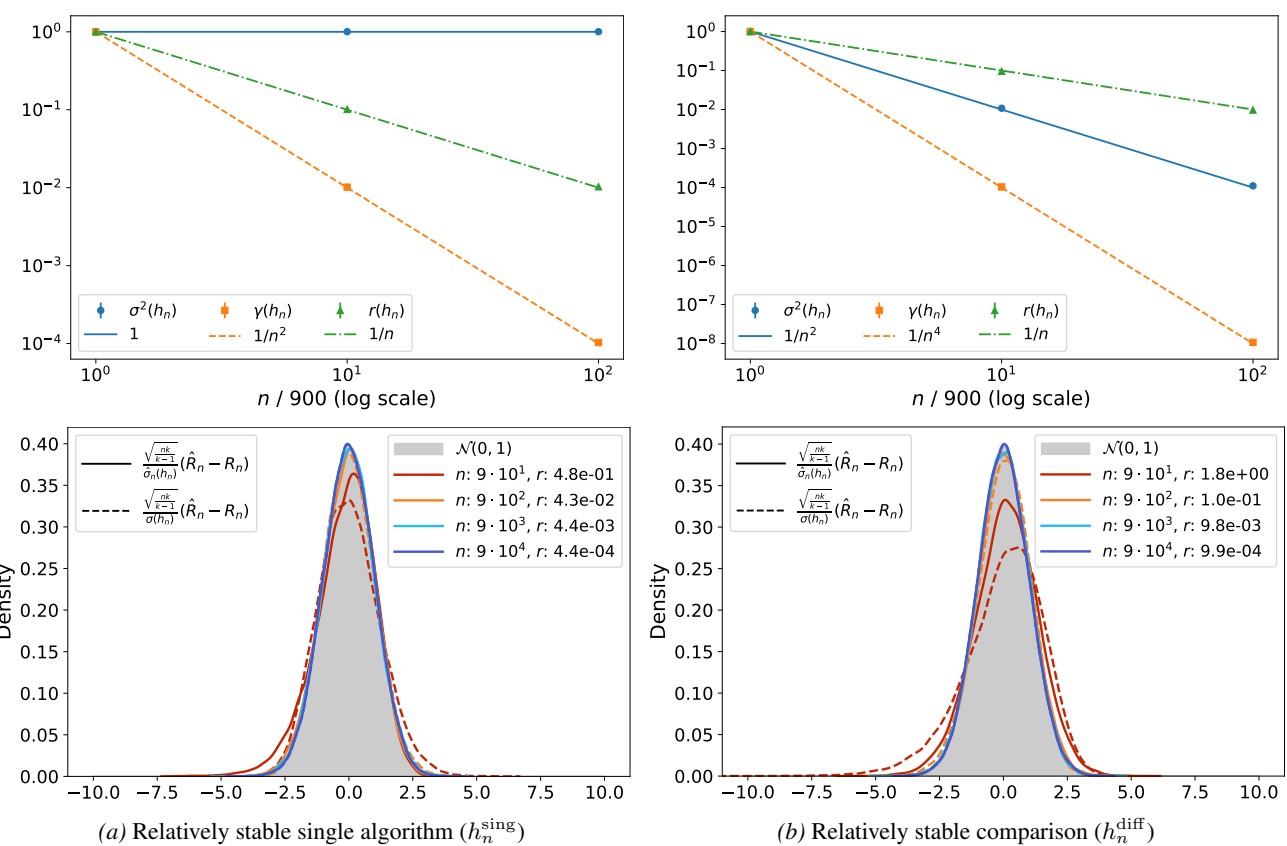

*(a) Relatively stable single algorithm ($h_n^{\mathrm{sing}}$)*   *(b) Relatively stable comparison ($h_n^{\mathrm{diff}}$)*

*Figure 4.* Ridge regression with $\lambda_n = \sqrt{n}$ when $\beta^\star = (3, 1, -5, 3, 0, 0, 0, 0, 0, 0)$. **Top:** $\sigma^2(h_n)$, $\gamma(h_n)$ and $r(h_n)$ all normalized by their values at $n = 900$. **Bottom:** (best viewed in color) KDE plots for $\frac{\sqrt{\frac{nk}{k-1}}}{\hat{\sigma}_n(h_n)}(\hat{R}_n - R_n)$ (solid curves) and $\frac{\sqrt{\frac{nk}{k-1}}}{\sigma(h_n)}(\hat{R}_n - R_n)$ (dashed curves).

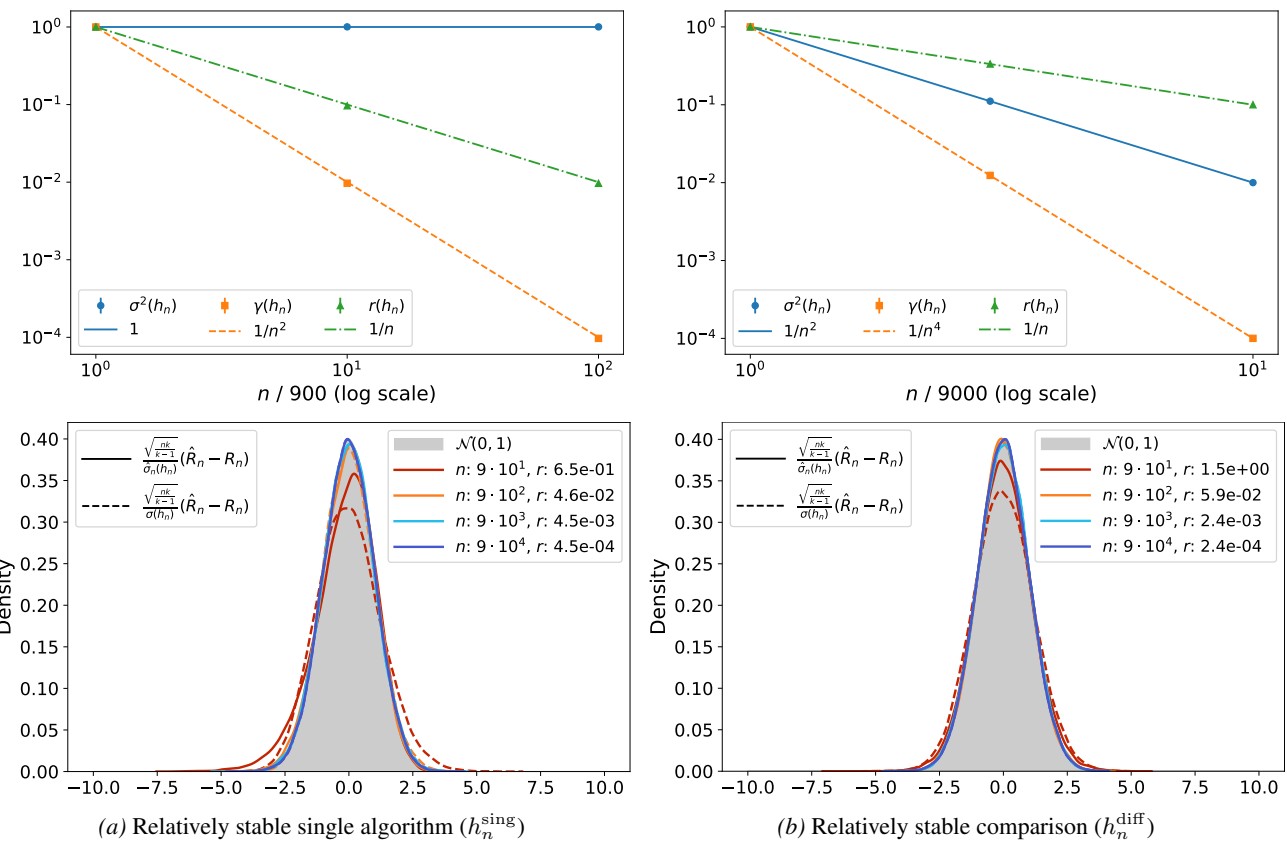

*(a) Relatively stable single algorithm ($h_n^{\mathrm{sing}}$)*   *(b) Relatively stable comparison ($h_n^{\mathrm{diff}}$)*

*Figure 5.* ST with $\lambda_n = \sqrt{n}$ when $\beta^\star = (3, 1, -5, 3, 4, -3, 10, 8, 5, 2)$. **Top:** $\sigma^2(h_n)$, $\gamma(h_n)$ and $r(h_n)$ all normalized by their values at $n = 900$ for single algorithm and at $n = 9000$ for comparison. **Bottom:** (best viewed in color) KDE plots for $\frac{\sqrt{\frac{nk}{k-1}}}{\hat{\sigma}_n(h_n)}(\hat{R}_n - R_n)$ (solid curves) and $\frac{\sqrt{\frac{nk}{k-1}}}{\sigma(h_n)}(\hat{R}_n - R_n)$ (dashed curves).

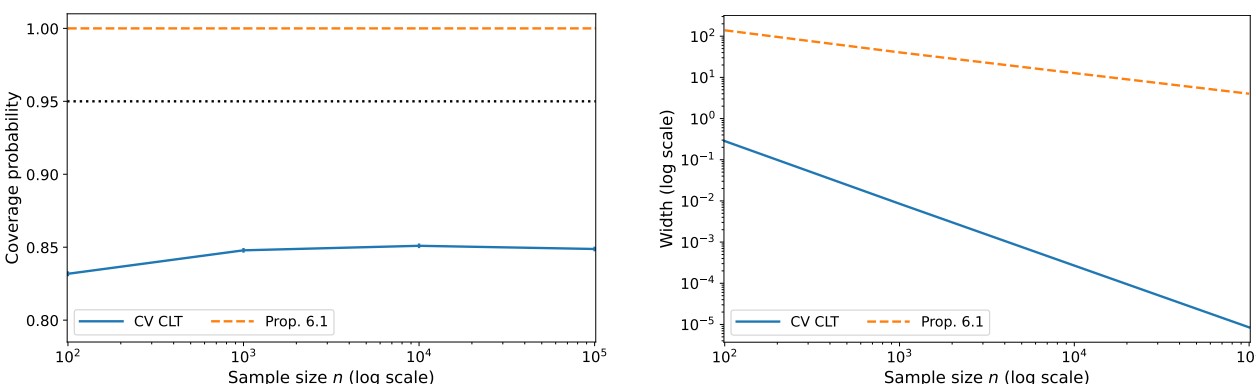

*Figure 6.* ST with $\lambda_n = \sqrt{n}$ when $\beta^\star = (3, 1, -5, 3, 0, 0, 0, 0, 0, 0)$. Test error coverage (left) and width (right) of the confidence interval derived from the cross-validation central limit theorem (Bayle et al., 2020) and the confidence interval built following Proposition 6.1, with target coverage $1 - \alpha$ where $\alpha = 0.05$. The asymptotic $(1 - \alpha)$-coverage CV CLT confidence interval undercovers for the relatively unstable comparison of two ST fits, while the confidence interval from Proposition 6.1 overcovers with width multiple orders of magnitude larger than the CV CLT interval.

