# OpenReview forum: "The Relative Instability of Model Comparison with Cross-validation"
_ICML.cc/2026/Conference — ICML 2026 spotlight_

### Official Review · Reviewer_m4TW · 2026-02-23

**Soundness:** 4
**Presentation:** 4
**Significance:** 4
**Originality:** 3
**Overall Recommendation:** 5
**Confidence:** 4

**Summary:**

The authors study the relative stability of two procedures (soft-thresholding and lasso). They prove that even if these procedures are stable individually (relatively to changing one observation of the data set), they are not stable when the hyperparameters also vary. This phenomenon has a direct consequence on inference based on CLT for cross-validation strategies: the confidence intervals built from such approaches can have a low coverage, resulting in inexact conclusion. The authors illustrate their theoretical findings with some experiments, showing that the limiting empirical distribution of CV error does not match the distribution expected from existing theory.

**Compliance With Llm Reviewing Policy:**

Affirmed.

**Final Justification:**

I am convinced that it is an excellent paper and I will maintain my score.

**Key Questions For Authors:**

- Could you discuss with more details the difference between the existing works you mentioned and yours?

**Limitations:**

yes

**Strengths And Weaknesses:**

Soundness and presentation

The theoretical results are precise, with well-formalized assumptions. I checked the first part of the proofs and did not find any errors. Even if the proofs are quite technical, the numerous comments between equations make them easy to follow and understand.
I have the same comments for the core of the manuscript. I am not an expert of the cross-validation literature but the paper was easy to read and the discussion also easy to understand. Moreover, the code used to produce the experiments is available in the supplementary materials.

Significance

The negative result presented here leads to rethink how inference based on cross-validation are used. It opens the way to designing new grounded inference procedures based on CV. As mentioned by the authors, the simplicity of assumptions (linear model, Gaussian inputs, independent noise) strengthens this negative result: if inference is not possible for some algorithm in this very favorable setting, it is likely to be the case for more complex scenarios.

Originality

I am not an expert of the cross-validation literature. I am not aware of any similar approaches. To the best of my knowledge, I find this work original.

I have some minor comments below.

- It would be interesting to discuss with more details the content of the paper mentioned in the paragraph before the notations (page 2) as they appear to be the closest to the results you present. In particular, Bates et al. 2024 have other negative results for the linear model.
- In other works (as in Bayle et al. 2020), one reason explaining the bad behavior of CV is the absence of independence between the different steps of CV (due to the fact that one observation is used K times). Can you elaborate on this? Does your analysis shed some light on this phenomenon?
- Preliminaries (page 2), maybe mention that $Z_i = (X_i, Y_i)$ here.
- Definition 2.1 Is there any rationale for studying $\gamma(h_n)/\sigma^2(h_n)$?
- Assumptions. Is the analysis still valid with a constant coefficient in the regression model? Could you extend it to the setting where $\epsilon$ is not independent of $X$? Finally, most results/elements of proof assume that the matrix $X^\top X$ is invertible, which may not be the case in practice, especially in high dimension. Is it possible to remove this assumption? If not, it should appear clearly in the text.
- Equation (6) As defined in Eq. (5), $R_n$ is random in both the data set and the cross-validation folds. Is it still the case in Eq. (6), which would be non-standard for a CLT?
- I find the discussion in the last paragraph of Section 4 difficult to follow. In my opinion, it would be clearer to have an additional proposition which states explicitly the two settings you mention (the different asymptotic regimes for $\sigma^2$).
- Figure 2 and 3, a more explicit description of the left/right plot would be nice. This can be done by simply adding $h^{sing}$ on the left and $h^{diff}$ on the right or by adding an explicit sentence in the text. What is $r$ in the top right captions? Do you mean $k$?
- Concusion : ``An open question for the reader is whether...'' I am not fond of such sentences, especially when the reader is asked to develop an entire new theory/algorithm.

---

> ### Author Rebuttal · Authors · 2026-03-31
>
> Thank you for your careful review of our paper and for your positive and constructive feedback. We are delighted that you found our work precise and easy to follow and we respond below to each point raised in order.
>
> - (this also responds to the reviewer’s sole Key Question) We will add more details about why the focus of these papers is different from ours. For Bates et al. (2024) in particular, in their Section 5.2.1, they indeed consider the particular case of low-dimensional linear regression with features and the noise terms sampled from the standard Gaussian as well, but the observation they make regarding CV confidence intervals is that the asymptotic regime of valid coverage kicks in later than it does for Nested CV and it appears they are looking at single algorithm evaluation, while we have demonstrated that it can not kick in at all in the model comparison setting for CV even though it does for single algorithm evaluation.
>
> - Indeed, the dependence between different steps of CV is the critical aspect that requires the relative stability condition in that paper. The key takeaway from our paper is that this assumption should not be taken lightly, as it can fail in even apparently well-behaved settings, particularly in the case of model comparison.
>
> - We will add this notational clarification about $Z_i$.
>
> - The reason for us to focus on this ratio, or more precisely its scaled version with $n$, is the key role it plays in a sufficient condition for valid inference with CV as derived in Bayle et al. (2020). The advantage of using $\gamma(h_n)$ is that it is one of the least restrictive notions of algorithmic stability in the literature, allowing for broad applicability for single algorithm evaluation. But $\gamma(h_n)$ is not scale-invariant, which is why the sufficient condition in Bayle et al. (2020) normalizes by $\sigma^2(h_n)$, since the stability needs to be sufficiently well-behaved *on the scale* of the actual fluctuations of the terms being averaged. Thus, this relative definition of stability is the right one to consider, although the denominator only really matters when it goes to zero asymptotically, which won’t often be the case for single algorithm evaluation but is much more likely in the case of model comparison of two algorithms, hence its emphasis in our paper.
>
> - Can the reviewer please clarify what is meant by a “constant coefficient in the regression model”? If you mean that $\beta^\star$ does not depend on $n$, then this is already the case in our analysis. If you mean that the regression model has an intercept, then we are nearly certain the analysis is still valid, but have not studied that case. Generally the intercept is not penalized in soft-thresholding/Lasso/ridge, and the penalty plays the key role in determining relative (in)stability, so we expect that the intercept will not play a role in determining relative (in)stability. Extending to the setting where $\varepsilon$ may depend on $\mathbf{X}$ or the high-dimensional setting are interesting directions, but as far as we can tell, these extensions would not be straightforward, so we prefer to leave them for future work. Note that since our asymptotic regime is low-dimensional, $n/p$ goes to infinity, in which case we believe it would very rarely be the case, even for non-Gaussian covariate entries, that $\mathbf{X}^\top \mathbf{X}$ is not invertible. Also, ST is only defined when $\mathbf{X}^\top \mathbf{X}$ is invertible, so that assumption cannot be removed from our ST results. We will add a comment about it in the text.
>
> - It is also the case in Eq. (6) indeed, making it a different flavor of CLT compared to the classic CLT, but still very relevant to construct valid confidence intervals when it holds, and natural as $R_n$ has many similarities with $\hat R_n$ and both are centered around the expected prediction error $\mathbb{E}[h_n(Z_0, Z)]$.
>
> - We apologize for the confusion, to make this clearer to readers, we will bring Lemma H.2 out of the appendix and state it formally here, as it covers exactly what the reviewer is asking for.
>
> - We will add a sentence to clarify that the left panel corresponds to $h_n^{\textrm{sing}}$ while the right one corresponds to $h_n^{\textrm{diff}}$. As for $r$, we defined it to be the relative loss stability $\frac{n \cdot \gamma(h_n)}{\sigma^2(h_n)}$, while $k$ is the number of folds in CV.
>
> - We apologize for our phrasing, and will remove the words “for the reader” from that sentence.

---

> > ### Author Rebuttal · Reviewer_m4TW · 2026-04-01
> >
> > Thank you for your detailed response.
> >
> > I am still convinced that it is an excellent paper and I will maintain my score.
> >
> > NB: By a constant coefficient, I meant a non-zero intercept. It could be interesting to pinpoint this (small) limitation (no intercept) in the paper.

---

> > > ### Author Response · Authors · 2026-04-08
> > >
> > > Thank you very much again for your time reviewing our paper and for your constructive feedback.

---

### Official Review · Reviewer_tfyt · 2026-02-27

**Soundness:** 4
**Presentation:** 4
**Significance:** 3
**Originality:** 4
**Overall Recommendation:** 5
**Confidence:** 4

**Summary:**

This paper proves that, for soft-thresholded and Lasso linear regression, cross validation for model comparison fails to generate relatively stable comparisons between instances of the regression algorithm with similar regularization parameters. This occurs even though the assessment of a single algorithm is relatively stable. Towards this goal, the authors also present a definition of relative stability (equivalent to a sufficient condition for the CV central limit theorem in Bayle et al. 2020) -- an algorithm or algorithm comparison is defined to be relatively stable if the ratio of its loss stability and the variance of the expected test loss over training datasets is $o(1/n)$.

The authors also present experiments on ST and Lasso, where they demonstrate the asymptotic behaviors of the loss stability and variance, and show that CV underestimates the variance of model comparison. They also show that fully parameterized regression is actually relatively stable, as well as ridge regression.

The paper concludes with a generic method to construct a relatively stable for individually relatively stable algorithms.

**Compliance With Llm Reviewing Policy:**

Affirmed.

**Final Justification:**

The authors addressed all of my weaknesses and questions satisfactorily, and upon reading the review and response to Reviewer m4TW I feel that I better understand the role of the ratio studied in the paper, and that I better understand the results in general. I raised my confidence score from a 3 to a 4 to reflect this.

**Key Questions For Authors:**

1. After the presentation of Theorem 3.1, you mention that the stringent assumptions only serve to strengthen your results. Is it possible to include one or two experiments loosening the assumptions and showing that the relative stability for CV model comparisons indeed only gets worse?

2. Do you expect similar results to hold in classification settings?

3. What is the key difference between Lasso and Ridge regression that makes CV model comparison relatively unstable for one and relatively stable for the other?

**Limitations:**

yes

**Strengths And Weaknesses:**

## Strengths

1. The theoretical results are presented very clearly and were made easy to understand by the supporting text.

2. The paper is clear about the nature of the results presented, their precise situation in the literature, and exactly what the results mean. There is no overstatement or extrapolation as far as I can tell.

3. The experiments were helpful to internalize the theoretical results and their consequences. It might be nice to include an additional visualization showing the estimated confidence intervals, as well as your corrected confidence intervals from Proposition 6.1.

## Weaknesses

1. Given this paper's focus on practitioners using CV to compare models, I think that this paper would have benefited from one or two simple experiments showing that CV for model comparison (in the Lasso regression setting) is relatively unstable for one or two real world datasets.

## Note

I skimmed the proofs and did not find any glaring flaws, but I admittedly lack the expertise and knowledge of statistics literature to verify them in detail.

---

> ### Author Rebuttal · Authors · 2026-03-31
>
> Thank you for your careful review of our paper and for your positive and constructive feedback. We are delighted that you found our work very clear and we respond below to the points raised.
>
> **Strength 3**: Following your suggestion, we will add two plots displaying coverage and width, as functions of the sample size, for the confidence intervals that are not valid, but would be if stability and thus normality held, and for the confidence intervals proposed in Proposition 6.1.
>
> **Weakness 1**: Our use of synthetic data was due to the need to compute properties of the data-generating distribution, such as $R_n$ and $\gamma(h_n)$, in order to evaluate our results. In real data, these properties are unknown, and hence we would be unable to evaluate stability or how close the distribution of interest is to normality with the targeted variance.
>
> **Question 1**: Apologies for the confusion, but we did not mean to imply that in less well-behaved settings the relative instability would necessarily be *worse*, just that it is likely to still be present (perhaps worse, but perhaps better) in other settings given that it is present already in such a well-behaved setting. As for experiments loosening the assumptions, we note that our Lasso experiment already uses a penalty that is chosen via CV (inner CV, not to be confused with the outer CV that is used to estimate/infer $R_n$), which places it beyond the setting in our theory, which assumes the penalty is chosen in a non-data-dependent way. As for even more complicated settings, unfortunately in general it is not easy to evaluate the loss stability of an algorithm, even in a simulation, due to it being an expectation of a nonlinear function of conditional expectations. As shown in Lemma L.2 of our paper’s appendix, we needed to derive computable unbiased estimators for the loss stability and these were specific to our linear model setup, so in general we do not know how to do this for other setups. Lemma L.1 shows that a more general formula is available for $\sigma^2(h_n)$, but unfortunately we need to compute both $\sigma^2(h_n)$ and $\gamma(h_n)$ to know the relative loss stability.
>
> **Question 2**: Purely as conjecture, we do believe similar results would hold in classification settings, as the instability seems to be due to non-smoothness in $h_n$ as a function of the data, which is equally applicable to classification loss.
>
> **Question 3**: As just alluded to, we believe that the non-smoothness in $h_n$ as a function of the data plays a role in the relative instability. In particular, soft-thresholding (ST), and similarly for the Lasso, is non-differentiable as a function of the OLS coefficient estimate right at the point where a coefficient gets set to zero, which is why we need at least one true coefficient to be zero for relative instability to hold (since ST/Lasso are consistent and hence for any non-zero coefficients, they would asymptotically never set them to zero so this non-differentiable point wouldn’t come into play). Similarly, ridge regression is a smooth function of the data everywhere. With just the setups considered in this paper, it is hard to make general statements or even compelling conjectures about generally what might drive relative (in)stability, but this was our understanding within the setups we were able to analyze rigorously.

---

> > ### Author Rebuttal · Reviewer_tfyt · 2026-04-01
> >
> > I thank the authors for their thoughtful response and the interesting note regarding the non-smoothness of $h_n$ as a factor in relative instability. I think that it would be interesting for a practitioner to be able to perform some test to determine whether their CV comparison method is stable, but I wholeheartedly acknowledge that this is out of scope for this paper. I maintain my confidence that this paper is worthy of acceptance.

---

> > > ### Author Response · Authors · 2026-04-08
> > >
> > > Thank you very much again for your time reviewing our paper and for your constructive feedback.

---

### Official Review · Reviewer_pY7o · 2026-03-11

**Soundness:** 4
**Presentation:** 3
**Significance:** 3
**Originality:** 4
**Overall Recommendation:** 5
**Confidence:** 3

**Summary:**

The paper studies the validity of cross-validation when comparing the predictive performance of two learning algorithms, focusing specifically on inference of the difference in mean squared errors for two regularized linear model with different tuning parameters. The key finding is that even when two models are individually stable, their difference in test errors may fail to satisfy the required stability conditions, leading to invalid CV-based confidence intervals for performance differences. Theoretical results are established in general settings and illustrated through concrete examples.

**Compliance With Llm Reviewing Policy:**

Affirmed.

**Final Justification:**

Authors have done a good job in this work. I will recommend accept.

**Key Questions For Authors:**

None

**Limitations:**

Yes

**Strengths And Weaknesses:**

Strength:

Overall, the paper provides a clear theoretical insight into a widely used methodology and highlights a previously underappreciated limitation of cross-validation-based inference. The theoretical analysis is rigorous, and illustrated through one of the most canonical examples where CV is used in practice.

Weaknesses:

The main weakness/limitation I can see is that the inference for the difference in test errors between two algorithms may not be as important as actually determinining the relative order of the two algorithms.

---

> ### Author Rebuttal · Authors · 2026-03-30
>
> Thank you for your careful review of our paper and for your positive and constructive feedback. We are delighted that you found our work rigorous and theoretically insightful.
>
> Regarding the importance of determining the relative order of the two algorithms, we completely agree, and indeed this is exactly how CV confidence intervals for model/algorithm comparison are generally used: if the confidence interval lies entirely above zero, then the first algorithm can be concluded to be better than the second algorithm with statistical significance, if the confidence interval lies entirely below zero, then the second algorithm can be concluded to be better than the first algorithm with statistical significance, and if the confidence interval contains zero, then the relative order of the two algorithms cannot be determined with statistical significance. Our results, about when such a confidence interval is valid or not, thus directly imply when the relative order of the two algorithms can be validly determined with statistical significance.

---

> > ### Author Rebuttal · Reviewer_pY7o · 2026-04-03
> >
> > The authors have answered my question. I have left the scores unchanged.

---

> > > ### Author Response · Authors · 2026-04-08
> > >
> > > Thank you very much again for your time reviewing our paper and for your constructive feedback.

---

### Official Review · Reviewer_wDr5 · 2026-03-12

**Soundness:** 3
**Presentation:** 3
**Significance:** 2
**Originality:** 3
**Overall Recommendation:** 4
**Confidence:** 3

**Summary:**

The paper examines the instability of model comparison using cross validation, focusing on the lasso and soft-thresholding estimator as penalized linear regression methods. These models are shown to have stabilities, allowing for appropriate uncertainty quantification in cross validation. However, when it comes to comparison of these stable models with slightly different tuning parameter values, the paper presents a negative result that the difference of losses for the two estimators/models could be relatively unstable. The theoretical results for the instability of model comparison are further illustrated with numerical examples.

**Compliance With Llm Reviewing Policy:**

Affirmed.

**Final Justification:**

While the rebuttal answered my questions adequately, there was no new information that would change my initial assessment of the paper. So, I will keep my original score.

**Key Questions For Authors:**

Q1. Do the main results in the paper about lasso and soft thresholding estimator have general implications beyond those specific methods?

Q2. According to the description on p.6 (line 319-322 in the right column), relative loss stability in both individual form and comparison setting depends on the true parameter vector. Could you provide some insights into why sparsity of the true parameter vector would matter in determining relative stability?

Q3. How is this instability related to the well known result in model comparison that overfitting models are more difficult to differentiate from the true model than underfitting models?

**Limitations:**

Yes.

**Strengths And Weaknesses:**

Soundness:

Borrowing the notion of relative loss stability from the literature of algorithmic stability and cross validation, the paper extends it to the difference in the loss for comparison of two models. The theoretical results obtained for lasso and soft-thresholding shed some light on limitations of comparing such models using CV. On a high level, when we take a difference between two related terms of similar order, the difference is expected to have a different stochastic order than the individual terms. I wonder if some modification of normalization of loss stability (n/variance) might be needed when it comes to loss difference.

Presentation:

The paper is overall clearly written and well organized. Section 4 with a different definition of n (not as a sample size but as the size of training sets used in CV) is a bit confusing to read. The numerical examples add clarity on the theoretical findings.

Significance:

The main results on lasso and soft-thresholding estimator seem interesting, and the message conveyed through those specific examples sounds important. However, as with many stylized theoretical results, the scope of the applicability of the results seems to be quite narrow. It is unclear what general implications they have. The proposition at the end of the paper aims at something more general as a remedy, but it is based on a loose union bound, which will be very conservative, thus not so practically useful.

Originality:

Extension of the relative loss stability to model comparison and the negative results about the two penalized linear regression methods seem novel to the best of my knowledge.

---

> ### Author Rebuttal · Authors · 2026-03-30
>
> Thank you for your careful review of our paper and for your positive and constructive feedback. We respond below to your comments and questions.
>
> **Soundness**: Indeed, the difference in stochastic order between individual terms and their difference is exactly why we emphasize the critical importance of the *relative* loss stability in this paper, as the (absolute) loss stability often *improves* from the individual terms to their (much smaller) difference. As you pointed out, the key to the relative loss stability is the normalization by variance, which we show in our paper reveals that even though the (absolute) loss stability for the difference in terms is often smaller than for the individual terms, in some settings (soft-thresholding and the Lasso) the variance term gets smaller even faster, so that the relative loss stability actually gets worse, leading to relative *in*stability.
>
> **Presentation**: We agree the definition of $n$ is unusual in Section 4, but we found that making it the full sample size causes all the earlier sections to be harder to read, and as these earlier sections are more the focus of the paper than Section 4, we chose the current definition to maximize legibility.
>
> **Significance/Q1**: Although our examples are specific and stylized, we claim they are not contrived or pathological but are generally considered well-behaved and regular (indeed we show that relative stability holds for our examples for individual algorithms), and note that our results are *negative* for relative stability and CV asymptotics of model comparison in these examples. Thus, the general implication is that practitioners should not generally expect relative stability and standard CV asymptotics to hold for practical settings/methods.
>
> **Q2**: As relative stability is an asymptotic property, what matters is the asymptotic behavior of the coefficient estimate as the sample size goes to infinity. Both soft-thresholding (ST) and the Lasso are consistent estimators, but they behave differently asymptotically for coefficient entries that are exactly zero and entries that are non-zero, as entries that are non-zero will asymptotically never be shrunk to zero. Shrinkage to zero plays a key role in our proof of relative instability, so we need at least one zero in the true coefficient vector, or else, at least asymptotically, no coefficient entries are shrunk to zero.
>
> **Q3**: We are not sure about the connection between our relative instability results and known results about underfitting/overfitting. While we show relative instability holds for ST/Lasso in sparse settings, we show it does not hold for ST/Lasso in dense settings, or for ridge regression in any settings, and it is not clear to us that underfitting/overfitting is what differentiates between the cases when relative instability holds or doesn’t. Related to our response to Q2 above, we believe the differentiator has more to do with the smoothness of the estimator as a function of the data.

---

> > ### Author Rebuttal · Reviewer_wDr5 · 2026-04-02
> >
> > I find the authors' response adequate, reflecting their current understanding of the problem and results. As there is no new additional information that would significantly change my initial assessment of the paper, I will maintain my score.

---

> > > ### Author Response · Authors · 2026-04-08
> > >
> > > Thank you very much again for your time reviewing our paper and for your constructive feedback.

---

### Decision · Program_Chairs · 2026-04-30

**Decision:**

Accept (spotlight)

**Comment:**

The paper provides rigorous theoretical insights into the widely used cross-validation method, highlighting a previously underappreciated limitation. Reviewers acknowledged the work's originality, specifically its novel extension of relative loss stability to model comparison and its technical analysis of lasso and soft-thresholding estimators. The (negative) results demonstrate the failure of CV-based inference even under simple and favorable settings. The presentation is of high quality. The paper should clearly be accepted.